# Efficient Generalization
# with Distributionally Robust Learning

**Soumyadip Ghosh, Mark S. Squillante**
Mathematical Sciences, IBM Research AI
Thomas J. Watson Research Center
Yorktown Heights, NY 20198, USA
`ghoshs, mss@us.ibm.com`

**Ebisa D. Wollega**
Department of Engineering
Colorado State University-Pueblo
Pueblo, CO 81001, USA
`ewolleg@csupueblo.edu`

## Abstract

Distributionally robust learning (DRL) is increasingly seen as a viable method to train machine learning models for improved model generalization. These minimax formulations, however, are more difficult to solve. We provide a new stochastic gradient descent algorithm to efficiently solve this DRL formulation. Our approach applies gradient descent to the outer minimization formulation and estimates the gradient of the inner maximization based on a sample average approximation. The latter uses a subset of the data sampled without replacement in each iteration, progressively increasing the subset size to ensure convergence. We rigorously establish convergence to a near-optimal solution under standard regularity assumptions and, for strongly convex losses, match the best known $O(\epsilon^{-1})$ rate of convergence up to a known threshold. Empirical results demonstrate the significant benefits of our approach over previous work in improving learning for model generalization.

## 1 Introduction

Consider a general formulation of the distributionally robust learning (DRL) problem of active interest. Let $\mathbb{X}$ denote a sample space, $P$ denote a probability distribution on $\mathbb{X}$, and $\Theta \subseteq \mathbb{R}^d$ denote a parameter space. Let us define $L_P(\theta) := \mathbb{E}_P[l(\theta, \xi)]$ to be the expectation with respect to (w.r.t.) $P$ of a loss function $l : \Theta \times \mathbb{X} \to \mathbb{R}$ representing the estimation error for a learning model with parameters $\theta \in \Theta$ over data $\xi \in \mathbb{X}$. Further define the worst-case expected loss function $R(\theta) := \mathbb{E}_{P^*(\theta)}[l(\theta, \xi)] = \sup_{P \in \mathcal{P}}\{L_P(\theta)\}$, which maximizes the loss $L_P$ over a well-defined set of measures $\mathcal{P}$. Letting $P_b$ denote a base distribution, this set often takes the form $\mathcal{P} = \{P \,|\, D(P, P_b) \le \rho, \int dP(\xi) = 1, P(\xi) \ge 0\}$ where $D(\cdot, \cdot)$ is a distance function on the space of probability distributions on $\mathbb{X}$ and the constraints limit the feasible candidates to be within a distance of $\rho$ from $P_b$. We seek to find parameters $\theta \in \Theta$ that solve the DRL problem formulated as

$$R(\theta_{rob}^*) = \min_{\theta \in \Theta}\Big\{ R(\theta) \Big\} = \min_{\theta \in \Theta}\Big\{ \sup_{P \in \mathcal{P}}\{L_P(\theta)\} \Big\}, \tag{1}$$

for a given $\mathbb{X}$ and $\mathcal{P}$. In practice, training with DRL amounts to dynamically reweighing the data using the inner optima $P^*(\theta)$ at any value the parameters $\theta$ take over the space $\Theta$. The inner maximization problem sets these weights to emphasize data that experience high loss at $\theta$. This reweighing approach arises from solid theoretical foundations and can provide strong guarantees on model generalization.

**DRL and Model Generalization:** In machine learning, optimal values for the model parameters $\theta$ are calculated from a finite training dataset (of size $N$) and this model is then used for inference over other test datasets, all of which are typically assumed to be identically distributed [33, 34]. The equal-weight *empirical* distribution $U_N = \{1/N\}$ over the finite training dataset is the non-parametric maximum likelihood estimator [23] of the (unknown) distribution underlying the datasets.

35th Conference on Neural Information Processing Systems (NeurIPS 2021).

Let $\theta_{erm}^*$ denote the minimizer of the empirical loss $L_{U_N}(\cdot)$ over $\Theta$. In real-world applications, any two finite datasets sampled from the same underlying distribution can violate the identical distribution assumption, leading to poor *generalization* when using $\theta_{erm}^*$ over other datasets [24]. Popular model selection techniques, such as cross-validation [32], seek to improve the estimation error between training and testing datasets, but they can be computationally prohibitive and lack rigorous guarantees.

With roots in non-parametric statistics [23] and optimization [28, 7], several studies [20, 2, 7, 21, 19] have proposed as an alternative approach the DRL formulation (1) using the empirical distribution $U_N$ over the finite training dataset as the base distribution $P_b$. This alternative approach explicitly treats the ambiguity in the identity of the true (stationary but unknown) distribution, denoted by $P_0$. In general, we know that $U_N$ is not equal to $P_0$ and it is highly likely that, at $\theta_{rob}^*$, the worst-case distribution $P^*(\theta_{rob}^*)$ is not equal to $P_0$. However, for Wasserstein distance metrics and an appropriately chosen value of $\rho$ in the set of measures $\mathcal{P}$, Blanchet et al. [2] show that $\mathcal{P}$ contains a distribution $P$ whose relevant loss function characteristics are the same as those for the true (unknown) distribution $P_0$ with high probability. A broad guideline is provided in [2, 21] such that these high probability guarantees are achieved by setting $\rho = O(\sqrt{d/N})$ for binary classification with logistic models. The DRL approach thus holds great promise and a general theory is actively being pursued.

**Goal:** Our main objective is to solve the DRL problem (1) in an efficient manner to ensure it is a viable alternative approach for model generalization. The primary difficulty is the minimax formulation, particularly the inner maximization problem. While its solution may be explicitly available in some cases – e.g., constraining $\mathcal{P}$ by certain Wasserstein distance metrics admits an explicit characterization of the robust objective function $\mathbb{E}_{P^*(\theta)}[l(\theta, \xi)]$ [2, 30, 8, 4, 5] – these reductions do not hold in general for all interesting Wasserstein distance metrics and they require solving a convex non-linear program [5]. Namkoong and Duchi [21] show that the inner maximization can be efficiently solved (see Section 2) under $\chi^2$-divergence constraints. Hence, we focus on the entire general class of $\phi$-divergence distance functions: $D_\phi(P, P_b) = \mathbb{E}_{P_b}[\phi(\frac{dP}{dP_b})]$, where $\phi(s)$ is a non-negative convex function having the value 0 only at $s = 1$. Members of this class include the modified $\chi^2$ divergence, with $\phi(s) = (s-1)^2$, and the Kullback-Leibler (KL) divergence, with $\phi(s) = s \log s - s + 1$. Defining the vector $P := (p_n)$ of dimension $N$ and setting the base $P_b$ to the uniform empirical distribution $U_N$, we then have that the loss function and constraint set $\mathcal{P}$ are given by $L_P(\theta) = \sum_{n=1}^N p_n l(\theta, \xi_n)$ and $\mathcal{P} = \{P \mid \sum_{n=1}^N p_n = 1, p_n \geq 0, \forall n, D_\phi(P, U_N) = \frac{1}{N} \sum_{n=1}^N \phi(N p_n) \leq \rho\}$, respectively.

**Related Work:** Assuming convex loss functions $l(\cdot, \xi_n)$, the DRL problem (1) is convex in $\theta$. Ben-Tal et al. [1] consider the typical Lagrangian dual algorithm used for this convex-concave case; see (10) in the supplement. Such a reformulation renders a standard stochastic optimization problem, which they solve by applying classical stochastic gradient descent (SGD) methods. However, when $l$ is non-convex, this approach cannot guarantee convergence to global optima. Moreover, Namkoong and Duchi [20] observe that the characteristics of certain dual variables can cause instability in SGD.

Namkoong and Duchi [20] therefore propose an alternative approach, for convex losses $l(\theta, \xi)$, that interleaves stochastic mirror-descent steps in each of the $\theta$ and $P$ variables. This requires a computationally demanding step that samples an $N$-sized non-uniform pmf, which requires $O(N)$ computational effort, and is only slightly smaller than the effort needed to solve the inner maximization problem *exactly* (see below). Coupled with the iterations in a composite dimension $d + N$, this results in slow convergence, as observed in our empirical results in the supplement. To rectify this, Namkoong and Duchi [21] propose to directly obtain the optimal $P^*(\theta)$ that defines $R(\theta)$, namely solving (1) as a large deterministic problem that is well-defined for finite $N$. For $\chi^2$-divergence, they show that the inner maximization can be reduced to two one-dimensional root-finding problems, which can be solved via bisection search, thus requiring an expensive $O(N \log N)$ computational effort (see Proposition 2) at each iteration. They also establish a guarantee that the deterministic algorithm for convex losses $l$ converges to an $\epsilon$-optimal solution with $O(N \epsilon^{-2})$ computational effort.

Recent work [10, 15] seeks to reduce the computational effort of this costly full-gradient method by developing an algorithm to efficiently estimate the gradient of the inner maximization problem through a multi-level Monte Carlo randomization procedure introduced by Giles [12]. Although there are important differences in the two algorithms and the theoretical arguments supporting each algorithm and its parameters (see the discussion below), both approaches hold great promise in theory especially for large datasets where, for convex losses $l$, Levy et al. [15] establish a computational effort complexity of $O(\epsilon^{-3})$ under their algorithm and we establish a computational effort complexity of $O(\epsilon^{-2})$ under the algorithm in [10]. Despite this great promise, our analysis and experiments

expose two adverse factors. Each iteration is more expensive because it solves multiple inner maximization problems. At the same time, the algorithm inherently produces an increase in the variance of the gradient estimator of the robust loss as a result of the added randomization of the Giles estimator; this can negatively impact both the computational effort and solution quality of the overall approach.

**Our Contributions:** Our main focus is to devise an algorithmic approach and theoretical results to efficiently solve DRL formulations and to improve generalization more consistently and more quickly than alternative methods. We consider in Section 2 a new SGD algorithm, tailored to large DRL problems, where our Algorithm 1 significantly reduces the expense of solving the inner maximization over the full training dataset by *subsampling* the support of $P$ from the (finite) training dataset in the iterates of the algorithm and estimating the robust loss via a sample average approximation. Subsampling (mini-batching) typically works well with SGD because the gradient estimates are unbiased. However, in the DRL context, our Theorem 3 shows that subsampling produces a bias in the estimation of the robust loss gradient because, with fixed mini-batch subsampling, chances are high that critical data that suffer high loss will be missed, leading to an optimistic estimation of the robust loss; empirical results (in the supplement) illustrate the important impact of this bias.

Namkoong and Duchi [21] bypass this important issue of bias by assembling the full-data gradient, whereas the Giles estimator approach of Levy et al. [15] and Ghosh and Squillante [10] randomizes the choice of the mini-batch size over the entire dataset. Although Levy et al. [15] provide an analysis of the bias induced by mini-batching, our algorithm is fundamentally different in that we are the first to solve the DRL formulation by assembling the mini-batch through sampling *without replacement* from the training dataset [11], whereas in strong contrast sampling *with replacement* is employed in [20, 21, 15]. Our alternative Giles estimator approach [10], devised independently of and simultaneously to the Giles estimator approach of Levy et al. [15], also employs sampling *without replacement*. While our general theory in the without-replacement sampling approach is harder to establish because the individual samples in the mini-batch are not distributionally independent, our analysis establishes stronger theoretical results that show our without-replacement sampling approach herein is able to match the best-known convergence guarantees of SGD. Thus the estimation is significantly improved for the same computational effort by only considering unique samples $\xi$ and their losses $l(\theta, \xi)$ in approximating $R(\theta)$. Indeed, an $M$-sized set sampled with replacement has on average only about $N(1 - e^{-M/N})$ unique support points (see the supplement); this issue leads Levy et al. [15] to show that the variance of their $R(\theta)$ estimator may not vanish with increasing batch size, thus contributing to their worse computational effort complexity which can not achieve the best-known SGD convergence guarantees.

Our Algorithm 1 reduces the identified inherent bias by *progressively increasing* the subsample size with the iterates up to the maximum size $N$. This represents the first such approach to simultaneously reduce bias and decrease the computational burden of solving large DRL formulations, together with the first analysis of a general progressively increasing subsampling-based approach in the context of DRL and bias reduction. Shamir [26] considers sampling with and without replacement, but in the context of standard (single-level) SGD, showing that SGD without replacement achieves similar performance on the same order as SGD with replacement, which is in strong contrast to our corresponding results in the DRL setting. We establish in Theorem 4 that convergence to a close neighborhood of a local optimal solution is assured, even for non-convex losses $l$, provided that the subsamples increase at a certain minimal rate. We further extend our analysis for strongly convex losses $l$ in Theorem 6 to show that our algorithm achieves $O(\epsilon^{-1})$ effort to reduce the optimality gap to $\epsilon$, matching the best SGD rates known for stochastic optimization [22]. The dependence on $N$ is via poly-log terms. This holds for all $\epsilon \geq \bar{E}^* = O(N^{-(1-\delta)})$ where $\delta$ is a small positive value. Hence, for strongly convex losses, we match the best rates of standard SGD w.r.t. $\epsilon$, like [21] (but without their $O(N)$ dependence), with only poly-log $N$ dependence, like [15] (but without their worse $\epsilon$ order). A forthcoming article extends these theoretical results to the case of convex losses $l$. Our theoretical results guide the setting of key algorithm parameter values to realize this optimally balancing of the fundamental tradeoff between computational effort and stochastic error. Additional theoretical results, all proofs and further technical details can be found in the supplement.

Our empirical results in Section 3 consider convex DRL formulations of binary classification problems. We compare the computation and generalization performance of our algorithm with the corresponding performance under the algorithms proposed in [20, 21, 15, 10] and under a regularized ERM formulation tuned via $k$-fold cross validation (CV), the standard practical approach for model generalization.

Our results show that Algorithm 1 attains equivalent or (often) superior generalization performance as $k$-fold CV and the other DRL methods. Meanwhile, our algorithm provides significant improvements in the computational effort required by the other DRL methods and orders of magnitude reductions in the computational effort required by $k$-fold CV. Furthermore, the key parameters of our DRL algorithm do not require fine tuning and are set based on our theoretical results, whereas the key parameters of the other DRL methods require fine tuning while also impacting both the solution quality and computation effort of the Giles approach. Additional empirical results, including the impact of key algorithmic parameters, and related technical details can be found in the supplement.

## 2  Algorithm and Analysis

Algorithm 1 presents our progressively sampled subgradient descent algorithm that follows SGD-like iterations for the outer minimization problem in (1) according to

$$\theta_{t+1} \;=\; \theta_t - \gamma_t \nabla_\theta \hat{R}_t(\theta_t) \;=\; \theta_t - \gamma_t G_t, \tag{2}$$

where $\gamma_t$ is the step size (or gain sequence or learning rate), $\hat{R}_t(\cdot)$ is a sample-average approximation of the robust loss $R(\cdot)$ from the inner maximization formulation over $D_\phi$-constrained $\mathcal{P}$, and $G_t := \nabla_\theta \hat{R}_t(\theta_t)$. Our algorithm uses progressively increasing sample sizes $M_t$ to estimate $G_t$. This view of (1) permits us to depart from the convex-concave formulations of Ben-Tal et al. [1] and consider non-convex losses $l$, as long as the subgradient $\nabla_\theta \hat{R}_t(\cdot)$ approximates the gradient $\nabla_\theta R(\cdot)$ sufficiently well. We first establish that the gradient $\nabla_\theta R(\cdot)$ exists, recalling that $R(\theta)$, as the optimal value of the inner maximization problem, is an extreme value function. Let us define the set $\Theta_\varnothing := \{\theta : l(\theta, \xi_{n_1}) = l(\theta, \xi_{n_2}), \; \forall n_1, n_2\}$ and, for a small $\varsigma > 0$, define the set $\Theta_{\varnothing,\varsigma} := \cup_{\theta_o \in \Theta_\varnothing} \{\theta : |\theta - \theta_o| < \varsigma\}$ to be the $\varsigma$-neighborhood of $\Theta_\varnothing$. We assume in Proposition 1 that the learning model precludes this neighborhood set in order to avoid a degenerate case in which the inner maximization objective function does not depend on the decision variables $p_n$ and the entire feasible set is optimal. The existence and form of $\nabla_\theta R(\theta)$ is then derived in part by exploiting Danskin's Theorem [29, Theorem 7.21].

**Proposition 1.** *Let the feasible region $\Theta$ be compact and assume $\Theta \subseteq \Theta^c_{\varnothing,\varsigma}$, for a small $\varsigma > 0$. Further suppose $\phi$ in the $D_\phi$-constraint has strictly convex level sets, and let $\rho < \bar{\rho}(N, \phi)$ with $\bar{\rho}(N, \phi)$ defined in the supplement. Then: (i) the optimal solution $P^*$ of $R(\theta) = \sup_{P \in \mathcal{P}}\{L_P(\theta)\}$ is unique, and the gradient is given by $\nabla_\theta R(\theta) = \sum_{n \in \mathcal{N}} p^*_n(\theta) \nabla_\theta l(\theta, \xi_n)$; and (ii) for all $\rho$, the gradient $\nabla_\theta R(\theta)$ is a sub-gradient of $R(\theta)$.*

---

**Algorithm 1** ProgressiveSSD$(\gamma, \{M_t\}, \theta_0, \rho)$

**Given**: Step size $\gamma$; Sample sizes $\{M_t\}_{t=1}^{\mathcal{T}}$, $M_{\mathcal{T}} = N$; Initial $\theta_0$; $D_\phi$ constraint $\rho$; small $\delta > 0$.

  **for** $t = 1, 2, \ldots, \mathcal{T}$ **do**
    {Sample $\mathcal{M}_t$ *without replacement*}
    $\mathcal{M}_t \leftarrow \varnothing$
    **for** $m = 1, \ldots, M_t$ **do**
      $\xi_m \sim \text{Uniform}(\mathcal{N} \setminus \mathcal{M}_t)$
      $\mathcal{M}_t \leftarrow \mathcal{M}_t \cup \{\xi_m\}$
    **end for**

    Assemble $\mathcal{Z}_t \leftarrow \{l(\theta_t, \xi_m), \; \forall m \in \mathcal{M}_t\}$
    $\rho_t \leftarrow \rho + c\left(\frac{1}{M_t} - \frac{1}{N}\right)^{(1-\delta)/2}$
    $P^*_t \leftarrow \text{InnerMax}(\mathcal{Z}_t, \mathcal{M}_t, \rho_t)$
    Set $G_t \leftarrow \sum_{m \in \mathcal{M}_t} p^*_{t,m} \nabla_\theta l(\theta_t, \xi_m)$
    Set $\theta_{t+1} \leftarrow \theta_t - \gamma G_t$
  **end for**
  **return** $\theta_{\mathcal{T}}$

---

**Algorithm 2** InnerMax$(\mathcal{Z}, \mathcal{M}, \rho)$

**Given**: loss values $\mathcal{Z}$; subsampled support $\mathcal{M}$; $D_\phi$ constraint $\rho$.

  $M \leftarrow |\mathcal{M}|$, base $P_b = \{\frac{1}{M}, \forall m \in \mathcal{M}\}$
  $\bar{z} \leftarrow \max_m \{z_m \mid z_m \in \mathcal{Z}\}$
  $\mathcal{M}' \leftarrow \{m \in \mathcal{M} : z_m = \bar{z}\}$ and $M' \leftarrow |\mathcal{M}'|$
  $P' \leftarrow \{\frac{1}{M'}\mathbb{I}\{m \in \mathcal{M}'\}, \; \forall m \in \mathcal{M}\}$
  **If** $D_\phi(P^*, P_b) \leq \rho$ **then**
    $P^* \leftarrow P'$ and **return** $P^*$
  **for** $\alpha \in [0, \bar{\alpha}]$ **do**
    **for** $\lambda \in [\underline{\lambda}, \bar{\lambda}]$ **do**
      $\mathcal{M}' \leftarrow \{m \mid \lambda \leq z_m - \alpha\phi'(0)\}$
      $P' \leftarrow \{\frac{1}{M}(\phi')^{-1}(\frac{z_m - \lambda}{\alpha}), m \in \mathcal{M}'\}$
      **If** $\sum_m p'_m = 0$, **then**
        $P^*(\alpha) \leftarrow P'$, and **break**
    **end for**
    **If** $D_\phi(P^*(\alpha), P_b) = \rho$, **then**
      $P^* \leftarrow P^*(\alpha)$ and **break**
  **end for**
  **return** $P^*$

---

Next, we construct the estimate $\hat{R}_t$ from the inner maximization problem restricted only to a relatively small subset $\mathcal{M}_t$ of the full dataset with size $|\mathcal{M}_t| = M_t$. Define $P := (p_m)$ of dimension $M_t$ and

define the objective coefficients $z_m := l(\theta, \xi_m)$. We then have

$$\hat{R}_t(\theta) \;=\; \max_{P=(p_m)} \sum_{m \in \mathcal{M}_t} p_m z_m \qquad \text{s.t.} \quad \sum_{m \in \mathcal{M}_t} \phi(M_t p_m) \le M_t \rho_t, \quad \sum_{m \in \mathcal{M}_t} p_m = 1, \; p_m \ge 0. \quad (3)$$

The uncertainty radius $\rho_t = \rho + \eta_t$ now changes with the iteration $t$ where $\eta_t$ represents a small additional inflation; Theorem 3 below motivates why this is included. Now suppose $P_t^*(\theta) = (p_{t,m}^*(\theta))$ is an optimal solution to (3). Then a valid subgradient for $\hat{R}_t(\theta_t)$ is obtained as an expression analogous to that in Proposition 1(i) under appropriate substitutions w.r.t. $\theta_t$, $P_t^*$ and $\mathcal{M}_t$. Writing the Lagrangian objective of (3) as $\mathcal{L}(\alpha, \lambda, P)$, with an explicit expression given in the supplement, we then have the optimal objective value $\hat{R}_t^*(\theta) = \min_{\alpha \ge 0, \lambda} \max_{\hat{p}_m \ge 0} \mathcal{L}(\alpha, \lambda, P)$; refer to [18]. The optimal primal and dual variables can then be obtained for various $\phi$ functions by the general Algorithm 2 to solve the Lagrangian formulations. Importantly, a worst-case bound on the computational effort required to obtain an $\epsilon$-optimal solution to (3) can be obtained.

**Proposition 2.** *For any $\phi$-divergence, Algorithm 2 finds a feasible primal-dual solution $(\tilde{P}_t^*, \tilde{\alpha}^*, \tilde{\lambda}^*)$ to (3) with an objective function value $\tilde{R}_t^*$ such that $|\hat{R}_t^*(\theta) - \tilde{R}_t^*| < \epsilon$ with a worst-case computational effort bounded by $O(M_t \log M_t + (\log \frac{1}{\epsilon})^2)$, where $\epsilon$ is a small precision parameter.*

The machine-precision $\epsilon$ does not relate to any formulation or algorithm parameters (e.g., $M_t, N, \rho$), and it is required because Algorithm 2 solves two one-dimension bisection searches in sequence. In the sequel we assume that $\epsilon$ is a fixed small value and Algorithm 2 returns the exact unique solution $(P_t^*, \alpha^*, \lambda^*)$ to problem (3), and that the computational effort is bounded by $O(M_t \log M_t)$.

## 2.1 Small-Sample Approximation of $\nabla_\theta R(\theta)$

In Algorithm 1, the gradient $\nabla_\theta R$ is approximated using the gradient expression in Proposition 1(ii) with the optimal $P_t^*$ for a small sample size $M_t$. This estimate is expected to suffer a bias because a subsample of the full dataset might miss data where loss is high when subsampling the dataset, and this leads to optimistic estimation of the robust loss $R(\theta)$ and its gradient. Let the mass vector $P^* = (p_1^*, \ldots, p_N^*)$ be the optimal solution to the full-data version of (3), i.e., with $M_t = N$, and let the mass vector $P_t^* = (p_1^*, \ldots, p_{M_t}^*)$ be the optimal solution when restricted to any subset $\mathcal{M}_t$.

**Assumption 1.** *The $\phi$-divergence satisfies uniformly for all $s$ and $\zeta < \zeta_0$ the continuity condition (for constants $\zeta_0, \kappa_1, \kappa_2 > 0$): $|\phi(s(1 + \zeta)) - \phi(s)| \le \kappa_1 \zeta \phi(s) + \kappa_2 \zeta$.*

The above condition, as described in [29], only allows for (local) linear growth in $\phi$, and it can be verified for many common $\phi$-divergences of interest including the modified $\chi^2$- and KL-divergences. Let $\mathbb{E}_t$ and $\mathbb{P}_t$ respectively denote expectation and probability w.r.t. the random set $\mathcal{M}_t$.

**Theorem 3.** *Suppose Assumption 1 and the assumptions of Proposition 1 hold, and further suppose the gradient $\nabla \hat{R}_t(\theta)$ is the optimal solution to (3) over a subsample $\mathcal{M}_t$ of size $M_t$ sampled uniformly without replacement from a set of size $N$. Define $\eta_t = c(\frac{1}{M_t} - \frac{1}{N})^{(1-\delta)/2}$ for small constants $c, \delta > 0$, and set the $D_\phi$-target in (3) to be $\rho_t = \rho + \eta_t$. Then, there exists a small positive $M'$ defined in the supplement and of order $o(N)$ such that, for all $M_t \ge M'$, the subgradient $\nabla_\theta \hat{R}_t(\theta)$ and full-gradient $\nabla_\theta R(\theta)$ satisfy for any $C < \infty$ and $1 - \bar{\tau}_t = O(\eta_t^{2\delta/(1-\delta)})$:*
$$\mathbb{P}_t(\eta_t^{-2} \|\nabla_\theta \hat{R}_t(\theta) - \nabla_\theta R(\theta)\|_2^2 \le C) \ge \bar{\tau}_t.$$

**Corollary 1.** *If the conditions for Theorem 3 are satisfied, then $\|\mathbb{E}_t[\nabla_\theta \hat{R}_t(\theta)] - \nabla_\theta R(\theta)\|_2^2 = O(\eta_t^2)$.*

The proof of Theorem 3 (in the supplement) includes exact expressions for the corresponding constants such as $\bar{\tau}_t$, whose dependence on the constant $C$ and on the magnitude of $\nabla_\theta R(\theta)$ is made explicit in Lemma 9. Theorem 3 provides a bound on the squared bias in approximating the true gradient $\nabla_\theta R(\theta)$ with the subsample estimate $\nabla_\theta \hat{R}(\theta)$ as a function of the sample size $M_t$. Levy et al. [15] develop an analogous bias result for the case when the subsample $\mathcal{M}_t$ is gathered through sampling support points $\xi_n$ *with replacement* by building on standard concentration bounds. In strong contrast, our $\mathcal{M}_t$ is sampled *without replacement* which, as previously motivated, is more practical in terms of avoiding repeated computation with sample points that get sampled multiple times. Unlike the with-replacement case, the samples within our $\mathcal{M}_t$ are no longer independent. Consequently, our analysis is built from the ground up starting with tools of probability related to sampling finite sets without replacement; see the start of Appendix A.3. Motivating the form of $\eta_t$, note that sampling a

subset of size $M$ uniformly without replacement from a larger finite set of size $N$ induces variance terms of the form $(\frac{1}{M} - \frac{1}{N})$ in place of the standard $\frac{1}{M}$.

To provide an outline of the proof, it starts by constructing $\tilde{P}^*$, a restriction of the (unique) $P^*$ onto the (random) subset $\mathcal{M}_t$ in the restricted problem (3), where $\tilde{p}_m^* \propto p_m^*$, $\forall m \in \mathcal{M}_t$. The assumptions ensure with probability at least $\bar{\tau}_t$ that $\tilde{p}_m^* \neq \mathbf{0}$. With the same high probability, $\tilde{P}^*$ is also a feasible solution to (3) when $\rho_t$ is inflated as assumed. Next, we establish that its objective function value is within $\eta_t$ of the optimum with high probability, which yields the desired result. The Corollary 1, which is the main form of Theorem 3 used in the sequel, follows from Theorem 17.4 in [13].

Note that the bias identified in Theorem 3 vanishes only with $M_t \nearrow N$ as $t \to \infty$. Since fixed bias in gradients violates a basic requirement for SGD [3, Section 4.3] that the gradient estimator $\mathbb{E}[\nabla_\theta \hat{R}_t(\theta)] = \Theta(\nabla_\theta R(\theta))$ (using standard time complexity notation [31]), then the convergence of (2) cannot be guaranteed when $M_t = M$ for all $t$, where $M < N$. To address this bias issue, in strong contrast with using the full-data gradient $\nabla_\theta R(\theta)$ [21] or randomizing the subset size $M_t$ over the entire dataset [15, 10], our algorithm chooses to grow the mini-batch size $M_t$ progressively with $t$. Bottou et al. [3, Section 5.1] show that even with unbiased gradient estimation, keeping step sizes $\gamma_t$ fixed and directly increasing the mini-batch size $M_t$ is more advantageous because (informally) the larger $M_t$ makes the gradient estimates more accurate. In our case, we grow the sample size in the SGD algorithm (2) to eliminate bias. Since this also provides a decrease in stochastic error as a consequence, it is no longer necessary to diminish the step size $\gamma_t$; indeed, doing so negates the benefits of the extra work in computing gradients using a larger $M_t$. Algorithm 1 therefore takes fixed-length steps $\gamma$. The maximum size $N$ is reached after a (large) finite number of iterations $\mathcal{T}$, at which point our algorithm reduces to a deterministic full-gradient optimization approach.

## 2.2 Convergence of Algorithm 1

Seen in the above light, for a finite $N$, the strictly increasing $M_t$ sequence will eventually end at an iterate $\mathcal{T} < \infty$ where $M_\mathcal{T} = N$, that is $\mathcal{T} := \inf_t\{M_t \geq N\}$. Hence, Algorithm 1 is not guaranteed to converge by the $\mathcal{T}$th iteration. We can nevertheless provide a finite-stop guarantee on the performance of the method over any loss function with Lipschitz gradients.

**Theorem 4.** *Suppose the constant step size $\gamma_t = \gamma$ satisfies $\gamma \leq \frac{1}{2L}$, a lower bound $R_{\text{inf}}$ exists for the robust loss function $R(\theta)$, $\forall \theta \in \Theta$, and the conditions of Theorem 3 hold. Then, the variance of the estimate $\nabla \hat{R}_t(\theta)$ calculated over the sampled-without-replacement $\mathcal{M}_t$ with subsample size $M_t$ obeys $\mathbb{E}[\|\nabla \hat{R}_t(\theta) - \mathbb{E}[\nabla \hat{R}(\theta)]\|_2^2] \leq C(\frac{1}{M_t} - \frac{1}{N})$. Further assume the gradient $\nabla_\theta R(\theta)$ is L-Lipschitz. Then, at termination,*

$$\sum_{t=1}^{\mathcal{T}} \|\nabla_\theta R(\theta_t)\|_2^2 \leq \frac{R(\theta_0) - R_{\text{inf}}}{\frac{\gamma}{2}(2 - L\gamma)} + C\frac{L\gamma + 1}{2 - L\gamma}\sum_{t=1}^{\mathcal{T}} \eta_t^2. \tag{4}$$

The first result in Theorem 4 establishes that the variance of $\nabla \hat{R}_t(\theta)$ obeys a bound which is as to be expected for sampling without replacement from any finite set [35]. Combining this with the bias (see Corollary 1) allows us to progressively decrease the mean squared error. This leads to the result (4) in Theorem 4, which shows that the sum of the gradients of $R(\theta_t)$ at iterates visited by the algorithm is bounded above, in particular, by $\sum_{t=1}^{\mathcal{T}}(\frac{1}{M_t} - \frac{1}{N})^{(1-\delta)}$. If this summation remains finite as $\mathcal{T} \to \infty$, then the upper bound of (4) remains finite, and thus the gradients $\|\nabla_\theta R(\theta_t)\|_2^2$ at the iterates converge to 0; in other words, the algorithm converges to a local optimal solution. The summation can converge for $M_t$ increasing moderately, such as at a polynomial rate.

Theorem 4 assumes that the gradient $\nabla_\theta R(\theta)$ of the robust loss is Lipschitz continuous. The gradients of such extreme value functions are in general not Lipschitz even if the objective function is Lipschitz. For example, a linear objective $l(\theta, \xi_n) = \theta^t \xi_i$ leads to $R(\theta) = \max_p \sum_i p_i \theta^t \xi_i$ and when maximized over a *polyhedral* constraint set it will not preserve the 0-Lipschitzness of the objective functions, because the optimal solutions $P^*$ are picked from the discrete set of vertices of the polyhedron and thus $\nabla_\theta R(\theta)$ is piecewise discontinuous. Our Proposition 1 assumptions yield an inner maximization with a non-zero linear objective over a *strictly convex* feasible set. The desired smoothness can then be obtained with some additional conditions on the loss functions $l(\theta, \xi_i)$. Proposition 5 provides one such condition where the Lipschitzness of $\nabla_\theta R(\theta)$ follows from

the Hessian of $R(\theta)$ being bounded in norm, which is often satisfied by common statistical learning losses, e.g., log-logistic and squared losses of linear models over compact spaces.

**Proposition 5.** *Assume the conditions in Proposition 1 hold. Further suppose that the Hessians $\nabla_\theta^2 l(\theta, \xi_n)$ exist, $\forall \theta$ and each $\xi_n$, and are bounded in Frobenious norm $\|\nabla_\theta^2 l(\theta, \xi_n)\|_F \leq L$, $\forall \theta, n$. Then, the robust loss also follows $\|\nabla_\theta^2 R(\theta)\|_F \leq M$ for some positive $M < \infty$.*

A key consideration then is to obtain $\theta_\mathcal{T}$ as close as possible to the minimizer $\theta_{\mathrm{rob}}$ that attains $R_{\mathrm{inf}}$. The tradeoff in (4) suggests that increasing $M_t$ aggressively will lead to smaller gradients at termination, but this will also increase the computational effort in each iteration. In the remainder of the section, we study this tradeoff under an additional assumption that the loss functions $l(\cdot, \xi)$ are strongly convex for each $\xi$, which we will show in Theorem 6 yields the strong convexity of $R(\theta)$. Consequently, it also provides a unique minimizer $\theta_{\mathrm{rob}}$ for (1) that satisfies $R_{\mathrm{inf}} := R(\theta_{\mathrm{rob}})$. We therefore seek a lower bound on how close the value of $R(\theta_\mathcal{T})$ and its gradient at termination of our progressively sampled Algorithm 1 can get to the unique optimal solution as a function of the sample growth sequence $\{M_t\}$, and also seek to establish convergence guarantees for the computational effort needed by our algorithm to attain a desired optimality gap larger than this lower bound.

Our notion of efficiency will be developed w.r.t. the total computational effort $W_t$ that is expended up until iterate $t$, where $W_t = \sum_{s \leq t} w_s$ and $w_s$ is the individual work in each iterate $s$. From the discussion following Proposition 2, we assume the relation $w_t = k_e M_t \log M_t$ holds for some constant $k_e$. Defining the ratio $\nu_t := M_{t+1}/M_t$ as the *growth factor* of the sequence $\{M_t\}$, we consider two important cases: (i) *Diminishing*-factor growth with $\nu_t \searrow 1$ as $t \uparrow$, e.g., the polynomial growth of $\nu_t = 1 + \frac{1}{t}$; and (ii) *Constant*-factor growth with $\nu_t = \nu > 1$, $\forall t$. Assuming $M_0 = 1$, the algorithm employs subsample set size $M_t = \lfloor \prod_{s=1}^{t} \nu_s \rfloor$ in iteration $t$, until a maximum of $\mathcal{T}$ iterations when $M_\mathcal{T} \geq N$. For constant growth sequences, we have $\mathcal{T} = \lceil \log(N)/\log \nu \rceil$, and we use the notation $\mathcal{T}(\nu)$ and $W_\mathcal{T}(\nu)$ to denote the total iterations and total computational effort as functions of $\nu$. Our final result characterizes guarantees on the rate at which the expected optimality gap $E_\mathcal{T} := \mathbb{E}_{\mathcal{T}-1}[R(\theta_\mathcal{T})] - R(\theta_{\mathrm{rob}})$ decreases w.r.t. $W_\mathcal{T}$. Recall that $\delta$ defines the parameter $\eta_t$ in Theorem 3, and denote by $E_0$ the optimality gap at the starting iterate. We drop the integrality requirement on sequences like $\lfloor \prod \nu_s \rfloor$ in our analysis of Theorem 6 to provide a clear and insightful exposition, but note that the conclusions remain unaltered.

**Theorem 6.** *Suppose all the conditions of Theorem 4 are satisfied and the loss functions $l(\theta, \xi_n)$ are c-strongly convex. Then the function $R(\theta)$ is c-strongly convex. Further suppose that $\gamma \leq \min\{\frac{1}{4L}, 4c\}$ is fixed. Then: (i) For diminishing growth sequences $M_t$, we have that $E_\mathcal{T} W_\mathcal{T} \to \infty$ as $\nu_t \to 1^+$; and (ii) For constant growth sequences, we have that the total effort $W_\mathcal{T}(\nu)$ is a decreasing function over $\nu \in (1, \infty)$, with $\lim_{\nu \to 1^+} (\nu - 1) W_\mathcal{T}(\nu) = k_e(N \log N - (N - 1))$. Moreover, $E_\mathcal{T}(\nu) \leq \bar{E}_\mathcal{T}(\nu)$ where $\bar{E}_\mathcal{T}(\nu)$ is an increasing function over $\nu \in (1, \infty)$ with its infimum $\bar{E}^* = \inf_\nu \bar{E}_\mathcal{T}(\nu) = (1/N^{1-\delta})(E_0 + 8CLc^2))$ attained as $\nu \to 1^+$. Finally, we have*
$$\lim_{\nu \to 1^+} W_\mathcal{T}(\nu) \ (\bar{E}_\mathcal{T}(\nu) - \bar{E}^*) = 8CLc^2(1-\delta)(1 - \tfrac{\gamma}{4c}) \cdot k_e \left( N^\delta \log N - (N^\delta - N^{-(1-\delta)}) \right).$$

Several key properties of our Algorithm 1 are apparent from Theorem 6 for the case of strongly-convex $R(\theta)$. To understand these results, first note (referring to the supplement for details) that if the full batch-gradient method is applied in each iteration ($M_t = N$), a strongly-convex objective $R$ would enjoy a linear (i.e., constant factor) reduction of size $(1 - \gamma/4c)$ in the error $R(\theta_t) - R(\theta_{\mathrm{rob}})$ for a step-size $\gamma$ chosen to satisfy the conditions of Theorem 6. The average optimality gap can be written as a sum of this deterministic error and an additional term representing the stochastic error induced by the subsampling of the support. The best convergence results for SGD methods on strongly convex objective functions ensure that $\epsilon$-expected-optimality-gap solutions are obtained with $O(\epsilon^{-1})$ work complexity [29]. Theorem 6(i) establishes that any general diminishing-factor growth of $M_t$ will lead to the stochastic error decreasing to zero much slower than the geometric drop in the deterministic error, and thus the stochastic error dominates. Consequently, there is a suboptimal reduction in the optimality gap $E_\mathcal{T}$ w.r.t. the total computational effort $W_\mathcal{T}$.

Constant factor sequences, however, can achieve a balance in the tradeoff between the rate of reduction in stochastic error and the drop in deterministic error. Note first that, for smaller $\nu > 1$, the sample size sequence $M_t$ grows more slowly, hence leading to larger terminating iteration $\mathcal{T}$ and total computational effort $W_\mathcal{T}$; an exact expression for $W_\mathcal{T}(\nu)$ is provided in the supplement for any $\nu \in (1, \infty)$. Consequently, $W_\mathcal{T}$ *falls* as the growth factor $\nu$ grows. This also leads to a poorer guarantee $\bar{E}_\mathcal{T}(\nu)$ on the attainable optimality gap and thus requiring $\nu \to 1^+$ to improve the

guarantees on $E_{\mathcal{T}}$. Theorem 6 characterizes an absolute limit in $\bar{E}^*$ on how much optimality gap reduction can be guaranteed in this regime because the algorithm takes only a finite $\mathcal{T}$ number of iterations, and shows $\bar{E}^* = O(N^{-(1-\delta)})$ independent of the step length $\gamma$. This theorem also leads to the following corollary that our algorithm attains any desired gap $\bar{E}^* + \epsilon$ with $O(\epsilon^{-1})$ effort.

**Corollary 2.** *Suppose all the conditions of Theorem 6 are satisfied and a solution with guaranteed expected optimality gap within $\bar{E}^* + \epsilon$ is desired. Then, there exists a $\nu_\epsilon \in (1, \infty)$ such that when Algorithm 1 is run with sample size sequence $M_t = \lfloor \nu_\epsilon^t \rfloor$, it terminates with $\mathcal{T}(\epsilon) = \lceil \log N / \log \nu_\epsilon \rceil$ steps and produces the desired solution with total computation effort $W_{\mathcal{T}}(\epsilon) = \frac{1}{\epsilon} \left( \kappa_1 \nu_\epsilon + \kappa_2 \nu_\epsilon (\nu_\epsilon^{1-\delta} - 1) + o(\nu_\epsilon^{1-\delta} 1) \right)$, where $\kappa_1, \kappa_2 \in (\eta, \infty)$ for a fixed $\eta > 0$ and for all $\epsilon \geq 0$. Moreover, $\nu_\epsilon \searrow 1$ as $\epsilon \to 0$.*

(The expression $a_n = o(b_n)$ denotes that $a_n/b_n \to 0$ as $n \to \infty$.) Expressions for the constants $\kappa_1$ and $\kappa_2$ are given in the proof of the corollary in the supplement. Since $\delta$ is an arbitrarily small chosen constant, $\kappa_1$ and $\kappa_2$ depend on $N$ only through poly-log terms. Hence, for such a desired $\epsilon$, our progressively subsampled algorithm for the DRL formulation matches the best rate results for the SGD family of algorithms for strongly convex stochastic optimization formulations.

## 3 Empirical Results

Numerous experiments were conducted to empirically evaluate our progressively sampled subgradient descent (PSSG) Algorithm 1, with the main objectives of reaching the optimal solutions of the DRL formulation and improving model generalization more consistently and more quickly than alternative methods. In addition, we seek to understand the characteristics of our PSSG method and assess the efficacy of the DRL approach in general as a viable alternative to regularized ERM in producing good model generalization. Specifically, we compare the performance of PSSG with that of competing algorithms to solve the DRL formulation (1), namely the full-support gradient (FG) algorithm of [21] and the multi-level Monte Carlo (Giles) algorithm of [15] and [10]; given the benefits we established here and in [10] for sampling without replacement over sampling with replacement, our empirical results focus on the multi-level Monte Carlo (Giles) algorithm of [10]. We also include in supplement B consideration of the standard SGD (fixed minibatch $M_t = M$) method to gauge the impact of the bias shown in Theorem 3; and further include consideration of the SGD method of [20] for a small dataset to demonstrate that the method was not competitive. These methods were all re-implemented since full source code was not made available with the corresponding publications.

Following [21], all examples use a logistic binary classification loss $l(\theta, (x, y)) = \log(1 + \exp(-y\theta^t x))$ where $x$ represents samples with $d$ features and $y$ represents the class labels $\pm 1$. All algorithms sampled the initial $\theta_0$ uniformly from the hypercube $[-1, 1]^d$. All experiments were implemented in Python 3.7 and run on a 16-core 2.6GHz Intel Xeon processor with 128GB memory. Additional empirical results and technical details can be found in the supplement.

**Generalization.** We investigate whether the DRL formulation (1) can improve the generalization of learning models against an ERM-trained model that is regularized via 10-fold CV. Experiments were conducted over 13 public-domain datasets as detailed in Table 1 with sizes ranging from $O(10^2)$ to $O(10^6)$. We include `MNIST`, a non-convex neural-network model, where we follow [15] and admit only the last logistic classification layer into the DRL formulation. Table 1 presents a comparison of the test misclassification produced by the DRL algorithms and the regularized ERM algorithm at termination for the 13 datasets. (The parameter settings for each algorithm are detailed below). The 95% confidence intervals (CIs) are calculated over 10 permutations of the datasets into training (80%) and testing (20%) sets. We highlight in bold the methods that produce a generalization error (within its CIs) that is clearly the best. As evident, at least one DRL method – always including PSSG – produces models of equal or better quality as the regularized ERM formulation for all datasets.

Recall that the DRL methods provide this level of performance by solving a single instance of the DRL formulation (1), thus avoiding the burdensome 10-fold CV enumeration. Table 1 provides the average CPU time in seconds recorded over the 10 permutations. The average time taken by the ERM 10-fold regularization in solving its formulation multiple times in a serial computing mode to identify the best regularization parameter (parameter $\lambda$ below) exceeds that taken on average to solve the DRL formulation by one to two orders of magnitude. The computation time of a single PSSG DRL run is of the same order as that of a single ERM run, as anticipated by Theorem 6 and its Corollary 2.

| Dataset | $\sqrt{\frac{d}{N}}$ | Test Misclassified (%) | | | | CPU Time (secs) | | | |
|---|---|---|---|---|---|---|---|---|---|
| | | FG | Giles | PSSG | ERM | FG | Giles | PSSG | ERM |
| adult* | 0.051 | 17.1±0.0 | 16.7±0.1 | 16.6±0.1 | 16.7±0.1 | 45 | 214 | 36 | 2542 |
| fabert[†] | 0.312 | **9.1±0.2** | 9.9±0.2 | **9.2±0.1** | 9.9±0.1 | 32 | 61 | 20 | 4128 |
| gina_agnostic[†] | 0.529 | **13.2±0.1** | 15.4±0.4 | **13.1±0.2** | 15.9±0.7 | 33 | 21 | 13 | 1765 |
| gina_prior[†] | 0.475 | 13.7±0.4 | 14.3±1.0 | **12.7±0.3** | 14.6±0.7 | 34 | 38 | 31 | 1147 |
| guillermo[†] | 0.463 | **30.7±0.2** | 34.1±0.3 | **30.7±0.2** | 32.0±0.4 | 908 | 505 | 116 | 31547 |
| hiv1* | 0.166 | 5.9±0.1 | 6.3±0.2 | **5.8±0.0** | 6.1±0.1 | 41 | 45 | 35 | 1012 |
| IMDB.drama [†] | 0.091 | 36.1±0.1 | 37.0±0.1 | 36.2±0.0 | 36.2±0.1 | 176 | 865 | 89 | 19436 |
| la1s.wc[†] | 2.029 | 9.3±0.0 | **8.3±0.3** | **8.2±0.1** | 9.0±0.1 | 17 | 47 | 12 | 2456 |
| MNIST[⊕] | 1.342 | 2.0±0.1 | 1.7±0.1 | **1.5±0.0** | 1.8±0.1 | 39 | 28 | 20 | 984 |
| OVA_Breast[†] | 2.660 | 3.2±0.1 | 3.8±0.4 | **3.0±0.1** | 3.4±0.2 | 140 | 23 | 37 | 4310 |
| rcv1[‡] | 0.242 | 5.7±0.0 | 5.3±0.0 | **5.1±0.0** | 6.3±0.0 | 2628 | 1271 | 543 | 701843 |
| riccardo[†] | 0.463 | 4.9±0.1 | 2.0±0.1 | **1.5±0.1** | 1.7±0.1 | 259 | 201 | 120 | 86575 |
| tr31.wc[†] | 3.305 | 2.7±0.3 | 2.7±0.4 | 2.7±0.2 | 2.9±0.3 | 6 | 14 | 6 | 987 |

Table 1: Comparison of the DRO and regularized ERM formulations over 13 publicly available machine learning (ML) datasets, from UCI* [17], OpenML[†] [6], MNIST[⊕] [14] and SKLearn[‡] [16].The first set of four columns provides a 95% confidence interval of the percentage misclassified over withheld test datasets. The second set of four columns provides the average CPU time taken.

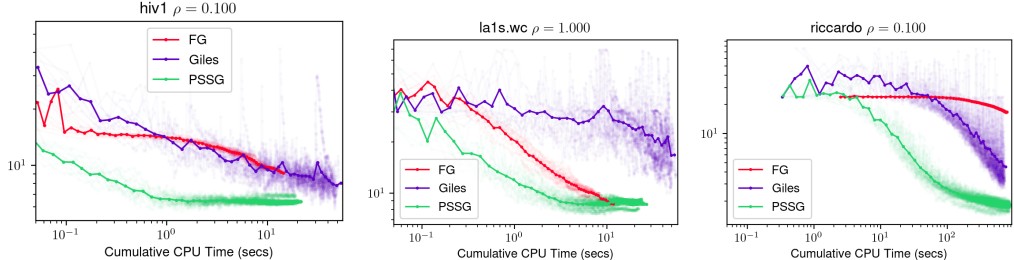

Figure 1: Comparison of PSSG (green), FG (red) and Giles (blue) on the fraction of misclassification in testing ($y$-axis) versus cumulative CPU time (log-scale $x$-axis) over the `hiv1`, `la1s.wc` and `riccardo` datasets, with $\rho$ set as noted.

This indicates a significant computational savings in using DRL because of the elimination of the expensive hyper-parameter tuning step.

The main parameter of the DRL formulation is the uncertainty radius $\rho$ for the set of measures $\mathcal{P}$, and following [2, 21] it is set to the same (base-10) order of magnitude as the value $\sqrt{d/N}$; Table 1 records this quantity for each dataset. We provide in supplement B detailed comparisons of the three DRL methods over multiple $\rho$ values (and other key parameter values) for several datasets. These results illustrate similar performance trends and insensitivities across the datasets relative to the $O(\sqrt{d/N})$ guideline for FG and PSSG (which are independent of its growth factor). The Giles method, however, exhibits complex interactions between $\rho$ and its minimum mini-batch size parameter that significantly impact both the solution quality and the computation time. Thus, fine tuning of these two key parameters of the Giles method is required to achieve the best quality solution and the smallest computational effort. In strong contrast, the interactions between $\rho$ and the key parameters of PSSG are relatively weak, with the growth factor only impacting the computational effort (and not the solution quality). Hence, it is sufficient to run PSSG with key parameters that are set based on our theoretical results without any fine tuning. This lack of fine tuning requirements is another important advantage of PSSG, in addition to its solution quality and computational advantages.

Among the DRL methods, PSSG takes the least time to solve the problem for all datasets (with one exception), often significantly so, while providing equal or better performance, including significant performance improvement over ERM in several cases. As expected, FG takes more time to solve the problem than PSSG for all datasets (comparable time for `tr31.wc`). For all but one dataset, Giles takes significantly more time than PSSG to solve the problem, by an order of magnitude in several cases. These longer computation times are due in large part to the added variability experienced by the Giles method arising from the Monte Carlo randomization. The effect of this added variability is

also evident in the consistently wider CIs of the misclassification errors for Giles, as well as in the wider range of CPU times with an isolated case terminating the fastest (`OVA_Breast`) and several cases terminating the slowest among DRL methods (e.g., `adult`, `fabert`, `IMDB.drama`, `la1s.wc`).

Figure 1 presents three representative comparisons of the empirical DRL runs over time, where the sample paths of all ten runs for each method are plotted along with their average shown in bold; corresponding plots for other datasets are provided in the supplement. The sample paths for PSSG (green) exhibit relatively low variability overall with reductions in variability as the sample size $M_t$ grows with the iterations $t$. In strong contrast, the sample paths for Giles (blue) exhibit much higher variability across the iterations, and are computationally more expensive, for the reasons explained above. Although FG (red) also exhibits low variability, it is clearly more expensive computationally.

**PSSG DRL.** Our Algorithm 1 starts with an initial sample size of 1, the fixed step length $\gamma = 0.5$ and, following Theorem 6, the *constant-growth factor* $\nu = 1.001$ is chosen close to 1. Although the parameter $\delta$ appears prominently in the inflation of $\rho$ to individual $\rho_t$ as defined in Theorem 3, $\delta$ is required only to be a small positive constant. Since the empirical results were found not sensitive to $\delta$, we set $\delta = 0.01$. All DRL algorithms solved the inner-maximization formulation to within $\epsilon$-accuracy where $\epsilon = 10^{-7}$. Each method monitors the performance of the current model in correctly classifying the withheld test validation set every $(1/20)$th of an epoch, and stops if the average of the last 20 misclassification fraction values does not improve more than $1\%$ when compared with the average of the previous 80 evaluations. Hence, an algorithm run stops if the misclassification error in the validation set over the current epoch is not improved by more than $1\%$ over the four past epochs.

Theorem 3 provides the fundamental relationship between the bias suffered by a fixed batch size SGD ($M_t = M$) method in solving (1) and the batch size $M$. The important impact of this bias is illustrated by empirical results presented in the supplement. Under a small growth factor of $\nu = 1.001$, the iterations of PSSG operate with $M_t = 1$ until $t \approx 400$ when it then rises to $M_t = 2$. Hence, PSSG initially enjoys the benefits of fast objective value reduction similar to SGD with fixed batch size, but then eventually eliminates the introduced bias and thus also enjoys fast convergence. PSSG therefore avoids the expense of hyper-parameter tuning the batch size of standard SGD for bias reduction.

We also consider the impact of two key algorithm parameters: sample size growth factor $\nu$ and step length $\gamma$. Following Theorem 6, these settings are broadly expected to impact the computation times and solution quality. Empirical results (in the supplement), contrasting the performance of PSSG keeping one of the parameters $\gamma$ and $\nu$ fixed while varying the other, show that the performance is insensitive to $\nu$ for values smaller than 1.001 and $\gamma$ larger than 0.5, indicating that an error floor $\bar{E}^*$ has been reached. Hence we use these parameter values for all experiments in Table 1.

**FG DRL.** The step lengths of the FG algorithm are determined by the LBFGS-B algorithm with a maximum of 0.5 for all experiments. This parameter can have a notable impact on the speed of convergence, and our results in the supplement show this to be the best choice over many datasets.

**Giles DRL.** The parameters of the Giles algorithm are a minimum mini-batch size of 5 and a stepsize sequence of $\gamma_t = 0.5 * (5000/(5000 + t))$ for all experiments. As noted above, the minimum mini-batch size has a significant impact on the performance of the Giles method. The value 5 was chosen after careful study over multiple datasets as described in the supplement.

**Regularized ERM.** The 10-*fold CV* procedure partitions the full training dataset into 10 equal parts and trains a *regularized* model over each dataset formed by holding out one of the 10 parts as the *validation* dataset. The ERM loss objective is regularized by a $\lambda \|\theta\|_2^2$ term, and a fixed-batch SGD (size 10) is used for each combination of partition and $\lambda$ values. Enumeration is employed to find the optimal $\lambda$ from a grid of 20 points in the range $[10^{-6}, 10^6]$, starting with $10^6$ and backtracking until the *average* performance over the 10 validation datasets does not improve for three $\lambda$ enumerations. Note that the computational benefits of PSSG would be even larger if all $\lambda$ values over all grid points were enumerated. Our results in the supplement show a significant amount of variability in the optimal $\lambda$ over all the datasets, with no evident pattern relating to dataset characteristics, which highlights the need for ERM computations over a wide range of $\lambda$ values for each dataset.

**Summary.** Our empirical results support our theoretical results and show that PSSG achieves the main objectives of reaching optimal solutions of the DRL formulation and improving model generalization more consistently and more quickly than other methods. In particular, PSSG produces equal or better quality models as FG, Giles and regularized ERM with significantly less computational effort, often by orders of magnitude, thus providing a strong alternative approach to generalization.

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

## Funding Transparency Statement

All work performed as part of this paper was conducted by the authors under the employment of our respective organizations; all computing was performed on internal resources of the IBM Research organization

