# Supplement: *Efficient Generalization with Distributionally Robust Learning*

Our main objectives in this paper are to devise an algorithmic approach and theoretical results that efficiently solve DRL formulations and that improve generalization more consistently and more quickly than alternative methods. This supplement contains additional theoretical and empirical results, together with corresponding technical details and discussions, in support of the main body of the paper. Specifically, Section A first covers supporting theoretical results and all the proofs of our theoretical results, including associated technical details. Then, Section B covers supporting empirical results from our numerous numerical experiments and all related technical details.

## A  Theoretical Results

Our presentation of additional theoretical results and proofs is organized as follows. Section A.1 presents the proofs of Proposition 1 and Proposition 5, whereas Section A.2 describes Algorithm 2 and presents the proof of Proposition 2. The proof of Theorem 3 is considered in Section A.3, while Section A.4 presents the proofs of Theorem 4 and Theorem 6. Each of these sections includes statements of the theoretical results from the main body of the paper in a self-contained manner, together with discussion of the theoretical results and associated technical details. Throughout this supplemental section, the following additional notation is used extensively. We write $a^T$ where $T$ denotes the transpose operator. For a sequence of positive-valued random variables (r.v.s) $\{A_n\}$: We write $A_n = o_p(1)$ if $A_n \xrightarrow{\text{P}} 0$ as $n \to \infty$; We write $A_n = \mathcal{O}_p(1)$ if $\{A_n\}$ is stochastically bounded, that is, for a given $\epsilon > 0$ there exists $c(\epsilon) \in (0, \infty)$ with $\mathbb{P}(A_n < c(\epsilon)) > 1 - \epsilon$ for sufficiently large $n$; If $\{B_n\}$ is another sequence of positive-valued r.v.s, we write $A_n = \mathcal{O}_p(B_n)$ if $A_n/B_n = \mathcal{O}_p(1)$.

### A.1  Uniqueness of $P^*(\theta)$, Form of $\nabla_\theta R(\theta)$, and Lipschitzness Conditions for $\nabla_\theta R(\theta)$

We start by establishing some basic properties of $\phi$-divergences.

**Lemma 7.** *Consider probability mass functions over a support of size $M_1$. Suppose*

$$\mathcal{P}(M_2) = \left\{ P = (p_m) \in \mathbb{R}^{M_1} \;\Big|\; \sum_m p_m = 1, \; p_m \geq 0, \; p_{m'} = 0, \; \forall m' = M_2 + 1, \ldots, M_1 \right\}$$

*to be the subset of all probability mass functions that place positive mass in only a subset of size $M_2 < M_1$ of the full support size $M_1$. Define $U_{M_2} := \left( \underbrace{\frac{1}{M_2}, \ldots, \frac{1}{M_2}}_{M_2}, \underbrace{0, \ldots, 0}_{M_1 - M_2} \right)$. Then, we have that $U_{M_2} = \arg\min_{P \in \mathcal{P}(M_2)} D_\phi(P, U_{M_1})$.*

*Proof.* For any $P \in \mathcal{P}(M_2)$, we conclude

$$D_\phi(P, U_{M_1}) = \sum_{m \leq M_2} \frac{1}{M_1} \phi(M_1 p_m) + \sum_{m > M_2} \frac{1}{M_1} \phi(0) = \frac{M_2}{M_1} \sum_{m \leq M_2} \frac{1/M_1}{M_2/M_1} \phi(M_1 p_m) + \sum_{m > M_2} \frac{1}{M_1} \phi(0)$$

$$\geq \frac{M_2}{M_1} \phi \left( \sum_{m \leq M_2} \frac{1/M_1}{M_2/M_1} M_1 p_m \right) + \sum_{m > M_2} \frac{1}{M_1} \phi(0) = D_\phi(U_{M_2}, U_{M_1}),$$

where in the last step Jensen's inequality is applied to the convex function $\phi$. $\qquad\square$

Lemma 7 shows that the $D_\phi$-distance between the uniform distribution $U_N$, which assigns equal mass to all $N$ support points, and any probability mass function (pmf) on an $M$-subset is minimized by the probability distribution $U_M$, and that the minimal distance is given by

$$D_\phi(U_M, U_N) = \frac{M}{N} \; \phi \left( \frac{N}{M} \right) + \frac{(N - M)}{N} \; \phi(0).$$

If $D_\phi(U_M, U_N) > \rho$, then the feasibility set admitted by the $D_\phi$-constraint in

$$\mathcal{P} = \left\{ P \mid D(P, P_b) \leq \rho, \int dP(\xi) = 1, P(\xi) \geq 0 \right\}$$

does not admit any pmf with mass only on $M$ support points. Note that $D_\phi(U_M, U_N)$ is also decreasing in $M$, because $D_\phi(U_M, U_N)$ is a convex mixture of $\phi(0)$ and $\phi(N/M)$. Since $\phi(s)$ is strictly increasing for $s > 1$, we have that $\phi(N/M_2) > \phi(N/M_1) > 1$ for $M_1 > M_2$. Hence, any convex combination of the form above will satisfy $D_\phi(U_{M_2}, U_N) \geq D_\phi(U_{M_1}, U_N)$, and thus, for a given $\rho$, there exists an $M'(\rho)$ such that the constraint in (3) only admits pmfs with mass on $M \geq M'(\rho)$ support points. As a consequence, the optimal solution of the problem (3) may lie on the intersection of up to $N - M'(\rho)$ hyperplane constraints of the form $p_i = 0, \ \forall i = M'(\rho) + 1, \ldots, N$. This may in turn lead to degenerate optimal solutions for some objective coefficients $l(\theta, \xi_i)$ of the linear program (3). We preclude this possibility by assuming that the parameter $\rho$ in (3) satisfies

$$\rho < \bar{\rho}(N, \phi) = \left(1 - \frac{1}{N}\right) \phi\left(\frac{N}{N-1}\right) + \frac{1}{N}\phi(0). \tag{5}$$

This assumption is included in Proposition 1.

Another cause for degeneracy in an optimization solution is if the objective function does not depend on the decision variables, in which case the entire feasible set is optimal. In the problem (3), this could happen if $l(\theta, \xi_n) = \ell, \forall n$, since the objective would be $\sum_n l(\theta, \xi_n) p_n = \ell \sum_n p_n = \ell$. Define the set $\Theta_\varnothing := \{\theta : l(\theta, \xi_{n_1}) = l(\theta, \xi_{n_2}), \ \forall n_1, n_2\}$ and, for a small $\varsigma > 0$, let the set $\Theta_{\varnothing,\varsigma} := \cup_{\theta_o \in \Theta_\varnothing} \{\theta : |\theta - \theta_o| < \varsigma\}$ define the $\varsigma$-neighborhood of $\Theta_\varnothing$. Then, we assume in Proposition 1 that $\Theta \subseteq \Theta_{\varnothing,\varsigma}^c$ for some small $\varsigma > 0$. In other words, for each $\theta \in \Theta$, there exists two sample points $\xi_{n_1}$ and $\xi_{n_2}$ such that $l(\theta, \xi_{n_1}) \neq l(\theta, \xi_{n_2})$.

**Proposition 1.** *Let the feasible region $\Theta$ be compact and assume $\Theta \subseteq \Theta_{\varnothing,\varsigma}^c$, for a small $\varsigma > 0$. Further suppose $\phi$ in the $D_\phi$-constraint has strictly convex level sets, and let $\rho < \bar{\rho}(N, \phi)$ with $\bar{\rho}(N, \phi)$ defined in (5). Then: (i) the optimal solution $P^*$ of $R(\theta) = \sup_{P \in \mathcal{P}}\{L_P(\theta)\}$ is unique, and the gradient is given by $\nabla_\theta R(\theta) = \sum_{n \in \mathcal{N}} p_n^*(\theta) \nabla_\theta l(\theta, \xi_n)$; and (ii) for all $\rho$, the gradient $\nabla_\theta R(\theta)$ is a sub-gradient of $R(\theta)$.*

**Proof of Proposition 1:** From the preceding discussion, our assumption that $\rho < \bar{\rho}(N, \phi)$ only admits feasible pmfs that assign non-zero mass to all support points. For strictly convex functions $\phi(\cdot)$, this then ensures that the problem (3) has a unique optimal solution $P^*$ when combined with the assumption that the objective coefficients $l(\theta, \xi_n) \neq \ell$ for all $n$ and some $\ell$. Moreover, $D_\phi(P^*, U_N) = \rho$.

For part (ii), first recall from the introduction that Ben-Tal et al. [1] describe the typical Lagrangian dual algorithm used for the convex-concave case, which is given by

$$R(\theta_{rob}^*) = \min_{\theta \in \Theta} \max_{p_n \geq 0} \min_{\alpha \geq 0, \lambda} \left\{ L_P(\theta) + \alpha(\rho - D_\phi(P, U_N)) + \lambda\left(1 - \sum_{n=1}^{N} p_n\right) \right\}$$

$$= \min_{\theta \in \Theta, \alpha \geq 0, \lambda} \alpha\rho + \lambda + \frac{\alpha}{N}\sum_{n=1}^{N} \phi^*\left(\frac{l(\theta, \xi_n) - \lambda}{\alpha}\right) \tag{6}$$

where the convex conjugate of $\phi$, namely $\phi^*(s) = \max_{u \geq 0}\{su - \phi(u)\}$, is known in closed form for various $\phi$-divergences such as modified $\chi^2$-divergence and $KL$-divergence. Now define $\mathcal{L}(\theta, \alpha, \lambda, P)$ as the Lagrangian in (6):

$$\mathcal{L}(\theta, \alpha, \lambda, P) = L_P(\theta) + \alpha(\rho - D_\phi(P, U_N)) + \lambda(1 - \sum_n p_n). \tag{7}$$

By (i), there exists a unique solution $P^*(\theta)$, and by Lagrangian duality principles [27, Lemma 2.1], a corresponding unique pair $(\alpha^*, \lambda^*)$ exists. We collectively call the primal and dual variables $v^*(\theta) = (\alpha^*(\theta), \lambda^*(\theta), P^*(\theta))$, and thus $R(\theta) = \mathcal{L}(\theta, v^*(\theta))$ where the first term $L_{P^*}(\theta) = \sum_n p_n^*(\theta) l(\theta, \xi_n)$. Differentiating using the chain rule, we have

$$\nabla_\theta R(\theta) = \nabla_\theta L_{P^*(\theta)}(\theta) + \nabla_\theta v^*(\theta) \, \nabla_v \mathcal{L}(\theta, v^*(\theta)) \tag{8}$$

$$= \sum_{n \in \mathcal{N}} p_n^*(\theta) \nabla_\theta l(\theta, \xi_n),$$

where the second term on the right in (8) vanishes because $\nabla_v \mathcal{L}(\theta, v^*(\theta)) = 0$ by the first order optimality conditions of $v^*$. The same result is obtained in a more general setting that allows for multiple solutions to the maximization problem [29, Theorem 7.21, p. 352]. $\square$

**Proposition 5.** *Assume the conditions in Proposition 1 hold. Further suppose that the Hessians $\nabla^2_\theta l(\theta, \xi_n)$ exist, $\forall \theta$ and each $\xi_n$, and are bounded in Frobenious norm $\|\nabla^2_\theta l(\theta, \xi_n)\|_F \leq L, \forall \theta, n$. Then, the robust loss also follows $\|\nabla^2_\theta R(\theta)\|_F \leq M$ for some positive $M < \infty$.*

**Proof of Proposition 5:** Further differentiating (8) using the chain rule and again applying the first order optimality conditions for $v^*$, we obtain

$$\nabla^2_\theta R(\theta) = \sum_{n \in \mathcal{N}} p_n^* \nabla^2_\theta l(\theta, \xi_n) + \nabla_\theta v^*(\theta) \, [\nabla^2_{\theta v} \mathcal{L}(\theta, v^*(\theta))]^T. \tag{9}$$

Since the $P^*$ are bounded, we have our desired result from the triangle inequality if we can show that the second term has bounded components.

Following [27, Lemma 2.2], the gradient $\nabla_\theta v^*(\theta)$ can be expressed as

$$\nabla_\theta v^*(\theta) = -\nabla^2_{\theta v} \mathcal{L}(\theta, v^*(\theta))[\nabla^2_{vv} \mathcal{L}(\theta, v^*(\theta))]^{-1}.$$

Let $\mu = (\alpha, \lambda)$. Further let $\phi'(s) = \mathrm{d}\phi(s)/\mathrm{d}s$ and $\phi''(s) = \mathrm{d}^2\phi(s)/\mathrm{d}s^2$ be the first and second derivative of $\phi(s)$ w.r.t. $s$, respectively. For (6), we then obtain the following components:

$$\nabla^2_{\mu\mu}\mathcal{L} = 0, \qquad\qquad \nabla^2_{pp}\mathcal{L} = -N\alpha \mathrm{Diag}(\phi''(NP)),$$

$$\nabla^2_{p\mu}\mathcal{L} = - \begin{bmatrix} \mathbf{e} \\ \phi'(NP) \end{bmatrix}, \qquad\qquad \nabla^2_{\theta v}\mathcal{L} = \begin{bmatrix} \nabla_\theta l_1(\theta, \xi_1) \\ \dots \\ \nabla_\theta l_N(\theta, \xi_N) \\ \mathbf{0} \\ \mathbf{0} \end{bmatrix},$$

where $\phi'(NP)$ and $\phi''(NP)$ represent the vectors of first and second derivatives of $\phi$ at the components of the vector $NP$, and $\mathbf{e}$ represents the vector of all ones.

We calculate the inverse of $\nabla^2_{vv}\mathcal{L}$ using the Schur complement of $\nabla^2_{pp}\mathcal{L}$ in $\nabla^2_{vv}\mathcal{L}$, along with the components $\nabla^2_{\mu\mu}\mathcal{L}$ and $\nabla^2_{p\mu}\mathcal{L}$. Note that, for a square matrix $M$ which can be partitioned into submatrices $A, B, C, D$ (as shown below) and where $A$ is invertible as $A^{-1}$, the Schur complement of $A$ is given by $M/A = D - CA^{-1}B$. We then have

$$M = \begin{bmatrix} A & B \\ C & D \end{bmatrix}, \qquad \text{and}$$

$$M^{-1} = \begin{bmatrix} A^{-1} + A^{-1}B(M/A)^{-1}CA^{-1} & -A^{-1}B(M/A)^{-1} \\ -(M/A)^{-1}CA^{-1} & (M/A)^{-1} \end{bmatrix}.$$

Applying this to $M = \nabla^2_{vv}\mathcal{L}$ with $A = \nabla^2_{pp}\mathcal{L}$, we obtain $D = \nabla^2_{\mu\mu}$ and $C = B^T = \nabla^2_{p\mu}\mathcal{L}$. Given the form of $\nabla^2_{\theta v}\mathcal{L}$, we need only compute the top-left element of $M^{-1} = [\nabla^2_{vv}\mathcal{L}]^{-1}$ in order to compute the second term in (9). The Schur complement is given by

$$M/A = \frac{1}{N\alpha} \begin{bmatrix} \mathbf{e} \\ \phi'(NP) \end{bmatrix} \mathrm{Diag}\left(\frac{1}{\phi''(NP)}\right) \begin{bmatrix} \mathbf{e}^T & \phi'(NP)^T \end{bmatrix}$$

$$= \frac{1}{N\alpha} \begin{bmatrix} \sum_n (1/\phi''(Np_n)) & \sum_n (\phi'(Np_n)/\phi''(Np_n)) \\ \sum_n (\phi'(Np_n)/\phi''(Np_n)) & \sum_n (\phi'(Np_n))^2/\phi''(Np_n) \end{bmatrix}.$$

Note that $M/A$ is a $2 \times 2$ matrix, and the inverse of a $2 \times 2$ matrix $\begin{bmatrix} a & b \\ c & d \end{bmatrix}$ is $\begin{bmatrix} d & -b \\ -c & a \end{bmatrix}(ad - bc)^{-1}$.

Since the $\phi$ are strictly convex, $\phi''(Np_n) \geq \delta > 0$ for some $\delta$ and any $p_n$. Then the term $(ad - bc)$ is the variance of a random variable taking values $\phi'(Np_n)$ with probability $1/\phi''(Np_n)$, and thus it is strictly positive, again because of the strict convexity of $\phi$. Finally, the terms $a, b, c, d$ are all finite because their denominators are strictly away from zero by $\delta$.

The term $A^{-1} + A^{-1}B(M/A)^{-1}CA^{-1}$ can similarly be found to have elements that are all finite (the optimal $\alpha^* < \infty$ as seen in the proof of Proposition 2 below), and hence the second term in (9) also has finite elements, rendering the desired result. $\square$

## A.2 Solving for $P^*(\theta)$ in $D_\phi$-Constrained Inner-Maximization

Let $\phi'(s) = \mathrm{d}\phi(s)/\mathrm{d}s$ be the derivative of $\phi(s)$ w.r.t. $s$, where $(\phi')^{-1}$ denotes its inverse. By assumption, $\phi$ is strictly convex, and thus $\phi'(s)$ is strictly increasing in $s$, which provides us with the existence of its inverse. The derivative $\phi'(s)$ plays a key role in the proof of Proposition 2 below. Refer to Figure 2 for an illustration of plots of $\phi$ as a function of $s$, $\phi'$ as a function of $s$, its inverse $(\phi')^{-1}(y)$ as a function of $y$, and finally $(\phi')^{-1}((z_i - \lambda)/\alpha)$ as a function of $\lambda$. The plots illustrate both cases where $\phi'(s) \to -\infty$ (e.g., KL-divergence $\phi(s) = s \log s - s + 1$) on the top row and $\lim_{s\to 0+} \phi'(s) > -\infty$ (e.g., modified $\chi^2$-divergence $\phi(s) = (s - 1)^2$) on the bottom row.

Recall that the inner maximization problem in (3) for a *random* subsample of size $M_t$, with a target $D_\phi$-divergence of $\rho_t$, uses a decision variable $P = (p_m)$ of dimension $M_t$. Writing the Lagrangian objective of (3) as

$$\mathcal{L}(\alpha, \lambda, P) = \sum_{m \in \mathcal{M}_t} z_m p_m + \lambda\left(1 - \sum_{m \in \mathcal{M}_t} p_m\right) + \frac{\alpha}{M_t}\left(M_t \rho_t - \sum_{m \in \mathcal{M}_t} \Phi(M_t p_m)\right), \quad (10)$$

we then have the optimal objective value $\hat{R}_t^*(\theta) = \min_{\alpha \geq 0, \lambda} \max_{\hat{p}_m \geq 0} \mathcal{L}(\alpha, \lambda, P)$; refer to [18]. The optimal primal and dual variables can be obtained for various $\phi$ functions by a general procedure to solve the Lagrangian formulations (10) for a given iteration $t$. In particular, we use the following general procedure, summarized in the main body of the paper as Algorithm 2 and expressed in expanded form below, noting that this basic approach has been pursued, either explicitly or in a similar spirit, in previous work such as Ben-Tal et al. [1], Namkoong and Duchi [20, 21], Ghosh and Lam [9].

**Algorithm 2.** *(Restated in expanded form.)*

1. *Case:* $\alpha^* = 0$ along with constraint $D_\phi(P_t^*, P_b) \leq \rho_t$.
   (a) Let $\mathcal{M}'_t = \{m \in \mathcal{M}_t : z_m = \max_{u \in \mathcal{M}_t} z_u\}$ and $M'_t = |\mathcal{M}'_t|$. Set $\alpha^* = 0$ in (10), and then an optimal solution is $P^*$ where $p_m^* = \frac{1}{M'_t}$, $\forall m \in \mathcal{M}'_t$, and $p_m^* = 0$, $\forall m \notin \mathcal{M}'_t$; see Proposition 2.
   (b) If $D_\phi(P^*, P_b) \leq \rho_M$, then **stop** and return $P^*$.

2. *Case:* constraint $D_\phi(P_t^*, P_b) = \rho_t$ with $\alpha^* \geq 0$.
   (a) Keeping $\lambda, \alpha$ fixed, solve for the optimal $P_t^*$ (as a function of $\lambda, \alpha$) that maximizes $\mathcal{L}(\alpha, \lambda, P)$, applying the constraint $p_m \geq 0$.
   (b) Keeping $\alpha$ fixed, solve for the optimal $\lambda^*$ using the first order optimality condition on $\mathcal{L}(\alpha, \lambda, P_t^*)$. Note that this is equivalent to satisfying the equation $\sum_{m \in \mathcal{M}_t} p_m^* = 1$. Proposition 2 shows that this step is at worst a bisection search in one dimension, but in some cases (e.g., KL-divergence) a solution $\lambda^*$ is available in closed form. The proof of Proposition 2 also provides finite bounds $[\underline{\lambda}, \bar{\lambda}]$ on the range over which we need to search for $\lambda^*$.
   (c) Apply the first order optimality condition to the one-dimensional function $\mathcal{L}(\alpha, \lambda^*(\alpha), P_t^*)$ to obtain the optimal $\alpha^* \geq 0$. This is equivalent to requiring that $\alpha^*$ satisfies the equation $\sum_{m \in \mathcal{M}} \phi(p_{t,m}^*) = \rho_t$. Proposition 2 shows that this is at worst a one-dimensional bisection search which embeds the previous step in each function call of the search.
   (d) Define the index set $\mathcal{N} := \{m \in \mathcal{M}_t \mid \lambda^* \leq z_m - \alpha^* \phi'(0)\}$, with $\mathcal{N} = \emptyset$ if $\phi'(s) \to -\infty$ as $s \to 0+$. Set

$$p_{t,m}^* = \begin{cases} \frac{1}{M_t}(\phi')^{-1}\left(\frac{z_m - \lambda^*}{\alpha^*}\right), & m \in \mathcal{N} \\ 0 & m \notin \mathcal{N} \end{cases}. \quad (11)$$

(Note that the proof of Proposition 2 explains the expression in (11).) **Return $P_t^*$.**

**Proposition 2.** *For any $\phi$-divergence, Algorithm 2 finds a feasible primal-dual solution $(\tilde{P}_t^*, \tilde{\alpha}^*, \tilde{\lambda}^*)$ to (3) with an objective function value $\tilde{R}_t^*$ such that $|\hat{R}_t^*(\theta) - \tilde{R}_t^*| < \epsilon$ with a worst-case computational effort bounded by $O(M_t \log M_t + (\log \frac{1}{\epsilon})^2)$, where $\epsilon$ is a small precision parameter.*

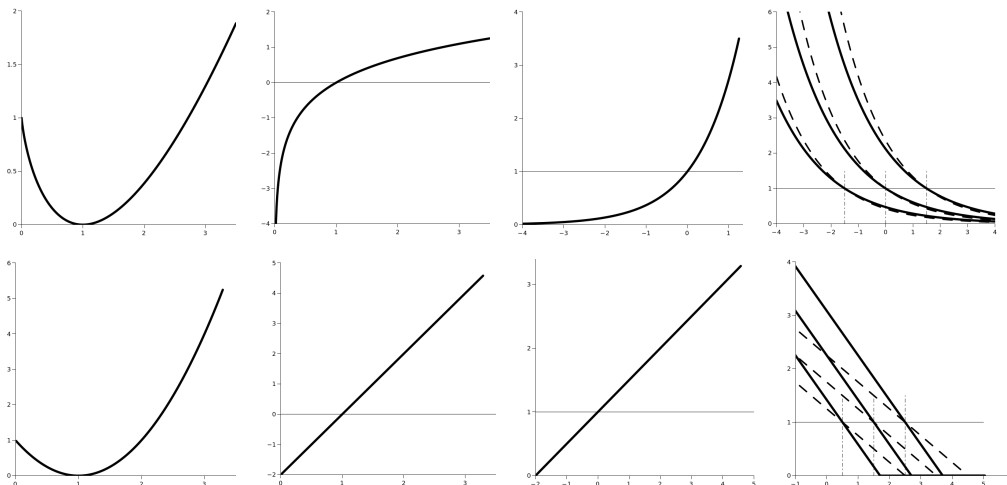

Figure 2: The plots on the top row are for the Kuhlback-Leibler divergence, and those on the bottom are for the modified $\chi^2$-divergence. Specifically, the top row from left-to-right considers: $\phi(s) = s \log s - s + 1$, $s \geq 0$; $\phi'(s) = \log s$, $s > 0$; $(\phi')^{-1}(y) = e^y$, $y \in \mathbb{R}$; $p^*_{z,\alpha}(\lambda) = (\phi')^{-1}((z-\lambda)/\alpha)$, $\lambda \in \mathbb{R}$. The bottom row from left-to-right considers: $\phi(s) = (s-1)^2$, $s \geq 0$; $\phi'(s) = 2(s-1)$, $s \geq 0$; $(\phi')^{-1}(y) = 1 + y/2$, $y \in \mathbb{R}$; $p^*_{z,\alpha}(\lambda) = (\phi')^{-1}((z-\lambda)/\alpha)$, $\lambda \in \mathbb{R}$. Note: On the top, $\phi'(s) \to -\infty$, so the inverse is always positive; for modified $\chi^2$-divergence on the bottom, it is positive only when $y \geq -2$. The last column plots $p^*_m$ from (11) for three $z_m$ and two $\alpha$.

**Proof of Proposition 2:** We eliminate the subscript $t$ in this proof for clarity of exposition; thus, the support is indexed over $m = 1, \ldots, M$.

To start, order all the $z_m$ into the increasing sequence $z_{(1)} \leq z_{(2)} \leq \ldots \leq z_{(M)}$, where the subscript notation $(i)$ denotes the index of the $i$th smallest $z_m$ value. The tightest bound on the cost of sorting the vector $(z_m)$ in increasing order is $O(M \log M)$.

We first handle the case when the $\phi$-divergence constraint is not tight and $\alpha^* = 0$. Substituting this in (10) shows that any optimal solution $\hat{P}^*$ places mass only within the set $\mathcal{M}'$ as defined in Step 1(a). Consider any such $P^*$, and let $U_M$ be as defined in the statement of Lemma 7. Then the lemma provides that, among all optimal solutions, $U_{M'}$ obtains the smallest divergence, and thus it is the best optimal candidate to meet the divergence constraint with slack. This is why the solution procedure stops in Step 1(b). The computational complexity of Step 1 is mainly due to determining the set $M'$, which can be part of the sorting operation that determines the sequence $z_{(m)}$ above.

Note that this case is basically precluded by the assumptions of Proposition 1. With the assumption that $\rho < \bar{\rho}(N, \phi)$, we have that the feasible region only allows for pmfs that have non-zero mass over the full support $M$. This in turn implies that for this case to hold, we need $l(\theta, \xi_n) = \ell$ for all $n$, which is also ruled out by the assumptions of Proposition 1.

For the case when the $\phi$-divergence constraint is tight, we proceed according to the corresponding three steps in Algorithm 2.

**Step 2(a).** Identify the set of indices $\mathcal{N}$ as defined above. Setting to zero the gradient of $\mathcal{L}(\alpha, \lambda, P)$ w.r.t. $P$, we obtain the expression in (11) for $P^*(\lambda, \alpha)$. For the case where $\phi'(s) \to -\infty$, all $p^*_m > 0$ since $\mathcal{N} = \emptyset$. In the case where $\phi'(s) \to K$ for some constant $K > -\infty$, the probability values need to observe the check on $(\lambda, \alpha)$ as given in (11) in order to satisfy the non-negativity constraint on $P^*$. The set $\mathcal{N} = \mathcal{N}(\lambda, \alpha)$ can be equivalently represented as $\mathcal{N} = \{(m) \mid (m) \geq (m_g)\}$ where $(m_g) = \inf_m \{(m) \mid z_{(m)} \geq \lambda + \alpha \phi'(0)\}$, i.e., the smallest ordered value $z_{(m)}$ satisfying the defining condition of $\mathcal{N}$. From (11), we observe that $p^*_{(m)}$ are strictly increasing in $m$ for any fixed $(\lambda, \alpha)$. Hence, the optimal probability allocation to support points increases in accordance with their $z_m$ values.

**Step 2(b).** Define

$$h_\alpha(\lambda) := \sum \hat{p}_m^* = \frac{1}{M} \sum_{(m) \geq (m_g)} (\phi')^{-1} \left( \frac{z_{(m)} - \lambda}{\alpha} \right).$$

We seek the $\lambda$ that attains $h_\alpha(\lambda) = 1$, i.e., $\sum \hat{p}_m^* = 1$. The above expression for $h_\alpha(\lambda)$ is a decreasing function of $\lambda$ for fixed $\alpha$, given the strict convexity of $\phi$. When $\phi'(0) > -\infty$, at $\bar{\lambda} = \bar{\lambda}(\alpha) = z_{(M)} - \alpha\phi'(0)$, the summation $h_\alpha(\bar{\lambda}) = 0$. For the $\phi'(0) \to -\infty$ case, it is sufficient to consider a point $\bar{\lambda}(\alpha) = z_{(M)} - \alpha(z_{(1)} - z_{(M)})$ to obtain that $h_\alpha(\bar{\lambda}) < 1$. On the other hand, at $\underline{\lambda} = z_{(1)}$, we have (note the properties of $\phi'$ in Figure 2) that

$$h_\alpha(z_{(1)}) = \frac{1}{M} \left( \underbrace{(\phi')^{-1}\left( \frac{z_{(1)} - z_{(1)}}{\alpha} \right)}_{=1} + \sum_{m>1} \underbrace{(\phi')^{-1}\left( \frac{z_{(m)} - z_{(1)}}{\alpha} \right)}_{>1} \right) \quad > \quad 1.$$

Hence, we only need to perform a bisection search on $[\underline{\lambda}, \bar{\lambda}]$, which can be performed with a computational effort of at most $O(\log 1/\epsilon)$ to get to within $\epsilon$ precision. The size $|\bar{\lambda} - \underline{\lambda}|$ of the search interval appears in the order constant, but does not change with $M$.

**Step 2(c).** To obtain $\alpha^*$, substitute the $P^*$ from (11) into the divergence constraint satisfied as an equality, which yields

$$D_\phi(\alpha) = \frac{1}{M} \sum_m \phi(Mp_m^*(\alpha)) = \rho.$$

We show below that the function $D_\phi(\alpha)$ is decreasing in $\alpha$, and hence a bisection search leads us to the optimal $\alpha^*$. The total computational effort then in estimating the correct $\alpha^*$ involves $\log(\frac{1}{\epsilon})$ function calls to the bisection search to find the $\lambda(\alpha)$ for the current iterate of $\alpha$. As noted before, each of these function calls takes at most $\log(\frac{1}{\epsilon})$. This, in addition to the time taken to sort the $z_m$ once in each run of the complete algorithm, provides us with the bound on the computational effort.

Consider any pair of $\alpha_1 < \alpha_2$. Let $\lambda^*(\alpha_i) = \lambda_i^*$ be the optimal value that attains $h_{\alpha_i}(\lambda_i^*) = 1$, $i = 1, 2$. The summation $h_\alpha(\lambda)$ is decreasing in $\alpha$ for a fixed $\lambda$, and thus $h_{\alpha_2}(\lambda_1^*) < 1$. Since $h_\alpha(\lambda)$ is a decreasing function of $\lambda$ for a fixed $\alpha$, we have that $\lambda_2^* < \lambda_1^*$.

Now, let $\underline{\lambda}_2$ be the value that satisfies the equality $(z_{(M)} - \underline{\lambda}_2)/\alpha_2 = (z_{(M)} - \lambda_1^*)/\alpha_1$, so that at $\underline{\lambda}_2$ the support point corresponding to the largest value $z_{(M)}$ has the same probability allocation (11) under $\alpha_2$ as under $\alpha_1$. For any $m < M$, we then have

$$\frac{z_{(m)} - \underline{\lambda}_2}{\alpha_2} = \frac{z_{(m)} - z_{(M)}}{\alpha_2} + \frac{z_{(M)} - \underline{\lambda}_2}{\alpha_2} = \underbrace{(z_{(M)} - z_{(m)})\left( \frac{1}{\alpha_1} - \frac{1}{\alpha_2} \right)}_{>0} + \frac{z_{(m)} - \lambda_1^*}{\alpha_1}.$$

Therefore, at $\underline{\lambda}_2$, we have that $h_{\alpha_2}(\underline{\lambda}_2) > 1$ and that $\lambda_2^* \in [\underline{\lambda}_2, \lambda_1^*]$. As a consequence, the mass allocated to $p_{(M)}^*$ also decreases. Define $\Delta := p_{(M)}^*(\alpha_1) - p_{(M)}^*(\alpha_2)$, and then from the preceding discussion $\Delta > 0$. Moreover, we have $p_{(M)}^*(\alpha_2) > 1/M$, else given the strict ordering of (11) over $\{(m)\}$, the total probability assigned over all support points will sum up to less than 1. Hence, $M p_{(M)}^*(\alpha_2) > 1$, to the right of the minima at $s = 1$ of $\phi(s)$.

The change in $D_\phi$ is bounded as follows:

$$M(D_\phi(\alpha_2) - D_\phi(\alpha_1)) = \left( \phi(p_{(M)}^*(\alpha_2)) - \phi(p_{(M)}^*(\alpha_1)) \right) + \sum_{m<M} \left( \phi(p_{(m)}^*(\alpha_2)) - \phi(p_{(m)}^*(\alpha_1)) \right)$$

$$= \phi'(\xi_M)(-\Delta) + \sum_{m<M} \left( \phi(p_{(m)}^*(\alpha_2)) - \phi(p_{(m)}^*(\alpha_1)) \right)$$

$$\leq -\phi'(\xi_M)\Delta + \left( \phi(p_{(M)}^*(\alpha_2)) - \phi(p_{(M)}^*(\alpha_2) - \Delta) \right)$$

$$= -\phi'(\xi_M)\Delta + \phi'(\underline{\xi}_M)\Delta \quad < \quad 0.$$

Here, the second equality applies the mean value theorem for some $\xi_M \in [p_{(M)}^*(\alpha_2), p_{(M)}^*(\alpha_1)]$. The first inequality is due to the highest increase in $\phi$ occurring if all the reduction in probability $\Delta$ is

picked up by the next support point $p^*_{(M-1)}$. Since the mass allocations are ordered, the best case is that $p^*_{(M-1)}(\alpha_1) = p^*_{(M)}(\alpha_2) - \Delta$ increases to $p^*_{(M-1)}(\alpha_2) = p^*_{(M)}(\alpha_2)$. The third equality again applies the mean value theorem for some $\underline{\xi}_M \in [p^*_{(M)}(\alpha_2) - \Delta, p^*_{(M)}(\alpha_2)]$, and the last inequality is due to the strict convexity of $\phi$.

Therefore, $D_\phi(\alpha)$ is a decreasing function of $\alpha$. To obtain a finite range $[\underline{\alpha}, \bar{\alpha}]$ that contains $\alpha^*$, first consider the $\phi'(s) > -\infty$ case. There exists an $\alpha_\tau$ such that $\lambda^*_\tau = \lambda^*(\alpha_\tau) = z_{(M)} - \tau$, for a small $\tau > 0$, and the corresponding $P^*_\tau$ places mass 1 on $z_{(M)}$ and zero elsewhere; this leads to $D_\phi(\alpha_\tau) = D_\phi(U_1, U_M)$, which is the highest distance possible as per the discussion following Lemma 7. On the other hand, as $\alpha \to \infty$, each $p^*_m \to 1/M$ as observed from (11) and Figure 2, and thus $D_\phi(\alpha) \to 0$ as $\alpha \to \infty$. Hence, there exists a range $[\underline{\alpha}, \bar{\alpha}]$ that contains $\alpha^*$.

The upper limit $\bar{\alpha}$ also applies when $\phi'(s) \to -\infty$, so only the lower limit $\underline{\alpha}$ needs be modified for this case. There exists a small $\delta_\tau$ such that $P^*_\tau$ places mass $1 - M\delta_\tau$ on $z_{(M)}$ and at most $\delta_\tau$ mass on $z_{(m)}$, $m < M$. As $\tau \to 0$, then $\delta_\tau \to 0$ and $D_\phi(\alpha_\tau) \to D_\phi(U_1, U_{M_1})$.

Note that when the conditions for Case 1 and Case 2 both hold, then the steps of Case 1 and Case 2 will both result in the same final outcome. $\square$

### A.3  Small-Sample Approximation of $\nabla_\theta R(\theta)$

We solve the DRL formulation by assembling the mini-batch through sampling without replacement from the training dataset [11], in strong contrast to the sampling with replacement employed in [20, 21, 15], and we analyze a general progressively increasing subsampling-based approach in the context of DRL and bias reduction. Indeed, the setup of the robust loss function enables us to extensively exploit the mathematical properties of statistically sampling a finite set without replacement. We therefore provide a brief summary here, together with a brief summary of the alternative sampling with replacement.

**Without-replacement Sampling:** Let $\{x_1, \ldots, x_N\}$ be a set of $N$ one-dimensional values with mean $\mu = \frac{1}{N} \sum_n x_n$ and variance $\sigma^2 = \frac{1}{N-1} \sum_n (x_n - \mu)^2$. Suppose we sample $M < N$ of these points uniformly without replacement to construct the set $\mathcal{M} = \{X_1, \ldots, X_M\}$. The probability that any particular set of $M$ subsamples was chosen is given by $\left(\frac{(N-M)!}{N!}\right)$. Denote by $\mathbb{E}_\mathcal{M}$ the expectation under this probability measure, and let $\bar{X} = \frac{1}{M} \sum_{m=1}^M X_m$ and $\bar{S}^2 = \frac{1}{M-1} \sum_{m=1}^M (X_m - \bar{X})^2$ represent the sample mean and sample variance, respectively. We then know that the expectation of the sample mean $\mathbb{E}_\mathcal{M}[\bar{X}] = \mu$ and of the sample variance $\mathbb{E}_\mathcal{M}[\bar{S}^2] = \sigma^2$ are both unbiased; refer to [35]. On the other hand, the variance of the sample mean

$$\mathbb{E}_\mathcal{M}[(\bar{X} - \mu)^2] = \left(\frac{1}{M} - \frac{1}{N}\right) \sigma^2$$

reduces to zero as $M \to N$. The term on the right above is fundamental to our results.

**With-replacement Sampling:** Suppose we have drawn $M < N$ samples from a training set of size $N$ uniformly, that is with replacement, to constitute the subsample $\mathcal{M}$. Let $I_n = \mathbb{I}\{\xi_n \in \mathcal{M}\}$, $n \in [N]$, take the value 1 if $\mathcal{M}$ contains $\xi_n$ at least once, and zero otherwise. Define $Y := \sum_{n=1}^N I_n$, which represents the number of unique support points in $\mathcal{M}$. We then compute its expected value as

$$\mathbb{E}Y = \sum_{n=1}^N \mathbb{E}I_n = \sum_{n=1}^N \mathbb{P}(\xi_n \in \mathcal{M}) = \sum_{n=1}^N 1 - \mathbb{P}(\xi_n \notin \mathcal{M}) = \sum_{n=1}^N 1 - \left(\frac{N-1}{N}\right)^M$$

$$= N\left(1 - \left(\frac{N-1}{N}\right)^M\right) = N\left(1 - \left(\left(1 - \frac{1}{N}\right)^N\right)^{\frac{M}{N}}\right) \approx N\left(1 - e^{-\frac{M}{N}}\right).$$

This slow exponential decay in the number of support points that can be sampled at least once in a subset of size $M$ shows that approximations of robust loss and its gradient constructed from a set $\mathcal{M}_t$ sampled with-replacement can suffer from slow convergence properties. Indeed, [15] assert

via constructing an example of a full dataset and associated loss functions $l$ that the variance of the estimate $\nabla \hat{R}$ of the robust loss gradient can be bounded away from zero for any $M$.

With this overview, we now begin our proof of Theorem 3 by addressing the feasibility of the restriction $\tilde{P}^*$ of the (unique, by assumption) optimal solution $P^*$ of the full-data problem onto the (randomly sampled) subset $\mathcal{M}_t$, where

$$\tilde{p}_m^* = \frac{p_m^*}{\sum_{j \in \mathcal{M}_t} p_j^*}, \quad \forall m \in \mathcal{M}_t.$$

Denote by $\mathcal{P}_t$ the feasibility set of (3) for the sampled $\mathcal{M}_t$, and recall from Section 2.1 the notational simplification that $\mathbb{E}_t = \mathbb{E}_{\mathcal{M}_t}$ and $\mathbb{P}_t = \mathbb{P}_{\mathcal{M}_t}$.

**Lemma 8.** *Suppose the $\phi$-divergence function has strictly convex level sets and $\rho < \bar{\rho}(N, \phi)$, where $\bar{\rho}(N, \phi)$ is defined in (5). Let the $D_\phi$-constraint target $\rho_t$ of the restricted problem (3) be set as stated in Theorem 3. Then, for $M \geq M_t'$, we have*

$$\mathbb{P}_t(\tilde{P}^* \in \mathcal{P}_t) \geq \max\{0, 1 - \tau_1\} \cdot \max\{0, 1 - \tau_2\}, \tag{12}$$

*where*

$$\tau_1 := \eta_t^{2\delta/(1-\delta)} \sigma^2(P^*), \qquad\qquad \tau_2 := \eta_t^{2\delta/(1-\delta)} \frac{\sigma^2(\phi)}{c_3^2},$$

$$\sigma^2(P^*) := \frac{1}{N-1} \sum_{n=1}^{N} (Np_n^* - 1)^2, \qquad\qquad \sigma^2(\phi) := \frac{1}{N-1} \sum_{n=1}^{N} (\phi(Np_n^*) - \rho)^2,$$

$$M' := \min \left\{ M \mid \eta_t \leq \max \left\{ 1 - \frac{1}{k_0}, \; \zeta_0, \; \frac{1}{k_0 \kappa_1} \right\} \right\}, \qquad c_3 := \frac{c - k_0(\kappa_1 \rho + \kappa_2)}{2}, \tag{13}$$

*and the constants $c$ and $k_0$ are chosen to yield $c_3 > 0$.*

Note here that $\sigma^2(\phi)$ calculates the variance in the vector $\phi(Np_n^*)$ for *any* $\phi$, though the formula makes it resemble the modified $\chi^2$-divergence $\phi(s) = (s-1)^2$. Recall that the constants $\zeta_0$, $\kappa_1$ and $\kappa_2$ arise from the local continuity of Assumption 1, and that $\eta_t = c(\frac{1}{M} - \frac{1}{N})^{(1-\delta)/2}$ from the statement of Theorem 3. Then $M'$ is defined in (13) as the *smallest* support size $M$ that leads to a prescribed positive value for the difference $(\frac{1}{M} - \frac{1}{N})$. Hence, as $N \to \infty$, $M' = o(N)$ in order to match this positive difference.

**Proof of Lemma 8:** In the notation of sampling without-replacement introduced above, define a set of scalar values $x_n(P^*) = Np_n^*$, $\forall n = 1, \ldots, N$. We use $\mu(P^*) = \frac{1}{N} \sum_n Np_n^* = 1$ and use the variance $\sigma^2(P^*)$ as in the statement of the lemma. By Chebychev's inequality, the sample-average $\bar{X}$ of an $M$-subsample chosen uniformly without replacement from this set satisfies

$$\mathbb{P}_t\left(\left|\bar{X}(P^*) - 1\right| > \eta_t\right) \leq \frac{1}{\eta_t^2} \mathbb{E}_t[(\bar{X}(P^*) - \mu)^2] \leq \left(\frac{1}{M_t} - \frac{1}{N}\right)^\delta \sigma^2(P^*).$$

Hence, $|\bar{X}(P^*) - 1| \leq \eta_t$ with probability at least $(1 - \tau_1)$. One implication of this is that $\sum_{\mathcal{M}_t} p_j^* \geq (M_t/N)(1 - \eta_t)$ with probability at least $(1 - \tau_1)$, and thus the restriction $\tilde{P}^*$ of $P^*$ to $\mathcal{M}_t$ is a pmf (i.e., its denominator is greater than zero) with probability converging to one as $M \to N$.

We now check if the restriction $\tilde{P}^*$ is feasible for the problem (3), namely whether $\mathbb{P}_t(D_\phi(\tilde{P}^*, U_{M_t}) > \rho_t)$ is small. Note that $\eta_t \searrow 0$ as $M_t \to N$; and rearrange $|\bar{X} - 1| \leq \eta_t$ to yield $\bar{X}^{-1} = (\frac{1}{M_t} \sum_{j \in \mathcal{M}_t} Np_j^*)^{-1} \leq 1 + k_0 \eta_t$ for all $M_t \geq M_t'$. Then for all $M_t \geq M_t'$, we obtain on the event $\mathcal{G} := \{|\bar{X}(P^*) - 1| \leq \eta_t\}$ that

$$D_\phi(\tilde{P}^*, U_{M_t}) = \frac{1}{M_t} \sum_m \phi\left(M_t \frac{p_m^*}{\sum_j p_j^*}\right) = \frac{1}{M_t} \sum_m \phi\left(Np_m^* \frac{1}{\frac{1}{M_t} \sum_{j \in \mathcal{M}_t} Np_j^*}\right)$$

$$\leq \frac{1}{M_t} \left(\sum_m \phi(Np_m^*)(1 + \kappa_1 k_0 \eta_t)\right) + \kappa_2 k_0 \eta_t, \tag{14}$$

where the last inequality follows from Assumption 1, applied in the form $\phi(s(1 + \zeta)) \leq \phi(s)(1 + \kappa_1 \zeta) + \kappa_2 \zeta$.

Let $\{x_n(\phi) = \phi(Np_n^*)\}_{n=1}^N$ be a vector of values associated with the $N$ support points. Then, for the (random) set of indices $\mathcal{M}_t$ chosen uniformly without replacement, $\mathbb{E}_t \bar{X}(\phi) = \mathbb{E}_t[\frac{1}{M_t} \sum_{m \in \mathcal{M}_t} \phi(Np_m^*)] = \mu(\phi) = \frac{1}{N} \sum_n \phi(Np_n^*) = D_\phi(P^*, U_N) = \rho$, again exploiting the fact that the $D_\phi$-constraint is tight at the optimal solution $P^*$ for the full-support optimization problem. Note that the corresponding population variance is given by $\sigma^2(\phi)$ as defined in the statement of the lemma.

Now define the desired event as $\mathcal{D} := \{D_\phi(\tilde{P}^*, U_{M_t}) \leq (\rho + c\eta_t)\}$. Then, inequality (14) renders the following bound on its complement

$$\mathbb{P}_t \left( \mathcal{D}^c | \mathcal{G} \right) = \mathbb{P}_t \left( D_\phi(\tilde{P}^*, U_{M_t}) > (\rho + c\eta_t) \mid \mathcal{G} \right)$$

$$\leq \mathbb{P}_t \left( \frac{1 + \kappa_1 k_0 \eta_t}{M_t} \sum_{m \in \mathcal{M}_t} \phi(Np_m^*) + \kappa_2 k_0 \eta_t > (\rho + c\eta_t) \right)$$

$$\leq \mathbb{P}_t \left( \left| \frac{1}{M_t} \sum_{m \in \mathcal{M}_t} \phi(Np_m^*) - \rho \right| > \frac{(c - \kappa_2 k_0 - \rho \kappa_1 k_0)}{1 + \kappa_1 k_0 \eta_t} \eta_t \right)$$

$$\leq \mathbb{P}_t \left( \left| \frac{1}{M_t} \sum_{m \in \mathcal{M}_t} \phi(Np_m^*) - \rho \right| > c_3 \eta_t \right) \quad \leq \quad \eta_t^{2\delta/(1-\delta)} \frac{\sigma^2(\phi)}{c_3^2},$$

where $c_3 = (c - \kappa_1 k_0 \rho - \kappa_2 k_0)/2 > 0$ by the assumptions on $c$ and the minimum $M_t'$, and where Chebychev's inequality is used in the last step. This yields the desired high probability guarantee from the elementary probability identity that $\mathbb{P}_t(\mathcal{D}) \geq \mathbb{P}_t(\mathcal{D}|\mathcal{G})\mathbb{P}_t(\mathcal{G})$. $\square$

Lemma 8 shows that the specific choice of $\rho_t$ leads to the restriction $\tilde{P}_t^*$ of the unique optimal $P^*$ to be feasible for (3) with probability converging to 1 as $M \to N$. We next establish that the bias in the estimation of the optimal objective is $O_p(\eta_t)$.

**Lemma 9.** *Under the assumptions of Lemma 8, there exists a $c_1 > 0$ such that*

$$\mathbb{P}_t \left( \eta_t^{-1} |\hat{R}_t(\theta) - R(\theta)| \leq c_1 \right) \geq \bar{\tau}_t$$

*where*

$$\bar{\tau}_t := \max\{0, 1 - \tau_3\} \cdot \hat{\tau}, \qquad\qquad \tau_3 := \eta_t^{2\delta/(1-\delta)} \frac{\sigma^2(R)}{c_1^2/4},$$

$$\hat{\tau} := \max\{0, 1 - \tau_1\} \cdot \max\{0, 1 - \tau_2\}, \quad \sigma^2(R) := \frac{1}{N-1} \sum_{n=1}^N ((z_n Np_n^*) - \mu(R))^2,$$

$$\mu(R) := \frac{1}{N} \sum_n z_n Np_n^*.$$

Note the definition of $\bar{\tau}_t$ in Lemma 9, which appears in the statement of Theorem 3. Further note that $C$ in Theorem 3 corresponds to $c_1$ in Lemma 9. The term $\hat{\tau}$ is the probability on the right hand side of (12).

**Proof of Lemma 9:** We split the optimality gap as $|\hat{R}_t(\theta) - R(\theta)| \leq |z^T P_t^* - z^T \tilde{P}^*| + |z^T \tilde{P}_t^* - z^T P^*|$, where $P^*$ is the (unique) solution to the full-support problem (1), $\tilde{P}_t^*$ its restriction to a sampled subset $\mathcal{M}$ of size $M$, and $P_t^*$ the (unique) solution to the optimization problem (3) on the subsampled support $\mathcal{M}_t$. We then rewrite the required probability and analyze each summand as

$$\mathbb{P}_t \left( |\hat{R}_t(\theta) - R(\theta)| > c_1 \eta_t \right) \leq \mathbb{P}_t \left( |z^T P_t^* - z^T \tilde{P}^*| > \frac{c_1}{2} \eta_t \right) + \mathbb{P}_t \left( |z^T \tilde{P}_t^* - z^T P^*| > \frac{c_1}{2} \eta_t \right). \tag{15}$$

For the first term, note that $z^T \tilde{P}^* \leq z^T P_t^*$ since $\tilde{P}^*$ is a feasible solution to the restricted problem (3) with probability $\hat{\tau}$ as shown in Lemma 8, while $P_t^*$ is its optimal solution. On the other hand, $P_t^*$

satisfies the $D_\phi$-divergence constraint at $\rho + \eta_t$, using the same arguments as for $P^*$ at $\rho$. Following along the same lines that lead up to the expression (8) of $\nabla_\theta R(\theta)$, we can similarly derive that $dR/d\rho = d\mathcal{L}/d\rho = \alpha^*$. As noted in the proof of Proposition 2, $\alpha^* > 0$. Thus, for sufficiently large $M_t$ such that $\eta_t$ is small, we obtain from a first-order Taylor expansion that $z^T P_t^* \le z^T P^* + c_2 \eta_t$ for some $c_2 \ge 1$.

Hence, there exists a $c_1 > 2c_2$ such that

$$\mathbb{P}_t \left( |z^T P_t^*] - z^T \tilde{P}^*| > \frac{c_1}{2}\eta_t \right) = 0.$$

We rewrite the second term on the right hand side of (15) as

$$z^T \tilde{P}_t^* - z^T P^* = \sum_{m \in \mathcal{M}_t} z_m \frac{p_m^*}{\sum_j p_j^*} - \sum_{n=1}^N z_n p_n^* \quad = \quad \frac{\frac{1}{M_t}\sum_{m \in \mathcal{M}_t} z_m N p_m^*}{\frac{1}{M_t}\sum_j N p_j^*} - \frac{1}{N}\sum_{n=1}^N z_n N p_n^*$$

$$= \frac{\bar{X}(R)}{\bar{X}(P^*)} - \frac{\mu(R)}{\mu(P^*)}, \tag{16}$$

where the last equality makes use of the sample and population means of the two $N$-dimensional vectors, namely $\{x_n(R) = z_n N p_n^*\}$ and the vector with components $x_n(P^*)$ introduced in the proof of Lemma 9.

The Taylor expansion of any smooth function $h(u, v)$ is given by

$$h(u,v) = h(u_o, v_o) + \nabla_\theta h(u_o, v_o) \begin{pmatrix} (u - u_o) \\ (v - v_o) \end{pmatrix} + \begin{pmatrix} (u - u_o) \\ (v - v_o) \end{pmatrix}^T \nabla_\theta^2 h(u_o, v_o) \begin{pmatrix} (u - u_o) \\ (v - v_o) \end{pmatrix} + r(u, v, u_o, v_o),$$

where the higher order terms $r(u, v, u_o, v_o)$ are $o(\|u - u_o\| \cdot \|v - v_o\|)$. Applying this to $h(u, v) = u/v$ with $u = \bar{X}(R)$, $u_o = \mu(R)$, $v = \bar{X}(P^*)$, $v_o = \mu(P^*)$ and $Y = r(u, v, u_o, v_o)$, we obtain

$$|h(u,v) - h(u_o, v_o)| = \left| \frac{\bar{X}(R)}{\bar{X}(P^*)} - \frac{\mu(R)}{\mu(P^*)} \right|$$

$$\le \frac{1}{\mu(P^*)}\left| \bar{X}(R) - \mu(R) \right| + \frac{\mu(R)}{\mu(P^*)^2}\left| \bar{X}(P^*) - \mu(P^*) \right|$$

$$+ \left| \frac{2\mu(R)}{\mu(P^*)^3}(\bar{X}(P^*) - \mu(P^*))^2 - \frac{1}{\mu(P^*)^2}(\bar{X}(R) - \mu(R))(\bar{X}(P^*) - \mu(P^*)) + Y \right|.$$

Note that $\mu(P^*) = 1$ and $\mu(R) > 0$ since the individual scenario losses are non-negative. Further, the proof of Lemma 8 shows that $\bar{X}(P^*) > 0$ with probability at least $(1 - \tau_1)$ as assumed here. This avoids the pathological case where the Taylor expansion of $h(u, v)$ above is undefined because $v$ or $v_o$ is zero. The above expression then yields

$$\mathbb{P}_t \left( \eta_t^{-1}\left| z^T \tilde{P}_t^* - z^T P^* \right| > \frac{c_1}{2} \right)$$

$$\le \mathbb{P}_t \left( \eta_t^{-1}\left| \bar{X}(R) - \mu(R) \right| > \frac{c_1}{6} \right) + \mathbb{P}_t \left( \eta_t^{-1}\left| \bar{X}(P^*) - 1 \right| > \frac{c_1}{6|\mu(R)|} \right)$$

$$+ \mathbb{P}_t \left( \eta_M^{-1}\left| \frac{2\mu(R)}{\mu(P^*)^3}(\bar{X}(P^*) - \mu(P^*))^2 - \frac{1}{\mu(P^*)^2}(\bar{X}(R) - \mu(R))(\bar{X}(P^*) - \mu(P^*)) + Y \right| > \frac{c_1}{6} \right).$$

From the previous applications of Chebychev's inequality in the proof of Lemma 8, we have that the first two terms are of order $\eta_t^{2\delta/(1-\delta)}$. The probability of the second term is already included in $\hat{\tau}$ of Lemma 8, and thus $\tau_3$ is the additional probability term that arises from the first term. Hence, each of the random variables in the first two terms are $O_p(\eta_t)$. The last term involves higher powers of the same random variables that appear in the first two terms. We show that, as expected, the probability of these terms is $o(\eta_t^{2\delta/(1-\delta)})$, or in other words, the random variables represented in these terms are $o_p(\eta_t)$.

Taking the symmetric random variable $Z = |\bar{X}(P^*) - 1|$ and an integer $j > 1$, then $\mathbb{P}(|Z|^j > \eta_t) = \mathbb{P}(|Z| > \eta_t^{1/j}) \le \mathbb{P}(|Z| > \eta_t^{1-\delta'})$, where $\delta' < \delta$ is smaller than the $\delta$ used in the definition of $\eta_t$. Applying Chebychev's inequality renders $\mathbb{P}(|Z|^j > \eta_t) \le O(\eta_t^{2\delta/(1-\delta)+2\delta'/(1-\delta)}) = o(\eta_t^{2\delta/(1-\delta)})$.

The same logic holds for each of the other summands in the remainder, and thus the last term is of smaller order than the first two. Hence, the total probability in this tail term is $o(\eta_t^{2\delta/(1-\delta)})$, which yields the final result. $\square$

**Theorem 3.** *Suppose Assumption 1 and the assumptions of Proposition 1 hold, and further suppose the gradient $\nabla \hat{R}_t(\theta)$ is the optimal solution to (3) over a subsample $\mathcal{M}_t$ of size $M_t$ sampled uniformly without replacement from a set of size $N$. Define $\eta_t = c(\frac{1}{M_t} - \frac{1}{N})^{(1-\delta)/2}$ for small constants $c, \delta > 0$, and set the $D_\phi$-target in (3) to be $\rho_t = \rho + \eta_t$. Then, there exists a small positive $M'$ defined in (13) and of order $o(N)$ such that, for all $M_t \geq M'$, the subgradient $\nabla_\theta \hat{R}_t(\theta)$ and full-gradient $\nabla_\theta R(\theta)$ satisfy for any $C < \infty$ and $1 - \bar{\tau}_t = O(\eta_t^{2\delta/(1-\delta)})$:*

$$\mathbb{P}_t(\eta_t^{-2}\|\nabla_\theta \hat{R}_t(\theta) - \nabla_\theta R(\theta)\|_2^2 \leq C) \geq \bar{\tau}_t.$$

**Proof of Theorem 3:** Recall that the constants $M'$ and $\bar{\tau}_t$ are defined in the statements of Lemma 8 and Lemma 9, respectively. Given the robust loss function $R(\theta) = \sum_n l(\theta, \xi_n)p_n^*$ and our approximation $\hat{R}_t(\theta) = \sum_{m \in \mathcal{M}} l(\theta, \xi_m)\hat{p}_m$ constructed from the subsampled $\mathcal{M}$, the mean-value theorem of calculus yields

$$(\nabla_\theta l(\theta, \xi_n))_u = \frac{\partial l(\theta, \xi_n)}{\partial \theta_u} = \frac{1}{h_{u,n}}(l(\theta + h_{u,n}\mathbf{e}_u, \xi_n) - l(\theta, \xi_n)),$$

where $h_{u,n}$ is a small positive value that depends on the component $\theta_u$ and on the sample $\xi_n$, with $\mathbf{e}_u$ the unit-vector in the $u$th coordinate. Let $\underline{h} = \min_{u,n} h_{u,n}$. We then have

$$\left|(\nabla_\theta \hat{R}_t(\theta) - \nabla_\theta R(\theta))_u\right| \leq \frac{1}{\underline{h}}\left|\sum_n l(\theta + h_{u,n}\mathbf{e}_u, \xi_n)^T(p_n^* - \hat{p}_n) - l(\theta, \xi_n)^T(p_n^* - \hat{p}_n)\right|$$

$$\leq \frac{1}{\underline{h}}\left|\left[\sum_n l(\theta + h_{u,n}\mathbf{e}_u, \xi_n)^T(p_n^* - \hat{p}_n)\right]\right| + \frac{1}{\underline{h}}\left|\left[l(\theta, \xi_n)^T(p_n^* - \hat{p}_n)\right]\right|.$$

Applying the same arguments as those used in the proof of Lemma 9, together with squaring and combining these terms over all $u$, renders the desired final result. $\square$

## A.4  Convergence of Algorithm 1

We start with the proof of Theorem 4, the first part of which is given by the following lemma.

**Lemma 10.** *Suppose the assumptions of Theorem 4 hold. Then, the variance of the estimate $\nabla \hat{R}_t(\theta)$ calculated over the sampled-without-replacement $\mathcal{M}_t$ with subsample size $M_t$ obeys*

$$\mathbb{E}[\|\nabla \hat{R}_t(\theta) - \mathbb{E}[\nabla \hat{R}(\theta)]\|_2^2] \leq C_\sigma \left(\frac{1}{M_t} - \frac{1}{N}\right).$$

*Proof.* Fix the sample size $M_t$ and the parameter $\theta$ for this proof, and let $\mathcal{M}_1$ and $\mathcal{M}_2$ represent two subsets of size $M_t$ sampled without replacement from the full dataset of size $N$. Respectively denote by $\hat{R}_\ell$ and $P_\ell^* = (p_{\ell,i_\ell}^*)$ the estimate of the robust loss and the worst case probability allocation at $\theta$ constructed from $\mathcal{M}_\ell$, for $\ell = 1, 2$. We then use the following equivalent form for variance:

$$\mathbb{E}[\|\nabla \hat{R}_t(\theta) - \mathbb{E}[\nabla \hat{R}(\theta)]\|_2^2] = \mathbb{E}[\|\nabla \hat{R}_1(\theta) - \nabla \hat{R}_2(\theta)\|_2^2]$$

$$= \mathbb{E}[\|\sum_{i_1 \in \mathcal{M}_1} \nabla l(\theta, \xi_{i_1})p_{1,i_1}^* - \sum_{i_2 \in \mathcal{M}_2} \nabla l(\theta, \xi_{i_2})p_{2,i_2}^*\|_2^2]$$

$$\leq \mathbb{E}[\|\sum_{i_1 \in \mathcal{M}_1} \nabla l(\theta, \xi_{i_1})p_{1,i_1}^*\|_2^2] + \mathbb{E}[\|\sum_{i_2 \in \mathcal{M}_2} \nabla l(\theta, \xi_{i_2})p_{2,i_2}^*\|_2^2] \leq 2L \, \mathbb{E}\sum_{i \in \mathcal{M}} (p_{1,i}^*)^2,$$

where the third line uses the $L$-Lipschitz assumption on $\nabla l$. The final expression also applies the fact that the solutions $P_\ell^*$, for $\ell = 1, 2$, are $M_t$-dimensional i.i.d. random variables because they are

functions of the $M_t$-sized vector of objective coefficients $Z_\ell := (z_{m,\ell})$ in the inner maximization formulation (3), each of which are in turn sampled i.i.d. via the sets $\mathcal{M}_\ell$.

To estimate the variance in the solution $P^*$, note first that $(P^*, Z)$ obey (along with the accompanying dual variables $(\alpha^*, \lambda^*)$) the first order optimality conditions on the Lagrangian $\mathcal{L}(\theta, \alpha, \lambda, P)$ in (7) that $\nabla_P \mathcal{L} = \nabla_\alpha \mathcal{L} = \nabla_\lambda \mathcal{L} = 0$. Only the first set of equations $\nabla_P \mathcal{L} = Z - \lambda \mathbf{e} + \alpha \phi'(MP) = 0$ involve $Z$, and the Jacobian of these equations w.r.t. $Z$ is a constant non-singular matrix. From the implicit function theorem, a continuous differentiable $g(\cdot)$ exists such that the relation $P^* = g(Z)$ holds.

Consider a generic function $g(Y)$ of a random variable $Y$ such that $g'(\cdot)$ exists. Let $Y_1$ and $Y_2$ be two i.i.d. replications of $Y$, and observe from the mean value theorem that

$$\mathbb{E}[|g(Y_1) - g(Y_2)|^2] = \mathbb{E}[|g(Y_1) - g(\zeta) + g(\zeta) - g(Y_2)|^2] \leq |g'(\zeta)|^2 \mathbb{E}[|Y_1 - Y_2|^2].$$

Thus, the variance in $g(Y)$ follows at least the same rate as the variance in $Y_1$ if the gradient $g'$ exists. Applying this to our case, note that the vector $Z$ is sampled without replacement from the full dataset $\{Z_m, \; m \in \mathcal{N}\}$ implicitly by the subsampled dataset $\mathcal{M}$, which leads us to the desired result. $\square$

**Theorem 4.** *Suppose the constant step size $\gamma_t = \gamma$ satisfies $\gamma \leq \frac{1}{2L}$, a lower bound $R_{\inf}$ exists for the robust loss function $R(\theta)$, $\forall \theta \in \Theta$, and the conditions of Theorem 3 hold. Then, the variance of the estimate $\nabla \hat{R}_t(\theta)$ calculated over the sampled-without-replacement $\mathcal{M}_t$ with subsample size $M_t$ obeys $\mathbb{E}[\|\nabla \hat{R}_t(\theta) - \mathbb{E}[\nabla \hat{R}(\theta)]\|_2^2] \leq C(\frac{1}{M_t} - \frac{1}{N})$. Further assume the gradient $\nabla_\theta R(\theta)$ is $L$-Lipschitz. Then, at termination,*

$$\sum_{t=1}^{\mathcal{T}} \|\nabla_\theta R(\theta_t)\|_2^2 \leq \frac{R(\theta_0) - R_{\inf}}{\frac{\gamma}{2}(2 - L\gamma)} + C\frac{L\gamma + 1}{2 - L\gamma} \sum_{t=1}^{\mathcal{T}} \eta_t^2. \tag{4}$$

 **Proof of Theorem 4:** For any $\theta$ and a set $\mathcal{M}_t$ sampled to have $M_t$ support points, Theorem 3 and Lemma 10 show that

$$\mathbb{E}_t\left[\|\nabla_\theta \hat{R}_t(\theta) - \nabla_\theta R(\theta)\|_2^2\right] \leq \mathbb{E}_t\left[\|\nabla_\theta \hat{R}_t(\theta) - \mathbb{E}_t[\nabla_\theta \hat{R}_t(\theta)]\|_2^2\right] + \|\mathbb{E}_t[\nabla_\theta \hat{R}_t(\theta)] - \nabla_\theta R(\theta)\|_2^2$$

$$\leq O(\eta_t^{2/(1-\delta)}) + O(\eta_t^2) \;=\; O(\eta_t^2).$$

Hence, the slower rate of decrease in the bias prevails as the rate at which the mean squared error decreases to zero. Elementary algebraic manipulations yield the following two implications:

$$\mathbb{E}_t\left[\|\nabla_\theta \hat{R}_t(\theta)\|_2^2\right] \;\leq\; C\eta_t^2 + \|\nabla_\theta R(\theta)\|_2^2, \quad \text{and} \tag{17}$$

$$-\mathbb{E}_t\left[\left(\nabla_\theta \hat{R}_t(\theta)\right)^T \nabla_\theta R(\theta)\right] \;\leq\; C\eta_t^2 - \|\nabla_\theta R(\theta)\|_2^2 - \mathbb{E}_t\left[\|\nabla_\theta \hat{R}_t(\theta)\|_2^2\right]. \tag{18}$$

We can therefore bound the expected robust loss at step $(t + 1)$ using

$$R_{\inf} \leq \mathbb{E}_t[R(\theta_{t+1})]$$

$$\leq \mathbb{E}_t[R(\theta_t)] - \gamma \mathbb{E}_t[\nabla_\theta R(\theta_t)^T \nabla_\theta \hat{R}_t(\theta_t)] + \frac{L\gamma^2}{2} \mathbb{E}_t\left[\|\nabla_\theta \hat{R}(\theta_t)\|_2^2\right]$$

$$\leq \mathbb{E}_t[R(\theta_t)] + \frac{C\gamma\eta_t^2}{2} - \frac{\gamma}{2}\|\nabla_\theta R(\theta_t)\|_2^2 + \left(\frac{L\gamma^2 - \gamma}{2}\right)\left(\|\nabla_\theta R(\theta_t)\|_2^2 + C\eta_t^2\right)$$

$$= \mathbb{E}_t[R(\theta_t)] + \frac{C\gamma\eta_t^2}{2}(L\gamma + 1) - \frac{\gamma}{2}(2 - L\gamma)\|\nabla_\theta R(\theta_t)\|_2^2. \tag{19}$$

Upon rearranging (19) and telescoping back to the initial iterate $\theta_0$, we obtain the desired result. $\square$

The main implication of the result in (4) of Theorem 4 is that if the summation term $\sum_{t=1}^{\mathcal{T}} \left(\frac{1}{M_t} - \frac{1}{N}\right)^{(1-\delta)}$ remains finite as $\mathcal{T} \to \infty$, then the sum of the norm of the gradients of $R(\theta_t)$ at iterates visited by the algorithm is bounded above, and hence the gradients $\nabla_\theta R(\theta_t)$ converge to 0; in other words, the algorithm converges to a local optimal solution. Even moderately increasing the subsample growth sequences $\{M_t\}$ can satisfy this condition, and in fact the sequences can be designed such that they satisfy this condition in addition to $M_t \nearrow N$ as $t \to \infty$ (that is, $\mathcal{T}$ is not finite), thus realizing a local optimal solution within the run of Algorithm 1.

**Example 1.** *Let $M_t = Nt^{k(1-\delta)}/(N + t^{k(1-\delta)})$ for any $k \geq 2$. The sequence can be seen to be increasing (assuming $M_t$ is a continuous function of t) from $\partial M_t(t)/\partial t = k(1-\delta)Nt^{k(1-\delta)-1}/(N + t^{k(1-\delta)})^2 > 0$ for all t. Moreover, $M_t < N$ and $\lim_{t\to\infty} M_t = \lim_{t\to\infty} N \cdot (1/(1 + N/t^{k(1-\delta)})) = N$. Therefore $M_t \nearrow N$ and $\mathcal{T}$ is not finite. In addition,*

$$\sum_{t<\infty} \left(\frac{1}{M_t} - \frac{1}{N}\right)^{(1-\delta)} = \sum_{t<\infty} \frac{1}{t^k} < \infty,$$

*by the assumption that $k \geq 2$. Thus, if all the other conditions of Theorem 4 are satisfied, this growth sequence yields $\nabla_\theta R(\theta_t) \to 0$ as $t \to \infty$.*

Example 1 shows that the progressively sampled subgradient algorithm can converge to locally optimal solutions. However, it may do so at the expense of heavy computational effort, since the sequence in the example grows at a diminishing rate with the iterations. Theorem 6 teases out the tradeoffs inherent in this question of computational efficiency for the class of strongly convex losses.

We now present a series of structural lemmas that build up to the proof of Theorem 6. We start with the first part of the result, which pertains to the strong convexity of the robust loss function $R(\theta)$ arising from the strong-convexity of the individual loss functions $l(\theta, \xi)$.

**Lemma 11.** *Suppose all the conditions of Theorem 4 are satisfied and the loss functions $l(\theta, \xi_n)$ are c-strongly convex. Then the function $R(\theta)$ is c-strongly convex.*

*Proof.* Since each $l(\theta, \xi_n)$ is c-strongly convex, we have

$$l(\theta_1, \xi_n) + \nabla_\theta l(\theta_1, \xi_n)^T(\theta_2 - \theta_1) + \frac{c}{2}\|\theta_2 - \theta_1\|_2^2 \leq l(\theta_2, \xi_n).$$

Taking any pmf $P$ with components $p_n$ and summing up each side, we obtain

$$\sum_n p_n \left(l(\theta_1, \xi_n) + \nabla_\theta l(\theta_1, \xi_n)^T(\theta_2 - \theta_1) + \frac{c}{2}\|\theta_2 - \theta_1\|_2^2\right) \leq \sum_n p_n l(\theta_2, \xi_n).$$

Since the above holds for any $P$, apply this for $P^*(\theta_1)$, the optimal pmf for the inner maximization that defines $R(\theta_1)$, with components $p_n^*(\theta_1)$. As previously discussed, if the $D_\phi$-constraint is tight enough, i.e., $\rho < \rho_1$, then $P^*(\theta_1)$ is unique for $\theta_1$, and thus the subgradient $\nabla_\theta R(\theta)$ corresponds with the gradient from Danskin's Theorem (see Shapiro et al. [29, Theorem 7.21, p. 352]). We then derive

$$\left(\sum_n p_n^*(\theta_1) l(\theta_1, \xi_n)\right) + \sum_n p_n^*(\theta_1) \nabla_\theta \left(l(\theta_1, \xi_n)\right)^T (\theta_2 - \theta_1) + \frac{c}{2}\|\theta_2 - \theta_1\|_2^2 \leq \sum_n p_n^*(\theta_1) l(\theta_2, \xi_n),$$

which leads to

$$R(\theta_1) + \nabla_\theta R(\theta_1)^T(\theta_2 - \theta_1) + \frac{c}{2}\|\theta_2 - \theta_1\|_2^2 \leq \max_{P \in \mathcal{P}} \sum_n p_n l(\theta_2, \xi_n) = R(\theta_2),$$

thus verifying that $R(\theta)$ is c-strongly convex. $\qquad\square$

The strong convexity of $R(\theta)$, along with the strict convexity of the feasible set $\mathcal{P}$ posited in Proposition 1(i), implies that there exists a unique minimizer $\theta_{\text{rob}}$ such that $R(\theta_{\text{rob}}) = R_{\text{inf}}$. This leads to a stronger version of the convergence shown in Theorem 4. Recall that the definition of the expected optimality gap in this case is $E_t := \mathbb{E}_t[R(\theta_t)] - R(\theta_{\text{rob}})$, and the sample size sequences are defined as $M_t(\nu) = \Pi_{s \leq t} \nu_s$ where $\nu_s > 1$ to ensure the subsample size grows starting with $M_0 = 1$.

**Lemma 12.** *Suppose the assumptions of Proposition 1(i), Theorem 4 and Lemma 11 hold. Let $r = 1 - \frac{\gamma}{4c} < 1$. Then, the expected optimality gap $E_{t+1}$ after t iterations satisfies*

$$E_{t+1} \leq r^{t+1} E_0 + \frac{CL\gamma^2}{2} \sum_{s=0}^{t} r^{t-s} \left(\prod_{u=0}^{s} \nu_u^{-(1-\delta)}\right). \tag{20}$$

*Proof.* Returning to the inequality (19) at the end of the proof of Theorem 4, we rewrite this for the strongly convex $R(\theta)$ as

$$
\begin{aligned}
E_{t+1} = \mathbb{E}_t[R(\theta_{t+1})] &- R(\theta_{\text{rob}}) \\
&\leq \left( \mathbb{E}_t[R(\theta_t)] - R(\theta_{\text{rob}}) \right) - \gamma \mathbb{E}_t[\nabla_\theta R(\theta_t)^T \nabla_\theta \hat{R}_t(\theta_t)] + \frac{L\gamma^2}{2} \mathbb{E}_t \left[ \|\nabla_\theta \hat{R}(\theta_t)\|_2^2 \right] \\
&\leq E_t + \frac{C\gamma\eta_t^2}{2} - \frac{\gamma}{2} \|\nabla_\theta R(\theta_t)\|_2^2 + \left( \frac{L\gamma^2 - \gamma}{2} \right) \left( \|\nabla_\theta R(\theta_t)\|_2^2 + C\eta_t^2 \right) \\
&\leq E_t \left( 1 - \frac{2\gamma - L\gamma^2}{4c} \right) + \frac{CL\gamma^2\eta_t^2}{2} \quad \leq \quad \left( 1 - \frac{\gamma}{4c} \right) E_t + \frac{CL\gamma^2\eta_t^2}{2}. \quad (21)
\end{aligned}
$$

The first inequality starts with the $L$-Lipschitzness of $\nabla_\theta R(\cdot)$, and the second inequality substitutes the relations in (17) and (18). The third inequality uses the $c$-strong convexity of $R(\theta)$, specifically the implication that $\|\nabla_\theta R(\theta)\|_2^2 / 2c \geq (R(\theta) - R(\theta_{\text{rob}}))$. The final inequality utilizes the conditions imposed on $\gamma$.

The form of (21) is quite informative, in that it clearly displays the tradeoff being addressed by the algorithm: the first summand provides a geometric reduction in the optimality gap, which is to be balanced with the stochastic error in the second summand. Note that, for growth sequences $M_t = M_o \prod_{s=1}^t \nu_s$ where $\nu_s > 1$, we have

$$
\frac{\eta_t^2}{\eta_{t-1}^2} = \left( \frac{\frac{1}{M_t} - \frac{1}{N}}{\frac{1}{M_{t-1}} - \frac{1}{N}} \right)^{(1-\delta)} = \frac{1}{\nu_t^{(1-\delta)}} \left( \frac{\frac{1}{M_0 \prod_{s=1}^t \nu_s} - \frac{1}{N}}{\frac{1}{M_0 \prod_{s=1}^t \nu_s} - \frac{1}{N} + \frac{1}{N}\left( 1 - \frac{1}{\nu_t} \right)} \right)^{(1-\delta)} \leq \frac{1}{\nu_t^{(1-\delta)}}.
$$

Therefore, the $\eta_s^2$ decreases at least as fast as $(1 - 1/N) \prod_{u=0}^s \nu_u^{-(1-\delta)} \leq \prod_{u=0}^s \nu_u^{-(1-\delta)}$, where the term $(1 - 1/N)$ arises from $\eta_0^2$. We combine this with telescoping the optimality gap in the first summand of (21), which leads to the expression in (20) as follows:

$$
\begin{aligned}
E_{t+1} &\leq rE_t + \frac{CL\gamma^2\eta_t^2}{2} \leq r^{t+1} E_0 + \frac{CL\gamma^2}{2} \sum_{s=0}^t \eta_s^2 r^{t-s} \\
&\leq r^{t+1} E_0 + \frac{CL\gamma^2}{2} \sum_{s=0}^t r^{t-s} \left( \prod_{u=0}^s \nu_u^{-(1-\delta)} \right).
\end{aligned}
$$

$\square$

From (20), we observe that the error after $t$ steps is a result of the balance between $r$ and $\nu_t$, and the summation in the second term is key to analyzing this tradeoff. We denote the summation as $I$ in the sequel. The rate at which the overall optimality gap drops to zero is then governed by the balance achieved between the fast geometric (or linear as termed by the optimization literature) drop in deterministic error by the factor $r$ and the rate at which the stochastic error, driven by the growth sequences $\nu_s$, recedes. We start by showing that *diminishing-growth* subsample sequences lead to a slow drop in the second term, that is the stochastic error, which in turn implies a slow decay of the overall expected optimization gap $E_{t+1}$. Recall that a sequence is said to have *diminishing-growth* if $\nu_s \searrow 1$ as $s \to \infty$, and that $W_t = \sum_{s \leq t} w_s$ represents the total computational effort expended up until the $t$th iterate, where $w_t = k_e M_t \log M_t$ represents the computational effort of the $t$th iterate for some constant $k_e$.

**Theorem 6** *Suppose all the conditions of Theorem 4 are satisfied and the loss functions $l(\theta, \xi_n)$ are $c$-strongly convex. Then the function $R(\theta)$ is $c$-strongly convex. Further suppose that $\gamma \leq \min\{\frac{1}{4L}, 4c\}$ is fixed. Then: (i) For diminishing growth sequences $M_t$, we have that $E_\mathcal{T} W_\mathcal{T} \to \infty$ as $\nu_t \to 1^+$; and (ii) For constant growth sequences, we have that the total effort $W_\mathcal{T}(\nu)$ is a decreasing function over $\nu \in (1, \infty)$, with $\lim_{\nu \to 1^+}(\nu - 1)W_\mathcal{T}(\nu) = k_e(N \log N - (N - 1))$. Moreover, $E_\mathcal{T}(\nu) \leq \bar{E}_\mathcal{T}(\nu)$ where $\bar{E}_\mathcal{T}(\nu)$ is an increasing function over $\nu \in (1, \infty)$ with its infimum $\bar{E}^* = \inf_\nu \bar{E}_\mathcal{T}(\nu) = (1/N^{1-\delta})(E_0 + 8CLc^2)$ attained as $\nu \to 1^+$. Finally, we have*

$$
\lim_{\nu \to 1^+} W_\mathcal{T}(\nu) \ (\bar{E}_\mathcal{T}(\nu) - \bar{E}^*) = 8CLc^2(1-\delta)(1 - \frac{\gamma}{4c}) \cdot k_e \left( N^\delta \log N - (N^\delta - N^{-(1-\delta)}) \right).
$$

**Proof of Theorem 6(i):** Since $\nu_s \searrow 1$, there exists sufficiently large $s_o$ such that $\nu_s^{(1-\delta)} r < 1$ for all $s \geq s_o$. Then the term $I$ in (20) can be written as

$$
I \leq \sum_{s=0}^{s_o} r^{t-s} \left( \prod_{u=0}^{s} \nu_u^{-(1-\delta)} \right) + \frac{1}{\nu_{s_o}^{(t-s_o)(1-\delta)}} \left[ 1 + r\nu_{s_o}^{(1-\delta)} + \ldots + (r\nu_{s_o}^{(1-\delta)})^{t-s_o} \right]
$$

$$
\leq \sum_{s=0}^{s_o} r^{t-s} \left( \prod_{u=0}^{s} \nu_u^{-(1-\delta)} \right) + \frac{1}{\nu^{(t-s_o)(1-\delta)}} \left[ \frac{1}{1 - r\nu_{s_o}^{(1-\delta)}} \right].
$$

Thus, the term $I$ forms the dominating term in determining the rate of convergence of $E_{t+1}$, which is $\prod_{s=s_o}^{t} \nu_{s_o}^{-(1-\delta)} \leq \prod_{s=s_o}^{t} \nu_s^{-(1-\delta)} = O(M_t^{-(1-\delta)})$.

We will now show that the computational effort $w_t = O(M_t \log M_t)$ is also $w_t = o(W_t)$. These together imply the desired result for the case of Theorem 6(i). Consider the function $h_\epsilon(k) = (1/k^\epsilon)^{1/k^{2\epsilon}}$ over $k \in [1, \infty)$ for any $\epsilon > 0$. Several properties of $h_\epsilon$ are apparent: $h_\epsilon(k) = h_1(\epsilon k)$ for any $\epsilon > 0$; $h_1(k) \in (0, 1]$ for all $k \in [1, \infty)$; and $\lim_{k \to 1^+} h_1(k) = \lim_{k \to \infty} h_1(k) = 1$. Moreover, $\arg\min_k h_1(k) = e^{1/2}$, where $h_1(e^{1/2}) = 0.832$. Hence, the family of functions $\{h_\epsilon\}$ (indexed by $\epsilon$) can be used to bound the sequence $\nu_s^{-1} \nearrow 1$ in the following sense: for a sufficiently large $t_0$ there exists an $\epsilon = \epsilon(t_0) > 0$ such that

$$
1 \geq \nu_s^{-1} \geq h_{\epsilon/2}(t) = \left( \frac{1}{t^{\epsilon/2}} \right)^{1/t^\epsilon} \geq 0, \quad \forall t \geq s \geq t_0(\epsilon).
$$

We then obtain

$$
\frac{W_t}{w_t} = \frac{\sum_{s=1}^{t} M_s \log M_s}{M_t \log M_t} \geq \sum_{s=t-t^\epsilon}^{t} \prod_{u=s+1}^{t} \nu_u^{-1} \geq t^\epsilon \prod_{u=t-t^\epsilon}^{t} \nu_u^{-1} \geq t^{\epsilon/2}.
$$

Hence, as $t \to \infty$, we have that $W_t/w_t \to \infty$. The total work $W_t$ performed up until iteration $t$ therefore grows faster than the instantaneous work $w_t$ (as represented by the instantaneous sample size $M_t$), but the second term in (20) does not drop any faster than the subsample size $M_t^{(1-\delta)}$, which leads to the desired result in Theorem 6(i). $\square$

In the sequel, we limit our discussion to the *constant growth* sequences that are relevant to the remaining parts of Theorem 6. A constant growth sequence has $M_t = \lfloor \nu^t \rfloor$ for $\nu \in (1, \infty)$, and the iteration when the growth sequence reaches the full training dataset size $N$ is $\mathcal{T}(\nu) = \lceil \log N / \log \nu \rceil$. As stated before, we ignore the requirement that $M_t$ and $\mathcal{T}$ be integers for the rest of the section for simplicity of exposition. The total number of iterations $\mathcal{T}(\nu)$ for any $\nu \in (1, \infty)$ is finite. At termination $\mathcal{T}(\nu)$, we write the cumulative computational effort by iteration $\mathcal{T}(\nu)$ as $W_{\mathcal{T}}(\nu) := \sum_{s=0}^{\mathcal{T}} w_s$. We begin by establishing the size of $W_{\mathcal{T}}(\nu)$.

**Lemma 13.** *We have:*
*(a) for any $\nu > 1$,*

$$
W_{\mathcal{T}}(\nu) = k_e \left\{ \left( \frac{\nu}{\nu - 1} \right) N \log N - (N - 1) \left( \frac{\nu}{\nu - 1} \right) \left( \frac{\log \nu}{\nu - 1} \right) \right\};
$$

*(b) $W_{\mathcal{T}}(\nu)$ is strictly decreasing over $\nu \in (1, N]$; and*
*(c)* $\lim_{\nu \to 1^+} (\nu - 1) W_{\mathcal{T}}(\nu) = k_e \{ N \log N - (N - 1) \}$.

*Proof.* From the definition of $W_{\mathcal{T}}(\nu)$, we derive

$$W_{\mathcal{T}}(\nu) = k_e \sum_{s \leq \mathcal{T}} M_s \log M_s = k_e \sum_{s \leq \mathcal{T}} \nu^s \log(\nu^s) = k_e \log \nu \sum_{s \leq \mathcal{T}} s\nu^s = k_e \nu \log \nu \sum_{s \leq \mathcal{T}} (s)\nu^{(s-1)}$$

$$= k_e \nu \log \nu \frac{\partial}{\partial \nu}\left( \sum_{s \leq \mathcal{T}} \nu^s \right) = k_e \frac{\nu \log \nu}{(\nu-1)^2}\left( \mathcal{T}\nu^{(\mathcal{T}+1)} - \nu^{\mathcal{T}}(\mathcal{T}+1) + 1 \right)$$

$$= k_e \frac{\nu \log \nu}{(\nu-1)^2}\left( \mathcal{T}\nu^{\mathcal{T}}(\nu-1) - \nu^{\mathcal{T}} + 1 \right)$$

$$= k_e \frac{\nu \log \nu}{(\nu-1)^2}\left( \frac{\log N}{\log \nu} N(\nu-1) - (N-1) \right),$$

where the last expression uses the relation $\nu^{\mathcal{T}} = N$. Simplifying the expression leads to the first result (a).

Turning to (b), consider the derivative

$$W_{\mathcal{T}}'(\nu) = \frac{-(N-1)}{(\nu-1)^2}\left( \frac{N \log N}{N-1} + 1 - \frac{\log \nu(\nu+1)}{\nu-1} \right).$$

To establish the desired result, it is sufficient to show that the expression within the parentheses is always positive. Note that we need only consider the cases $\nu \leq N$ because any larger value is in effect truncated to $N$. Define

$$\omega(\nu) := \frac{\log \nu(\nu+1)}{\nu-1} \quad \text{and calculate its derivative} \quad \omega'(\nu) = \frac{1}{(\nu-1)^2}\left[ \nu - \frac{1}{\nu} - 2\log \nu \right].$$

The derivative is non-negative since the term $\omega_2(\nu) = \nu - 1/\nu - 2\log \nu$ within the parentheses takes the value 0 as $\nu \to 1^+$ and it is increasing in $\nu \in (1, N]$, since $\omega_2'(\nu) = (1 - 1/\nu)^2$. Thus, the function $\omega(\nu)$ is increasing in $\nu$; this, in combination with the fact that $\omega(N) = (N \log N)/(N-1) + \log N/(N-1) < (N \log N)/(N-1) + 1$, yields the desired result.

Finally, the limit in (c) is readily obtained by multiplying both sides of the equation in (a) with $(\nu-1)$ and noting from L'Hospital's rule that $\lim_{\nu \to 1^+} \log \nu/(\nu-1) = 1$ for the second term. $\square$

Following (20), we establish an upper bound $\bar{E}_{\mathcal{T}}$ on how the expected optimality gap $E_{\mathcal{T}}$ at iteration $\mathcal{T}$ drops as a function of the various parameters related to the algorithm and the objective function.

**Lemma 14.** *For any $\nu \in (1, \infty)$, the expected optimality gap at termination $\mathcal{T}(\nu)$ is bounded by*

$$E_{\mathcal{T}}(\nu) \leq \bar{E}_{\mathcal{T}}(\nu) = \begin{cases} \frac{1}{N^{(1-\delta)}}\left[ E_0 + \frac{CL\gamma^2}{2} \frac{1-r^{(\log N/\log \nu)}N^{(1-\delta)}}{1-r\nu^{(1-\delta)}} \right], & \nu < (1/r)^{(1/(1-\delta))} \\ \frac{1}{N^{(1-\delta)}}\left[ E_0 + \frac{CL\gamma^2}{2}\left( \frac{\log N}{\log \nu} \right) \right], & \nu = (1/r)^{(1/(1-\delta))} \\ r^{\left(\frac{\log N}{\log \nu}\right)}\left[ E_0 + \frac{CL\gamma^2}{2} \frac{1-r^{(-\log N/\log \nu)}N^{-(1-\delta)}}{1-\left(\nu^{-(1-\delta)}/r\right)} \right], & \nu > (1/r)^{(1/(1-\delta))} \end{cases}.$$

*Proof.* We analyze the summation term $I$ in (20) of Lemma 12 for the three cases listed in the above result. For the top case where $\nu^{(1-\delta)} < 1/r$, the term $I$ can be written as

$$I = \frac{1}{\nu^{t(1-\delta)}}\left[ 1 + r\nu^{(1-\delta)} + \ldots + (r\nu^{(1-\delta)})^{(t-1)} \right] = \frac{1}{\nu^{t(1-\delta)}}\left[ \frac{1 - (r\nu^{(1-\delta)})^t}{1 - r\nu^{(1-\delta)}} \right].$$

The last expression in turn shows that the optimality gap $E_{t+1}$ decreases at the dominating (slower) rate of $\nu^{t(1-\delta)}$ such that the term $I$ vanishes to zero. Upon substituting $t = \mathcal{T} = \log N/\log \nu$, we obtain the corresponding expression for the top case in the theorem.

Now, for the bottom case, assume that $\nu^{(1-\delta)} > 1/r$, which allows the term $I$ to be written as

$$I = r^t\left[ 1 + \frac{1}{r\nu^{(1-\delta)}} + \ldots + \frac{1}{(r\nu^{(1-\delta)})^{(t-1)}} \right] = r^t\left[ \frac{1 - \frac{1}{(r\nu^{(1-\delta)})^t}}{1 - \frac{1}{r\nu^{(1-\delta)}}} \right].$$

The last expression shows that the optimality gap $E_{t+1}$ is dominated by the (slower) rate of the deterministic convergence $r^t$. Once again, substituting $t = \mathcal{T} = \log N / \log \nu$ renders the corresponding expression for the bottom case in the theorem.

Finally, for the middle case $\nu^{(1-\delta)} = 1/r$, we start by observing that

$$\lim_{f \to 1^-} \frac{1 - f^{(t+1)}}{1 - f} = \lim_{f \to 1^-} (t+1) f^t = (t+1), \tag{22}$$

where the second expression applies L'Hospital's rule. The corresponding desired result for the middle case can now be obtained by taking the left- or right-limit as appropriate of the top or bottom cases. □

The total number of iterations $\mathcal{T}(\nu)$ for any $\nu \in (1, \infty)$ is finite, i.e., $\mathcal{T}(\nu) < \infty$, and thus we cannot expect to match the corresponding guarantee $\bar{E}_{\mathcal{T}(\nu)}$ to any arbitrarily desired optimality precision. We can in fact establish that the infimum of $\bar{E}_{\mathcal{T}(\nu)}$ over $\nu \in (1, \infty)$, denoted by $\bar{E}^*$, exists and is available in closed form.

**Lemma 15.** *The upper bound $\bar{E}_{\mathcal{T}}(\nu)$ in Lemma 14 is increasing in $\nu$, and attains its infimum when $\nu \to 1^+$ as*

$$\bar{E}^* = \inf_{\nu \in (1, \infty)} \bar{E}_{\mathcal{T}}(\nu) = \frac{1}{N^{(1-\delta)}} \left[ E_0 + \frac{CL\gamma^2}{2(1-r)} \right].$$

*Proof.* For this proof, we respectively denote by $f'(\nu)$ and $f''(\nu)$ the first and second derivatives of a function $f$ of $\nu$. We begin by establishing that the error bound $\bar{E}_{\mathcal{T}}(\nu) \geq \bar{E}^*$ for all $\nu \in (1, \infty)$. Let us rewrite the difference between them for $\nu^{1-\delta} \in (1, 1/r)$ using the expression in Lemma 14 as

$$\Delta \bar{E}_{\mathcal{T}}(\nu) := \bar{E}_{\mathcal{T}}(\nu) - \bar{E}^* = \frac{CL\gamma^2}{2} N^{-(1-\delta)} \left[ \frac{1 - (r\,\nu^{1-\delta})^{\mathcal{T}}}{1 - r\,\nu^{1-\delta}} - \frac{1}{1-r} \right], \tag{23}$$

where $\mathcal{T} = \mathcal{T}(\nu) = \log N / \log \nu \to \infty$ as $\nu \to 1^+$. We mainly present the analysis for the case of $r\nu < 1$ above, noting that the $r\nu > 1$ case follows analogously by the symmetry of their respective expressions, as we further explain below.

First observe that $\lim_{\nu \to 1^+} \Delta \bar{E}_{\mathcal{T}}(\nu) = 0$. Denote by $t_1(\nu)$ the first term within the brackets in (23). We then establish that $t_1(\nu)$ is increasing in $\nu$ by showing that $t_1'(\nu) > 0$, which establishes that $\bar{E}_{\mathcal{T}}$ is increasing in $\nu^{1-\delta} \in (1, 1/r)$.

Substituting $\tilde{\nu} = \nu^{1-\delta}$, since $t_1'(\tilde{\nu}) = t_1'(\nu)(1-\delta)\nu^{-\delta}$, it is sufficient to show that $t_1'(\tilde{\nu}) > 0$ for all $\nu^{1-\delta} \in (1, 1/r)$. The corresponding range for $\tilde{\nu}$ is $(1, 1/r)$, and $\mathcal{T}(\tilde{\nu}) = (1-\delta) \log N / \log \tilde{\nu}$. Let $f_n(\nu) = 1 - (r\tilde{\nu})^{\mathcal{T}(\tilde{\nu})}$ and $f_d(\tilde{\nu}) = 1 - r\tilde{\nu}$. Then, to establish that $t_1'(\tilde{\nu}) > 0$, it is sufficient to show that $g(\tilde{\nu}) := f_n'(\tilde{\nu}) f_d(\tilde{\nu}) - f_d'(\tilde{\nu}) f_n(\tilde{\nu}) > 0$. We derive that $f_n'(\tilde{\nu}) = -(r\tilde{\nu})^{\mathcal{T}} (\mathcal{T}'(\tilde{\nu}) \log r)$ and $f_d'(\tilde{\nu}) = -r$, which yields

$$g(\tilde{\nu}) = r - (r\tilde{\nu})^{\mathcal{T}} \left[ r + \log r\ \mathcal{T}'(\tilde{\nu})(1 - r\tilde{\nu}) \right],$$

where $T'(\tilde{\nu}) = -\log N / (\tilde{\nu}(\log \tilde{\nu})^2)(1 - \delta) < 0$ and $\log r < 0$. We now have that $\lim_{\tilde{\nu} \to 1/\tilde{r}} g(\tilde{\nu}) = 0$, and thus the desired result that $g(\tilde{\nu}) \geq 0$ over $\tilde{\nu} \in (1, 1/r)$ is obtained if its derivative in turn can be shown to satisfy $g'(\tilde{\nu}) < 0$. Denote by $g_1(\tilde{\nu}) = (r\tilde{\nu})^{\mathcal{T}}$ and $g_2(\tilde{\nu}) = \log r\ \mathcal{T}'(\tilde{\nu})(1 - r\tilde{\nu})$, and rewrite $g(\tilde{\nu}) = r - g_1(\tilde{\nu})[r + g_2(\tilde{\nu})]$. We therefore conclude that

$$g'(\tilde{\nu}) = -[g_1'(\tilde{\nu})(r + g_2(\tilde{\nu})) + g_2'(\tilde{\nu}) g_1(\tilde{\nu})].$$

Since $g_1(\tilde{\nu})$ and $g_2(\tilde{\nu})$ are each non-negative, we have the desired outcome if their derivatives are also non-negative. To see that this holds, we simplify their expressions to obtain

$$g_1'(\tilde{\nu}) = T'(\tilde{\nu}) \log r\ g_1(\tilde{\nu}) \quad \text{and} \quad g_2'(\tilde{\nu}) = T''(\tilde{\nu})(1 - r\tilde{\nu}) - rT'(\tilde{\nu}),$$

where $T''(\tilde{\nu}) = (1-\delta) \log N (\log \tilde{\nu} + 2) / (\tilde{\nu}^2 (\log \tilde{\nu})^3) > 0$ for all $\tilde{\nu} \in (1, \infty)$. Hence, both $g_1'(\tilde{\nu}) \geq 0$ and $g_2'(\tilde{\nu}) \geq 0$, thus rendering that the first term $t_1(\tilde{\nu})$ is increasing, and implying in turn that $\bar{E}_{\mathcal{T}}(\nu)$ is also increasing in $\nu^{1-\delta} \in (1, 1/r)$.

For the case when $\nu^{1-\delta} \in (1/r, \infty)$, observe that

$$\Delta\bar{E}_{\mathcal{T}}(\nu) := \bar{E}_{\mathcal{T}}(\nu) - \bar{E}^* \;=\; E_0 \left[ r^{\mathcal{T}} - \frac{1}{N^{1-\delta}} \right] + \frac{CL\gamma^2}{2} \; r^{\mathcal{T}} \; \left[ \frac{1 - (r(\nu)^{1-\delta})^{-\mathcal{T}}}{1 - (r\nu^{1-\delta})^{-1}} - \frac{1}{1-r} \right]. \tag{24}$$

The first term is increasing in $\nu$ since $\mathcal{T}(\nu)$ is decreasing in $\nu$ and $r < 1$; note that $r^{\mathcal{T}} > N^{-(1-\delta)} = \nu^{-\mathcal{T}(1-\delta)}$ by the supposition that $\nu^{(1-\delta)} > 1/r$. Observe the similarity of the second term to the expression for $\Delta\bar{E}$ that was analyzed for the $\nu^{(1-\delta)} < 1/r$ case. In particular, if we let $y = 1/\nu^{1-\delta}$, then $y \le r$ and we are interested in how the term changes as $y \searrow 0$ receding away from the critical value of $r$. The properties of the term can be analyzed analogously to the above case to show that their behaviour is reciprocated, in that as $\nu \to \infty$ the error gap $\Delta\bar{E}_{\mathcal{T}}$ grows. We omit the details for the sake of brevity, since the results follow in an analogous manner from the above analysis.

Hence, the error $\bar{E}_{\mathcal{T}}(\nu)$ is bounded below by $\bar{E}^*$ everywhere in $\nu \in (1, \infty)$, and moreover the analysis above shows that this infimum is obtained as $\nu \to 1^+$. $\qquad\square$

Lemma 15 shows that there exists an infimum beyond which our Algorithm 1 cannot be guaranteed to reduce the upper bound $\bar{E}_{\mathcal{T}(\nu)}$ on the optimality gap $E_{\mathcal{T}(\nu)}$ for any constant-factor growth parameter $\nu$ or step size $\gamma$. Figure 3 provides an illustration of the results provided by Lemmas 14 and 15.

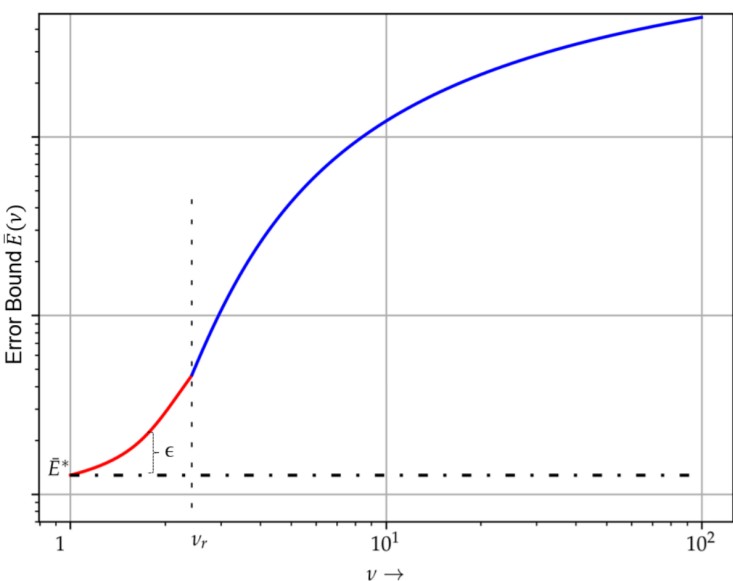

Figure 3: Illustration of guaranteed expected optimality gap at finite-step $\mathcal{T}$ termination when Algorithm 1 is run with a constant growth sequence $M_t = \lfloor \nu^t \rfloor$ for $\nu > 1$. The critical value $\nu_r$ satisfies $\nu_r^{1-\delta} = 1/r$. The three regimes obeyed by the guaranteed optimality gap $\bar{E}(\nu)$ as described in Lemma 14 is depicted on and to either side of this critical value $\nu_r$. The best error guarantee $\bar{E}^*$, defined in Lemma 15, is attained as $\nu \to 1^+$.

Note that the proof of Lemma 15 defines $\Delta\bar{E}_{\mathcal{T}}(\nu) := \bar{E}_{\mathcal{T}}(\nu) - \bar{E}^*$ to be the attainable gap at any $\nu \in (1, \infty)$, and shows that $\Delta\bar{E}_{\mathcal{T}}(\nu) \to 0$ as $\nu \to 1^+$. Our next lemma characterizes a finer analysis by providing a rate for this convergence.

**Lemma 16.** *We have the limit*

$$\lim_{\nu \to 1^+} (\nu^{(1-\delta)} - 1)^{-1} \Delta\bar{E}_{\mathcal{T}}(\nu) = \frac{1}{N^{(1-\delta)}} \frac{CL\gamma^2 r}{2(1-r)^2}.$$

*Proof.* Since $\Delta\bar{E}_{\mathcal{T}}(\nu)$ increases as $\nu$ increases, we need only be concerned with the left limit of $\Delta\bar{E}_{\mathcal{T}}(\nu)$ and therefore we again focus on the top expression in Lemma 14. Define $f(\nu) :=$

$r^{\log N/\log \nu} N^{1-\delta}$ to obtain

$$\frac{\Delta \bar{E}_\mathcal{T}(\nu)}{(\nu^{(1-\delta)} - 1)} = \frac{1}{N^{(1-\delta)}} \frac{CL\gamma^2}{2} (\nu^{(1-\delta)} - 1)^{-1} \left[ \frac{1 - f(\nu)}{1 - r\nu^{(1-\delta)}} - \frac{1}{1 - r} \right]$$

$$= \frac{1}{N^{(1-\delta)}} \frac{CL\gamma^2}{2} (\nu^{(1-\delta)} - 1)^{-1} \left[ \frac{r(\nu^{(1-\delta)} - 1) - (1 - r)f(\nu)}{(1 - r\nu^{(1-\delta)})(1 - r)} \right]$$

$$= \frac{1}{N^{(1-\delta)}} \frac{CL\gamma^2}{2} \left[ \frac{r}{(1 - r\nu^{(1-\delta)})(1 - r)} - \frac{(1 - r)f(\nu)}{(\nu^{(1-\delta)} - 1)(1 - r\nu^{(1-\delta)})(1 - r)} \right].$$

Then, for the second term within the brackets containing $f(\nu)$, we have

$$\lim_{\nu \to 1^+} \frac{r^{\left(\frac{\log N}{\log \nu}\right)}}{\nu^{(1-\delta)} - 1} = -\lim_{\nu \to 1^+} r^{\left(\frac{\log N}{\log \nu}\right)} \frac{\log N}{(1 - \delta)\nu^{(1-\delta)}(\log \nu)^2} = -\lim_{y \to \infty} \frac{\log N}{(1 - \delta)} y^2 r^{(y \log N)} e^{\frac{1-\delta}{y}} = 0,$$

where the first equality applies L'Hospital's rule, the second equality makes the variable transformation $y = 1/\log \nu$, and the limit follows from $r < 1$. Applying the limit $\nu \to 1^+$ to the first term within the brackets then yields the desired result. $\qquad\square$

The proof of the final part of Theorem 6 follows from Lemma 13 and Lemma 16.

**Proof of Theorem 6 (ii):** By multiplying and dividing each term within the limits of Lemmas 13 and 16 by $(\nu - 1)$ and $(\nu^{(1-\delta)} - 1)$, respectively, we rewrite the left hand side as

$$W_\mathcal{T}(\nu) \ \Delta \bar{E}_\mathcal{T}(\nu) = ((\nu - 1)W_\mathcal{T}(\nu)) \ \frac{\nu^{(1-\delta)} - 1}{\nu - 1} \ \left( \frac{\Delta \bar{E}_\mathcal{T}(\nu)}{\nu^{(1-\delta)} - 1} \right).$$

Then, upon applying the limits in Lemmas 13 and 16 and noting that the limit of $\frac{\nu^{(1-\delta)}-1}{\nu-1}$ as $\nu \to 1^+$ is $(1 - \delta)$ by L'Hospital's rule, we obtain the desired result. $\square$

As explained in the main body of the paper, we have from (21) that a strongly-convex objective $R$ would enjoy a linear (i.e., constant factor) reduction of size $r$ in the error $R(\theta_t) - R(\theta_{\text{rob}})$ if the full batch-gradient method is applied in each iteration ($M_t = N$), for a step-size $\gamma$ chosen to satisfy the conditions of Theorem 6. The average optimality gap for our algorithm can then be written as a sum of this deterministic error and an additional term representing the stochastic error induced by the subsampling of the support. We establish in Theorem 6(i) that any general diminishing-factor growth of $M_t$ will lead to the stochastic error decreasing to zero much slower than the geometric drop in the deterministic error, and thus the stochastic error dominates. It follows that there is a suboptimal reduction in the optimality gap $E_\mathcal{T}$ w.r.t. the total computational effort $W_\mathcal{T}$, which grows in proportion to the sample size as opposed to constant-factor growth sequences.

However, constant factor sequences can trade off the rate of reduction in stochastic error against the drop in deterministic error. For strongly convex $R(\theta)$, Theorem 6 shows that our progressively sampled method is guaranteed to achieve any optimality gap of size $\epsilon + \bar{E}^*$ with $O(\epsilon^{-1})$ computational effort, where the constants contain only poly-log terms in $N$. Hence, we are able to match the guarantees on SGD for standard optimization formulations to the DRL formulation within the error limit $\bar{E}^*$, which itself drops at the rate $O(N^{-(1-\delta)})$. This is summarized in our corollary of Theorem 6.

**Corollary 2.** *Suppose all the conditions of Theorem 6 are satisfied and a solution with guaranteed expected optimality gap within $\bar{E}^* + \epsilon$ is desired. Then, there exists a $\nu_\epsilon \in (1, \infty)$ such that when Algorithm 1 is run with sample size sequence $M_t = \lfloor \nu_\epsilon^t \rfloor$, it terminates with $\mathcal{T}(\epsilon) = \lceil \log N / \log \nu_\epsilon \rceil$ steps and produces the desired solution with total computation effort $W_\mathcal{T}(\epsilon) = \frac{1}{\epsilon}\left(\kappa_1 \nu_\epsilon + \kappa_2 \nu_\epsilon(\nu_\epsilon^{1-\delta} - 1) + o(\nu_\epsilon^{1-\delta} 1)\right)$, where $\kappa_1, \kappa_2(\nu_\epsilon) \in (\eta, \infty)$ for a fixed $\eta > 0$ and for all $\epsilon \geq 0$. Moreover, $\nu_\epsilon \searrow 1$ as $\epsilon \to 0$.*

**Proof of Corollary 2:** Lemma 14 shows that the attainable expected optimality gap $\bar{E}_\mathcal{T}(\nu)$ is an increasing function of $\nu$, along with an infimum of $\bar{E}^*$. Thus, for a given $\epsilon$, there exists a parameter choice $\nu_\epsilon$ such that $\bar{E}_\mathcal{T}(\nu_\epsilon) = \bar{E}^* + \epsilon$. The increasing nature of $\bar{E}_\mathcal{T}(\nu)$ also renders that $\nu_\epsilon \searrow 1$ as $\epsilon \to 0$.

For ease of exposition, we drop the subscript $\epsilon$ in $\nu_\epsilon$ for the rest of this proof. We provide the desired result for $\nu^{1-\delta} \in (1, 1/r)$, that is the case where $\nu$ is in a neighbourhood close to 1; the case where

$r\nu^{1-\delta} > 1$ follows analogously by symmetry, as observed in the proof of Lemma 15. From the same proof, we again make the substitution $\tilde{\nu} = \nu^{1-\delta}$ and note that

$$\Delta \bar{E}_{\mathcal{T}}(\tilde{\nu}) = \frac{CL\gamma^2}{2} \ N^{-(1-\delta)} \ \left[ t_1(\tilde{\nu}) - \frac{1}{1-r} \right],$$

where $t_1(\tilde{\nu}) = (1 - r^{\mathcal{T}} N^{1-\delta})/(1 - r\tilde{\nu})$ and $\mathcal{T} = \mathcal{T}(\tilde{\nu}) = (1-\delta)\log N / \log \tilde{\nu}$. Given that $r < 1$, it follows that $\lim_{\nu \to 1+} r^{\mathcal{T}} = 0$ and hence $\lim_{\nu \to 1+} t_1(\tilde{\nu}) = 1/(1-r)$. We further follow the proof of Lemma 15 to note that

$$t_1'(1) := \lim_{\nu \to 1+} t_1'(\tilde{\nu}) = \lim_{\nu \to 1+} \frac{g(\tilde{\nu})}{(1 - r\tilde{\nu})^2} = \frac{r}{(1-r)^2},$$

where the function $g(\tilde{\nu})$ is is defined within the body of the proof as

$$g(\tilde{\nu}) = r - r^{\mathcal{T}} N^{1-\delta} \left( r - \log r \frac{\log N}{\tilde{\nu}(\log \tilde{\nu})^2(1-\delta)}(1 - r\tilde{\nu}) \right) .$$

The limit is obtained as above by noting that $\lim_{\nu \to 1+} r^{\mathcal{T}}/(\log \tilde{\nu})^2 = 0$, following the arguments in the proof of Lemma 16. Moreover, higher order derivatives can also be obtained to be finite from the arguments in Lemma 15. Thus, we have the Taylor expansion of the optimality gap as

$$\Delta \bar{E}_{\mathcal{T}}(\tilde{\nu}) = \frac{CL\gamma^2}{2} \ N^{-(1-\delta)} \ \left[ \frac{r}{(1-r)^2}(\nu^{1-\delta} - 1) + O((\nu^{1-\delta} - 1)^2) \right] . \qquad (25)$$

On the other hand, we have from Lemma 13 that

$$W_{\mathcal{T}}(\nu) = k_e \left( \frac{\nu}{\nu - 1} \right) \left\{ N \log N - (N-1) \left( \frac{\log \nu}{\nu - 1} \right) \right\}$$

$$= k_e \left( \frac{\nu}{\nu - 1} \right) \left\{ N \log N - (N-1) \left( 1 - \frac{1}{2}(\nu - 1) + O((\nu - 1)^2) \right) \right\}. \qquad (26)$$

The last expression is obtained from the Taylor expansion of $f(\nu) := \log \nu / (\nu - 1)$ around $\nu = 1$. Note that $f(\nu) < 1$ and decreasing as $\nu \nearrow$, and thus $f'(\nu) < 0$ for $\nu > 1$. Moreover, we have that $\lim_{\nu \to 1+} f(\nu) = 1$ and $\lim_{\nu \to 1+} f'(\nu) = -1/2$.

Combining (25) and (26), and using the Taylor expansion $\nu^{1-\delta} - 1 = (1-\delta)(\nu - 1) - \delta(1-\delta)(\nu - 1)^2 + O((\nu - 1)^3)$, we obtain

$$W_{\mathcal{T}}(\nu)\Delta \bar{E}_{\mathcal{T}}(\nu)$$

$$= \frac{CL\gamma^2 k_e \nu}{2N^{(1-\delta)}} \left\{ (N \log N - (N-1))t_1'(1) \left( \frac{\nu^{1-\delta} - 1}{\nu - 1} \right) + \frac{N-1}{2}t_1'(1) \left( \nu^{1-\delta} - 1 \right) + o\left( \nu^{1-\delta} - 1 \right) \right\}$$

$$= \frac{CL\gamma^2 k_e \nu}{2N^{(1-\delta)}} \left\{ (N \log N - (N-1))t_1'(1)(1-\delta) + \frac{N-1}{2}t_1'(1) \left( \nu^{1-\delta} - 1 \right) + o\left( \nu^{1-\delta} - 1 \right) \right\}.$$

This yields the desired result, with the constants

$$\kappa_1 := \frac{(1-\delta)CL\gamma^2 k_e r \left( N \log N - (N-1) \right)}{2N^{(1-\delta)}(1-r)^2} \quad \text{and} \quad \kappa_2 := \frac{CL\gamma^2 k_e r(N-1)}{4N^{(1-\delta)}(1-r)^2} . \qquad \square$$

We note that, upon substituting $r = 1 - \gamma/4c$ (refer to the statement of Lemma 12)), the value of $\kappa_1$ is seen to match the constant on the right hand side in the last line of Theorem 6(ii). This, of course, is as expected under the limit $\nu \to 1^+$.

In practical terms, Lemma 16 shows that the performance at $\mathcal{T}$ improves as $\nu \to 1^+$, but this also increases the value of $\mathcal{T}$ and hence the total effort $W_{\mathcal{T}}$. The expression in Lemma 13(a) provides practitioners with a strong grasp on the total effort that is expended up until termination $\mathcal{T}(\nu)$ for any $\nu$, and since this relation is monotonic, it can be inverted to obtain the $\nu$ closest to 1 that can be used for a fixed computational budget $W$. This in turn allows the user to obtain the best $\bar{E}_{\mathcal{T}}(\nu)$ error bound for the given budget $W$.

A forthcoming article extends these theoretical results to the case of convex losses.

# B   Empirical Results

We present additional empirical results that complement those in the main body of the paper, organized as follows. Section B.1 presents numerical comparisons among the various algorithms used to solve the DRO formulation (1), namely our PSSG algorithm, the FG algorithm of Namkoong and Duchi [21], the Giles algorithm of Levy et al. [15] and Ghosh and Squillante [10], the SGD method of Namkoong and Duchi [20] and the standard SGD (fixed minibatch $M_t = M$) method, together with the 10-fold CV regularized ERM method. Given the benefits that we established in this paper (as well as in [10]) for sampling without replacement over sampling with replacement, our empirical results focus on the Giles algorithm of Ghosh and Squillante [10]. This collection of algorithmic comparisons is performed over the datasets considered in the main body of the paper, and also includes further discussions of the results and technical details. Section B.2 provides additional details on the collection of datasets used to support our empirical results.

## B.1   Additional Results

We first present detailed empirical results that compare the performance of PSSG against FG and Giles in solving (1) over thirteen publicly available datasets, all of which are also included in Table 1 of the main body of the paper. Our aim is to study the impact of the key algorithmic parameters for each method together with that of the main tunable parameter $\rho$ of the DRO formulation (1).

**PSSG.** We investigate the impact of two key parameters of Algorithm 1, namely the sample growth $\nu$ and step length $\gamma$, on the quality of the solution produced by the PSSG method, as a function the parameter $\rho$ of the DRO formulation. Theorem 6 establishes the rate at which the expected optimality gap drops as the sample constant-growth factor $\nu \to 1^+$, and also provides how the step length $\gamma$ for the iterates affect this rate. We therefore start by keeping $\gamma = 0.5$ fixed, and present in Table 2 and Table 3 misclassification percentage results for $\nu$ varying over $\{1.001, 1.01, 1.1\}$. For each dataset, the PSSG setting(s) that produced the best performance for each value of $\rho$ are highlighted in bold, with an emphasis towards providing guidance on one setting of $\nu$ that performed well over all $\rho$ values; and the best mean outcome(s) over all methods for each dataset is further highlighted in red. Additional settings are highlighted in italics if they match the highlighted setting in bold. The results firstly show that the value of $\nu$ seems to have a moderate impact on the solution quality over each fixed $\rho$ in a vast majority of cases. For many datasets (e.g., `hiv1`, `la1s.wc`, MNIST, `rcv1`, `riccardo` and `tr31.wc`), the misclassification error is seen to basically follow the trend of increasing with $\nu$, which is consistent with the prediction from Theorem 6 for the (expected) robust loss function. Moreover, the results for the value of $\rho$ from the $O(\sqrt{d/N})$ principle seem to do well in almost all datasets. The same holds in the reverse direction when $\nu$ is fixed and $\rho$ is varied. It is also clear that $\nu = 1.001$ performs best most consistently, thus motivating our use of this value in the results of the main body of the paper. We lastly note that PSSG provides the best mean outcomes (highlighted in bold red) for all but two datasets, and even in the case of these latter two particular datasets (`fabert`, `imdb.drama`), PSSG provides comparable quality performance with overlapping confidence intervals (CIs) (highlighted in bold).

Next, for the RCV1 dataset, Figure 4 (left) contrasts the performance of PSSG keeping $\gamma = 1.0$ fixed while varying $\nu$. Our results show that the performance is insensitive to $\nu$ for values smaller than 1.001. Theorem 6, which applies only when individual losses $l(\theta, \xi)$ are strongly convex, predicts the existence of an error floor $\bar{E}^*$ as $\nu \to 1^+$; the results in the plot suggest that a similar error floor is reached for $\nu = 1.001$. Figure 4 (right) contrasts the performance of PSSG by fixing $\nu = 1.001$ and varying $\gamma$, where our results show that, while $\gamma$ has some effect, PSSG is relatively insensitive to the chosen step length beyond $\gamma = 0.5$. This motivates our choice of $\gamma = 0.5$ for all experiments in the main body of the paper.

**FG.** This method uses the classical Armijo backtracking line-search algorithm [22] to compute the step length to follow in each iteration. The maximum (initial) step length forms an important parameter for the Armijo line-search, since a small initial value might subject the iterates to the local irregularities of the optimization surface and lead to suboptimal local optima, while too aggressive of an inital step length might produce significant variability. Accordingly, Table 2 and Table 3 present our study of the impact of varying the initial maximum step length over $\{0.1, 0.5, 1.0\}$ on the quality of the solution produced by the FG method, as a function of the set of $\rho$ values of the DRO formulation. We observe that the step length of $0.1$ seems to more consistently produce poorer quality

| Algorithm | | FG | | | Giles | | | | PSSG | | |
|---|---|---|---|---|---|---|---|---|---|---|---|
| | | Maximum step-length | | | Minimum $M_t = 2^k + 1$, where $k$ is | | | | Sample growth factor $\nu$ | | |
| Dataset | $\rho$ | 0.1 | 0.5 | 1.0 | 1 | 2 | 3 | 4 | 1.001 | 1.01 | 1.1 |
| adult 45222 | 0.001 | 18.5±1.2 | 17.6±0.0 | 17.4±0.0 | **16.7±0.1** | **16.7±0.1** | 16.9±0.2 | 17.0±0.1 | **16.6±0.1** | *16.7±0.1* | 17.2±0.0 |
| | 0.01 | 17.9±0.2 | 17.4±0.1 | 17.3±0.1 | **16.7±0.1** | **16.7±0.1** | *16.8±0.1* | 17.0±0.1 | **16.6±0.1** | *16.7±0.1* | 17.1±0.0 |
| | 0.1 | 17.7±0.0 | 17.3±0.0 | **17.1±0.1** | **16.7±0.1** | *16.8±0.1* | *16.8±0.1* | 17.0±0.2 | **16.6±0.1** | **16.6±0.1** | 17.0±0.0 |
| | 1.0 | 19.0±0.3 | 19.6±0.6 | 19.6±0.6 | 18.0±0.2 | 17.4±0.5 | 17.2±0.1 | 17.4±0.2 | 22.8±0.6 | 23.7±0.1 | 16.9±0.1 |
| fabert 8237 | 0.001 | 20.7±0.4 | 20.9±0.3 | 19.2±0.8 | *10.0±0.3* | 10.2±0.3 | 10.7±0.3 | 11.2±0.3 | 9.8±0.1 | 10.9±0.3 | 16.6±0.2 |
| | 0.01 | 19.9±0.3 | 18.5±0.3 | 16.6±0.8 | **9.9±0.2** | *10.1±0.2* | 10.6±0.3 | 11.0±0.2 | 9.6±0.1 | 10.3±0.2 | 15.2±0.2 |
| | 0.1 | 14.4±0.2 | 13.6±0.4 | 12.8±0.3 | **9.9±0.3** | *10.0±0.3* | 10.5±0.2 | 10.8±0.3 | 9.6±0.3 | 9.9±0.1 | 11.8±0.3 |
| | 1.0 | 9.5±0.2 | *9.2±0.2* | **9.1±0.2** | 11.1±0.6 | 10.9±0.2 | 12.1±1.2 | 11.5±0.6 | 9.4±0.3 | **9.2±0.2** | *9.3±0.2* |
| gina_agnostic 3468 | 0.001 | 14.6±0.1 | **13.2±0.3** | 13.7±0.3 | 16.3±1.1 | 19.2±2.4 | 16.4±0.6 | 16.9±0.7 | 15.0±0.3 | 13.8±0.2 | *13.2±0.2* |
| | 0.01 | 14.6±0.2 | 13.8±0.2 | 13.9±0.2 | *15.5±0.4* | *15.7±0.3* | 16.1±0.7 | 16.5±0.6 | 15.3±0.3 | 13.7±0.3 | 13.4±0.1 |
| | 0.1 | 14.2±0.2 | 14.0±0.2 | 14.2±0.2 | *15.8±0.8* | **15.4±0.5** | 16.7±0.6 | 17.7±0.3 | 15.3±0.5 | 14.7±0.4 | **13.1±0.2** |
| | 1.0 | 17.4±1.0 | 16.9±0.9 | 16.5±0.9 | 23.2±3.7 | 20.3±0.7 | 20.6±2.1 | 19.9±1.0 | 15.1±0.4 | 17.2±1.0 | **13.1±0.3** |
| gina_prior 3468 | 0.001 | 15.3±0.6 | **13.7±0.4** | **13.7±0.4** | *14.3±1.0* | *14.9±0.8* | 15.0±0.5 | 16.5±1.0 | 13.6±0.3 | *13.0±0.4* | 12.9±0.5 |
| | 0.01 | 15.2±0.6 | *14.0±0.5* | *14.0±0.5* | *14.5±1.0* | *14.9±0.6* | 15.5±0.8 | 15.6±1.0 | 14.0±0.3 | 13.3±0.7 | **12.7±0.5** |
| | 0.1 | 14.8±0.5 | 14.2±0.6 | 14.2±0.6 | *14.8±0.7* | *14.9±0.8* | 15.0±0.5 | 16.3±0.5 | 14.6±0.5 | 14.7±0.5 | **12.7±0.4** |
| | 1.0 | 15.4±0.7 | 16.5±0.9 | 16.5±0.8 | 16.5±0.9 | 17.5±1.8 | 20.4±1.6 | 17.7±1.3 | 14.4±0.5 | 16.9±0.5 | *12.8±0.5* |
| guillermo 20000 | 0.001 | 33.6±0.8 | 31.3±0.2 | 31.4±0.2 | 35.5±1.2 | *34.3±0.3* | 34.7±0.4 | **34.1±0.5** | 32.2±0.2 | 33.2±0.3 | **30.7±0.3** |
| | 0.01 | 32.4±0.3 | 31.2±0.2 | 31.0±0.2 | **34.1±0.4** | 34.6±0.3 | 35.0±0.3 | 34.8±0.1 | 32.7±0.3 | 34.2±0.4 | 31.0±0.3 |
| | 0.1 | **30.7±0.3** | 31.0±0.3 | 31.2±0.3 | 35.7±0.2 | 35.6±0.5 | 35.1±0.4 | 35.0±0.5 | 34.5±0.8 | 37.4±0.2 | 31.0±0.3 |
| | 1.0 | 40.9±1.5 | 41.1±0.6 | 41.5±0.8 | 37.5±0.6 | 36.4±0.5 | 36.1±0.4 | 36.0±0.6 | 39.2±0.3 | 40.8±0.2 | *30.8±0.5* |
| hiv1 5830 | 0.001 | 15.0±0.2 | 8.5±0.3 | 7.6±0.2 | *6.4±0.3* | *6.5±0.2* | 6.6±0.1 | 7.2±0.4 | 6.0±0.0 | 6.0±0.1 | 6.7±0.2 |
| | 0.01 | 7.1±0.6 | 8.5±0.1 | 7.1±0.1 | *6.4±0.3* | 6.6±0.4 | 7.0±0.3 | 7.0±0.2 | 6.1±0.0 | 6.2±0.1 | 6.5±0.2 |
| | 0.1 | 8.3±0.9 | 6.9±0.2 | 6.7±0.1 | **6.3±0.2** | 6.7±0.3 | 7.1±0.3 | 7.4±0.5 | 6.0±0.2 | 6.3±0.1 | 6.3±0.1 |
| | 1.0 | 6.1±0.5 | *6.0±0.2* | **5.9±0.1** | 7.6±0.4 | 8.2±0.6 | 8.5±0.6 | 9.1±0.6 | **5.8±0.1** | 6.1±0.1 | 6.3±0.0 |
| IMDB.drama 120919 | 0.001 | *36.2±0.1* | *36.2±0.1* | **36.1±0.1** | 37.0±0.1 | 37.6±0.2 | 38.2±0.2 | 39.1±0.3 | 36.7±0.0 | 36.8±0.1 | 36.8±0.1 |
| | 0.01 | *36.2±0.1* | 37.1±0.5 | 36.5±0.3 | 37.2±0.1 | 37.6±0.1 | 38.3±0.2 | 39.3±0.3 | 36.7±0.1 | 36.8±0.1 | 36.9±0.1 |
| | 0.1 | 38.8±0.1 | 39.4±0.6 | 38.5±0.2 | 37.2±0.1 | 37.8±0.1 | 38.1±0.2 | 39.3±0.2 | **36.2±0.1** | **36.2±0.1** | **36.2±0.1** |
| | 1.0 | 43.8±0.3 | 42.9±0.4 | 42.9±0.9 | 41.3±0.5 | 40.4±0.2 | 39.3±0.3 | 39.8±0.4 | 36.3±0.1 | **36.2±0.1** | **36.2±0.1** |
| la1s.wc 3204 | 0.001 | 28.4±0.2 | 12.4±0.0 | 11.2±0.1 | 9.3±0.4 | **8.3±0.3** | 8.7±0.5 | 9.1±0.4 | 8.6±0.1 | 9.2±0.1 | 10.7±0.1 |
| | 0.01 | 26.6±2.0 | 12.2±0.2 | 11.1±0.2 | 8.7±0.5 | 9.6±1.0 | **8.3±0.3** | 8.7±0.5 | 8.6±0.1 | 8.9±0.1 | 10.4±0.2 |
| | 0.1 | 15.1±0.6 | 11.0±0.2 | 10.8±0.2 | 10.5±0.9 | *8.5±0.4* | 8.8±0.6 | 10.0±1.0 | **8.2±0.2** | 8.5±0.2 | 9.8±0.2 |
| | 1.0 | 9.7±0.3 | 9.8±0.4 | **9.3±0.1** | 14.0±2.0 | 14.4±4.1 | 13.4±2.5 | 13.9±3.0 | *8.3±0.2* | 8.7±0.2 | 9.0±0.2 |
| MNIST 3001 | 0.001 | 3.7±0.2 | 2.8±0.2 | 2.7±0.3 | 3.3±1.6 | **1.7±0.1** | 3.6±3.2 | 2.6±0.6 | 1.6±0.1 | 1.8±0.1 | 2.0±0.1 |
| | 0.01 | 3.4±0.2 | 2.2±0.1 | **2.0±0.1** | 2.2±0.3 | 1.9±0.1 | 1.9±0.3 | 2.0±0.2 | **1.5±0.0** | 1.6±0.1 | 2.1±0.1 |
| | 0.1 | 3.0±0.2 | 2.4±0.2 | 2.5±0.2 | 2.4±0.6 | *1.8±0.3* | 2.2±0.4 | 2.1±0.3 | 1.6±0.1 | 1.6±0.2 | 1.9±0.2 |
| | 1.0 | 2.7±0.2 | 3.1±0.4 | 3.1±0.2 | 3.1±1.2 | 4.3±1.1 | 2.4±0.4 | 2.4±0.6 | 1.7±0.1 | 1.6±0.2 | 2.0±0.1 |
| OVA_Breast 1545 | 0.001 | *3.3±0.1* | **3.2±0.1** | **3.2±0.1** | 6.4±0.5 | 7.5±1.4 | 8.3±3.5 | 11.3±4.9 | *3.1±0.1* | 3.3±0.1 | 3.4±0.1 |
| | 0.01 | **3.2±0.1** | **3.2±0.1** | **3.2±0.1** | 4.5±0.5 | 5.8±4.1 | *3.9±0.7* | **3.8±0.4** | 3.2±0.1 | 3.2±0.1 | 3.3±0.1 |
| | 0.1 | **3.2±0.1** | **3.2±0.1** | **3.2±0.1** | 5.7±1.0 | *3.9±0.7* | **3.8±0.6** | 5.1±1.9 | 3.2±0.1 | *3.1±0.1* | **3.0±0.1** |
| | 1.0 | 3.5±0.3 | 3.6±0.4 | 4.2±0.5 | 6.8±0.0 | 12.2±4.2 | 8.3±2.0 | 5.7±1.1 | 3.3±0.2 | 3.2±0.1 | *3.1±0.2* |

Table 2: Comparisons of the three DRO algorithms (FG, Giles, PSSG) over the first 10 of the 13 publicly available machine learning datasets; refer to B.2 for details on these datasets. Each algorithm solves DRO formulations for $\rho = 0.01$, $\rho = 0.1$ and $\rho = 1.0$. For the FG algorithm, the critical parameter of the maximum step length given to the line-search subroutine is varied over $0.1$, $0.5$ and $1.0$. In the Giles algorithm experiments, the minimum sampled batch size, constructed as $2^k + 1$, is varied over $k = 1, 2, 3, 4$. The PSSG algorithm experiments vary the sample growth factor $\nu$ over $1.001$, $1.01$ and $1.1$, keeping the step-lengths fixed at $\gamma = 0.5$. All other algorithm settings are held constant over all the experimental runs. Each column provides the 95% confidence interval of the percentage misclassified over the withheld test datasets. In the collection of columns for each algorithm, the settings with either the best or next-best performance are highlighted for each $\rho$ value, if they are close to or overlapping with the overall best. We suggest the best setting to use for each method in each dataset over all values of $\rho$ by highlighting in **bold** where a clear winner is apparent, and the rest are highlighted in *italics*. The result with the best mean outcomes over all methods in each dataset is additionally colored **red**.

| Algorithm | | FG | | | Giles | | | | PSSG | | |
|---|---|---|---|---|---|---|---|---|---|---|---|
| | | Maximum step-length | | | Minimum $M_t = 2^k + 1$, where $k$ is | | | | Sample growth factor $\nu$ | | |
| Dataset | $\rho$ | 0.1 | 0.5 | 1.0 | 1 | 2 | 3 | 4 | 1.001 | 1.01 | 1.1 |
| rcv1 | 0.001 | 13.6±0.2 | 10.3±0.2 | 9.0±0.1 | 6.1±0.0 | 6.2±0.0 | 6.2±0.0 | 6.4±0.0 | 6.1±0.0 | 8.1±0.0 | 11.2±0.0 |
| 804414 | 0.01 | 12.1±0.1 | 9.8±0.2 | 8.7±0.2 | 6.0±0.0 | 6.0±0.0 | 6.1±0.0 | 6.2±0.0 | 6.0±0.0 | 7.8±0.0 | 10.2±0.0 |
| | 0.1 | 9.0±0.0 | 8.7±0.0 | 8.1±0.0 | 5.6±0.0 | 5.6±0.0 | 5.8±0.0 | 6.0±0.0 | 5.6±0.0 | 7.0±0.0 | 8.3±0.0 |
| | 1.0 | 6.0±0.0 | 5.8±0.0 | **5.7±0.0** | **5.3±0.0** | 5.7±0.0 | 6.0±0.0 | 6.3±0.1 | **5.1±0.0** | 5.5±0.0 | 5.8±0.0 |
| riccardo | 0.001 | 24.1±0.1 | 7.9±0.5 | 5.1±0.4 | 2.7±0.1 | 2.5±0.2 | 2.6±0.1 | 3.1±0.2 | *1.6±0.1* | 2.9±0.1 | 8.6±0.1 |
| 20000 | 0.01 | 17.9±0.6 | 6.4±0.6 | **4.9±0.4** | 2.3±0.1 | 2.3±0.1 | **2.0±0.1** | 2.5±0.3 | *1.6±0.1* | 2.4±0.1 | 7.3±0.1 |
| | 0.1 | 12.5±0.5 | **4.9±0.4** | 10.9±1.6 | 2.2±0.1 | *2.1±0.2* | 2.1±0.2 | 2.8±0.2 | **1.5±0.1** | 1.8±0.1 | 5.5±0.1 |
| | 1.0 | *5.0±0.2* | 5.8±0.3 | 5.2±0.4 | 4.8±2.6 | 6.0±3.5 | 5.1±1.6 | 5.5±2.4 | 1.7±0.1 | **1.5±0.1** | 4.4±0.2 |
| tr31.wc | 0.001 | 19.0±1.6 | 9.1±1.0 | 6.3±0.4 | 6.8±2.6 | 5.9±2.6 | 6.8±3.7 | 4.2±0.8 | 2.8±0.2 | **2.7±0.3** | 4.2±0.5 |
| 927 | 0.01 | 13.5±0.5 | 6.9±0.6 | 4.7±0.7 | 4.7±1.2 | 3.4±0.8 | 3.6±0.8 | *3.1±0.4* | 2.8±0.2 | 2.8±0.2 | 3.1±0.4 |
| | 0.1 | 9.8±0.5 | 4.9±0.4 | 3.7±0.4 | 5.3±1.9 | **2.7±0.7** | *3.1±0.5* | 3.2±0.9 | 3.1±0.3 | 2.8±0.4 | 2.8±0.4 |
| | 1.0 | *2.8±0.5* | **2.7±0.3** | 3.1±0.4 | 9.5±3.9 | 13.1±5.8 | 7.6±5.3 | 9.6±4.2 | **2.7±0.3** | 3.2±0.5 | *2.8±0.3* |

Table 3: Comparisons of the three DRO algorithms (FG, Giles, PSSG) over the remaining 3 of the 13 publicly available machine learning datasets; refer to B.2 for details on these datasets. The details on the experimental setup used to produce these results is exactly as given in the caption of Table 2.

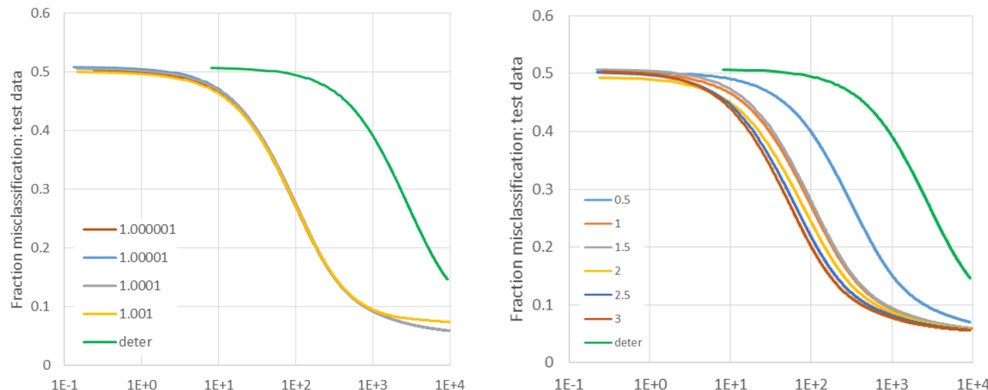

Figure 4: The *average* misclassification performance of the FG (green lines, 'deter' in legend) and PSSG (other colored lines) algorithms over the `rcv1` dataset: (left) keeping $\gamma = 1.0$ fixed and varying $\nu$ from 1.000001 to 1.001 for PSSG; and (right) keeping $\nu = 1.001$ fixed and varying $\gamma$ from 0.5 to 3 for PSSG. Log-scale computation time (in seconds) on $x$-axis.

solutions. The step lengths of 0.5 and 1.0 have an almost equal share of good quality solutions over the datasets and differing $\rho$, and thus we employed 0.5 as the maximum step length in the results of the main body of the paper. Here the DRO parameter $\rho$ does have a more pronounced impact on the solution quality in comparison with PSSG.

**Giles.** This method is subject to the most variability among the three DRO methods, primarily due to the added noise of the mini-batch size randomization at each iteration and the form of its gradient estimate. As evident from the Giles columns of Table 2 and Table 3, the CIs of the test misclassification estimates produced by Giles are the widest of the three DRO methods.

A key parameter that allows some control on the variability of the Giles method is the minimum mini-batch size that can be sampled. The mechanics of the method requires sizes of the form $M = 2^k + 1$; we refer to Levy et al. [15] and Ghosh and Squillante [10] for a detailed description. Table 2 and Table 3 present our study of the impact of varying the value of $k$ over $\{1, 2, 3, 4\}$ on the quality of the solution produced by the Giles method, as a function of the set of $\rho$ values of the DRO formulation. As apparent from the results, it is hard to pick one "good" setting for the value of $k$ that provides the best or near-best performance over all datasets, and for any value of $k$ there are datasets where performance is significantly worse for all values of $\rho$. The value of $k$ also interacts markedly with the

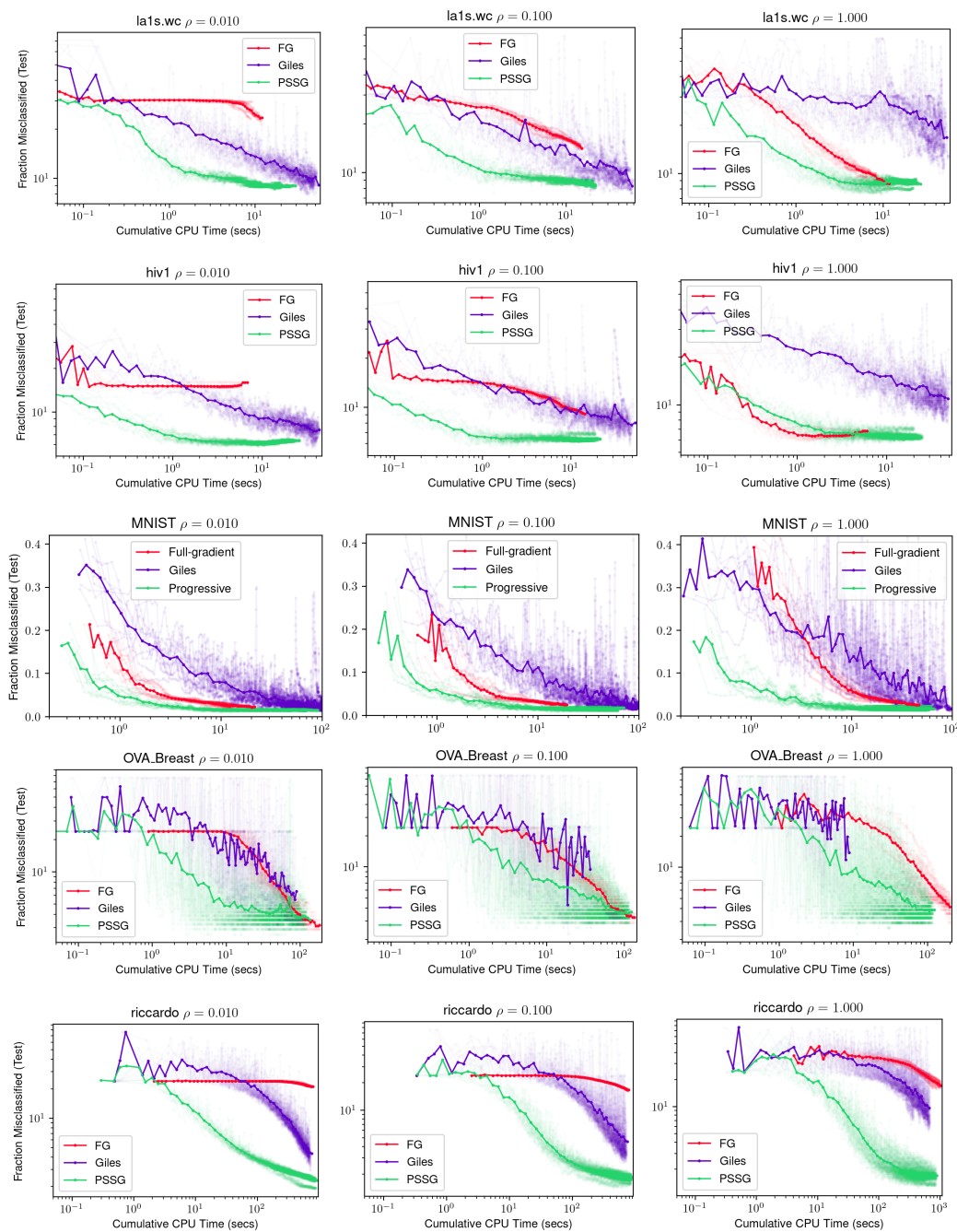

Figure 5: Comparisons of PSSG (green), FG (red) and Giles (purple) on fraction of misclassification in testing ($y$-axis) versus cumulative CPU times ($x$-axis in log-scale) over a representative collection of five datasets with (left column of plots) $\rho = 0.01$, (middle column of plots) $\rho = 0.1$, and (right column of plots) $\rho = 1.0$. The five datasets are (top row of plots) `la1s.wc`, (second row of plots) `hiv1`, (third row of plots) MNIST, (fourth row of plots) `OVA_Breast`, and (bottom row of plots) `riccardo`.

$\rho$ values over each dataset, with many $\rho$ settings producing drastically poorer results. These results indicate that no single setting of $k$ should be attempted, and instead parameter tuning needs to be performed to identify the value of $k$ that best fits the dataset and the $\rho$ value.

**Optimization Iterate Paths.** Figure 5 presents a detailed comparison of the paths of the optimization runs under the PSSG, FG and Giles algorithms over a representative collection of five publicly available datasets also considered in Table 2 and Table 3. For each algorithm, the fractional misclassification performance over the testing data from 10 experimental runs are presented, where each run uses a different random partition of the data into training and testing datasets. An average over these 10 runs is also plotted with a thicker line. For each dataset, the leftmost plot solves (1) with $\rho = 0.01$, the middle plot solves (1) with $\rho = 0.1$, and the rightmost plot solves (1) with $\rho = 1.0$. Each algorithm uses the best parameter setting overall judged from the results of Table 2 and Table 3, and thus PSSG uses $\nu = 1.001$ and $\gamma = 0.5$ in all cases. FG sets its maximum step length to $0.5$ for `riccardo` and to $1.0$ for the remaining datasets. Giles sets its minimum mini-batch size parameter $k$ to 1 for `hiv1`, to 3 for `MNIST` and `riccardo`, and to 4 for `OVA_Breast`.

Consistent with our results in the main body of the paper, we observe that PSSG is significantly faster than FG for every value of $\rho$ considered. Each of the iterations of the FG method bears significant computational cost, while the initial iterations of the PSSG method utilize small mini-batch sizes with a light computational burden. This allows the PSSG method to open up a sizeable lead from the start, and it reaches the best test-misclassification values faster than the other two methods. Further consistent with our results in the main body of the paper, we observe that PSSG outperforms Giles for each value of $\rho$ considered. The iteration paths of the Giles method are visibly impacted by the additional variance induced by the mini-batch Monte Carlo randomization step. This results in a slowdown of the convergence of the method, and also in the wider CIs in the outcomes, as has been noted earlier. We observe that Giles sometimes outperforms FG and other times FG outperforms Giles, but once again PSSG outperforms both in each case.

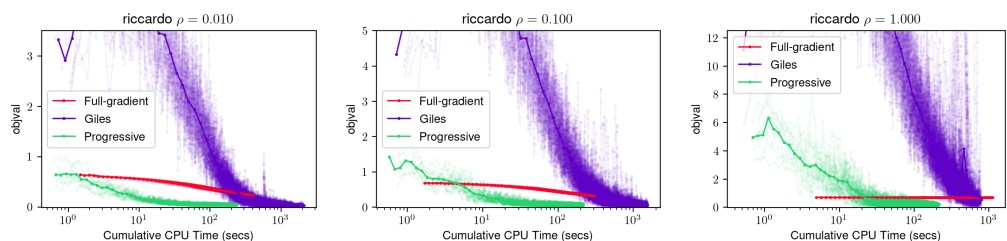

Figure 6: Comparisons of PSSG (green), FG (red) and Giles (purple) on training robust loss (minimization objective) ($y$-axis) versus cumulative CPU time ($x$-axis, differing ranges) over the `riccardo` dataset with (left) $\rho = 0.01$, (center) $\rho = 0.5$ and (right) $\rho = 1.0$. Log-scale cumulative samples $x$-axis.

**Robust loss Objective** $R(\theta_t)$**.** Figure 6 presents representative comparisons among the PSSG, Giles and FG methods on the `riccardo` dataset with respect to the DRO formulation objective, i.e., the obtained robust training loss. Note that the Giles estimate of the robust loss not only suffers a high noise factor which remains relatively constant as the iteration count grows, but the iterations of this method seem to indicate an initial phase where the objective function *worsens*. As $\rho$ increases, the noise and the magnitude of the initial movement in an adverse direction of the Giles method worsen even further. This can potentially lead to premature termination of the algorithm if the criterion were to monitor the training robust loss values; recall the our implementation terminates when the mis-classification loss on a small held-out dataset from the training set shows no further signs of improvement.

In strong contrast, the noise in the robust loss estimate under the PSSG algorithm shrinks. The latter happens because of the increasing batch size under PSSG as the iterate count grows, which ensures that algorithm termination with the robust loss criterion will perform well. The FG method shows almost no variability, as can be expected; however, it is again prohibitively expensive to compute and slow to converge.

**SGD Comparisons.** Theorem 3 in the main body of the paper identifies the bias faced when a fixed mini-batch size SGD algorithm is employed to solve the DRO formula-

tion (1). The next set of results investigates the size of this bias over the `rcv1` dataset.

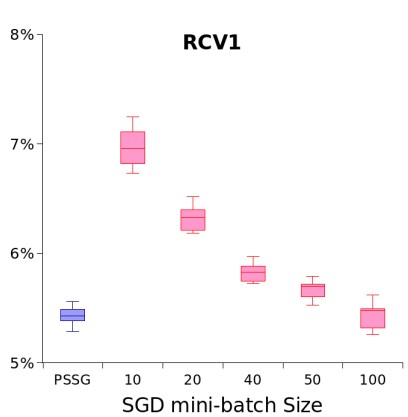 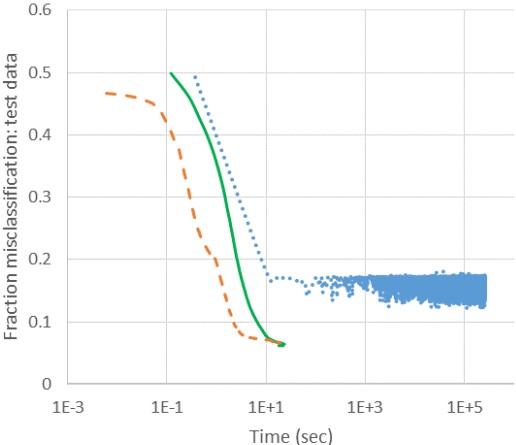

Figure 7: An evaluation of the bias suffered by standard SGD over the `rcv1` dataset for different batch sizes compared with that of the DRO formulation solved by our PSSG algorithm.

Figure 8: Comparisons of PSSG (orange, dashed lines), FG (green, solid lines), and a single run of the Namkoong and Duchi [20] algorithm (blue) on fraction of misclassification in testing ($y$-axis) versus computation times ($x$-axis) over the `hiv1` dataset with $\rho = 0.1$ and log-scale $x$-axis.

Figure 7 presents a comparison of the performance of standard SGD over various mini-batch sizes against our PSSG algorithm. Empirical CI estimates from 10 experimental runs for PSSG and the fixed-batch ($M_t = M$) SGD are presented for the fractional misclassification performance over the testing dataset. The results show that the SGD method reports a higher misclassification error than the solutions of our PSSG method, which is as a result of the bias identified in Theorem 3. Recall that this bias arises because a small fixed subsample size in each iteration can easily miss those elements $\xi$ of the training dataset that suffer from high loss $l(\theta_t, \xi)$ at the current iterate $\theta_t$, thus yielding an optimistic estimate of the robust loss $R(\theta_t)$ and prematurely terminating the search for a robust solution to (1). The results show that, for the `rcv1` dataset, this bias in the outcome drops to insignificance only as $M \to 100$. Hence, our DRO algorithm with its progressively grown subsample size avoids the expense of the hyper-parameter tuning of the batch size required by standard SGD for bias reduction.

This conclusion is further supported by the results in Figure 9, which presents the test misclassification errors from the standard SGD and PSSG methods over the `rcv1` dataset for various settings of $\rho$ and illustrate the relative insensitivity of the output of PSSG to values of $\rho$ from 0.001 to 1. The batch size of 10 for standard SGD does indeed produce bias, which in turn affects its performance in estimating solutions for the DRO formulation (1). However, a complex dependency exists between the bias of the standard SGD algorithm and the parameter $\rho$ of the DRO problem formulation, which further reiterates the message that PSSG saves on not having to tune the batch size of the standard SGD for each instance of the DRO problem. Therefore, the main advantages of our DRO PSSG algorithm include that it does not need any such parameter tuning and it efficiently provides a high-quality solution to (1).

For one of the smallest datasets (`hiv1`), we also consider performance comparisons with the primal-dual method proposed in Namkoong and Duchi [20] that attempts to address the bias in standard SGD as a solver for the DRO formulation (1). Figure 8 compares the fractional test misclassification loss of PSSG and FG against the corresponding results for a single run of the primal-dual proximal algorithm of Namkoong and Duchi [20]. The latter algorithm is computationally prohibitive, running for over 2 days, and the end results fall short of the optimal solution. This is as anticipated based on the corresponding discussion in the introduction.

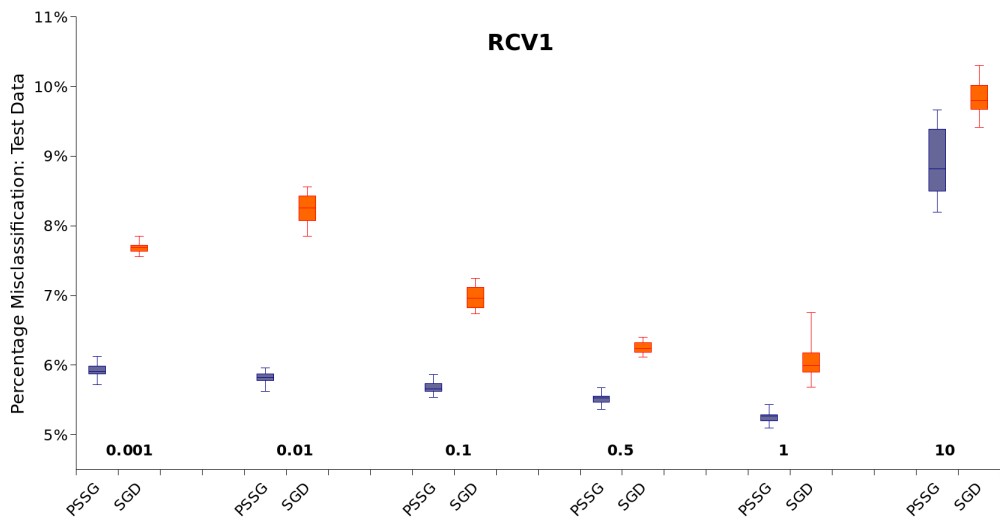

Figure 9: Comparisons of the misclassification performance of the standard SGD (red boxes) and PSSG (blue boxes) algorithms over the `rcv1` dataset, keeping algorithm parameters fixed and varying $\rho$. The $x$-axis labels contain the algorithm name and the $\rho$ value used.

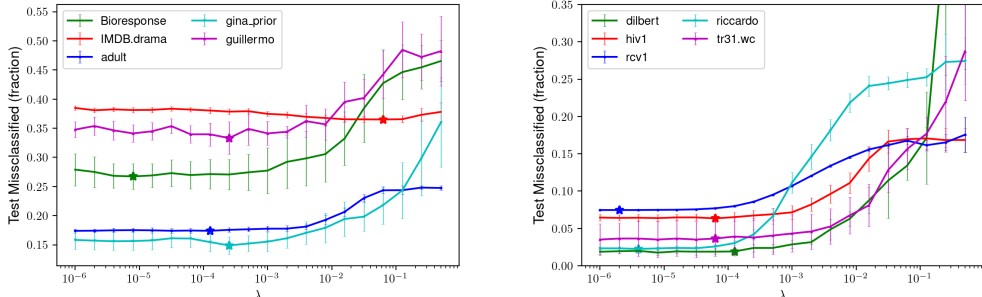

Figure 10: Illustration of optimizing the parameter $\lambda$ for the ERM regularized formulation by enumeration, with the average test misclassification (%) over 10 partitions of the data plotted on the $y$-axis versus $\lambda$ on the $x$-axis (in log-scale). The left plot shows results for five datasets that achieve a best performance of $15\%$ or higher and the right plot shows results for five datasets that achieve a best performance of $9\%$ or lower.

**Regularization of ERM.** Figure 10 presents the outcome of the full set of enumerations for a representative sample of 10 datasets to elicit the best value of the penalty parameter $\lambda$ in the regularized ERM objective of $L_{U_N}(\theta) + \lambda\|\theta\|^2$. The SGD algorithm is used to solve each instance of the regularized formulation in each of the 10 partitions of the dataset. A mini-batch size of 10 was used along with a step size sequence $\gamma_t = 0.25 * (5000./(5000. + t))$. The plots on the left provide the regularization enumeration for five datasets that achieved a best test misclassification error of 15% or higher, whereas the plots on the right provide the regularization enumeration for five datasets that achieved a best test misclassification error of 9% or lower. The $\lambda$ value chosen by the backtracking enumeration approach (described in Section 3) is marked for each dataset.

It is clear from these results in Figure 10 that the hyperparameter tuning of $\lambda$ is non-trivial. The shape of the curve of the mean outcomes of the regularization formulation varies significantly across the datasets, and moreover the variability exhibited is significantly impacted by the dataset characteristics. While generalization performance seems to improve as $\lambda \to 0$ (except for `imdb.drama`), it is not clear that a single $\lambda$ value can be picked to perform well over all the datasets. This is in sharp contrast to the DRO formulation, where the choice of $\rho = 0.1$ or $\rho = 1.0$ (as the most likely orders of the $\sqrt{d/N}$ values) provides good generalization performance.

Recall that PSSG provides this level of performance by solving a single instance of the DRO formulation (1), thus avoiding the burdensome 10-fold CV enumeration. As observed in the discussion of the CPU times provided in Table 1, the time taken by PSSG to solve each DRO formulation is on average of the same order as that taken by SGD to solve a single instance of the ERM formulation for a single $\lambda$ value. This indicates a significant computational savings in using our DRO approach because of the elimination of the expensive hyper-parameter tuning step, with the time required under PSSG being one to two orders of magnitude superior to the time taken by the ERM regularization (enumeration with backtracking) procedure.

**In Summary.** Our empirical results above and in Section 3 support our theoretical results and show that PSSG achieves the main objectives of reaching optimal solutions of the DRO formulation and improving model generalization more consistently and more quickly than other methods. In particular, PSSG provides models of equal or better quality as those from FG, Giles and regularized ERM but with significantly less computational effort, even orders of magnitude less effort in many cases, and thus provides a strong alternative machine learning approach to improve model generalization. Once again, the main advantages of our DRO algorithm include that it does not need any further parameter tuning, it efficiently provides a solution to (1), and it naturally provides a strong generalization guarantee.

## B.2   Experimental Datasets

Our empirical results are based on thirteen publicly available machine learning datasets that were obtained from UCI [17], OpenML [6], MNIST [14] and SKLearn [16]. We briefly highlight here some of the details of a representative sample of these datasets. The *HIV-1 Protease Cleavage* dataset helps develop effective protease cleavage inhibitors by predicting whether the HIV-1 protease will cleave a protein sequence in its central position ($y = 1$) or not ($y = -1$). After preprocessing following [25], this dataset has $N = 5830$ samples of $d = 160$ feature vectors using orthogonal binary representation, of which 991 are cleaved and 4839 are non-cleaved. The *Adult Income* dataset comprises $N = 48842$ observations of 14 attributes used to predict whether the annual income of each adult is above \$50K ($y = 1$) or not ($y = -1$). Using binary encoding of the categorical attributes, the data is transformed into ($d = 119$)-dimensional features. The *Reuters Corpus Volume 1* (RCV1) dataset comprises $N = 804414$ samples each with $d = 47236$ features. The purpose of this dataset is to classify each sample article as either belonging to a corporate/industrial category ($y = 1$) or not ($y = -1$) based on its content. The *Riccardo* dataset has $d = 4296$ features and a moderate count of $N = 20000$ samples, of which 5000 are labeled as class 1 ($y = 1$) and the remaining 15000 are not ($y = -1$).

Additional details on these four datasets, as well as full information on all thirteen publicly available machine learning datasets considered in our experiments, can be obtained from the corresponding publicly available sources UCI [17], OpenML [6], MNIST [14] and SKLearn [16], as noted in Table 1.