# OpenReview forum: "Efficient Generalization with Distributionally Robust Learning"
_NeurIPS.cc/2021/Conference — NeurIPS 2021 Poster_

### Official Review · Reviewer_6QkA · 2021-07-17

**Rating:** 6
**Confidence:** 3

**Summary:**

This paper develop and analyze an efficient double-loop-based stochastic algorithms for solving DRO problems.

**Ethics Review Area:**

["I don’t know"]

**Main Review:**

This paper is mainly an extension of Namkoong and Duchi [21]. The authors utilize Danskin's theorem to ensure the existence and form of subgradient of the inner function, thus can avoid solving the min-max problem directly. As the algorithm scheme mimics "bilevel optimization", the author significantly reduces the expense of solving the inner maximization over the full training dataset by subsampling technique in the iterates of the algorithm and estimating the robust loss via a sample average approximation. Moreover, the authors can control the inherent bias from the subsampling by progressively increasing the subsample size with the iterates up to the maximum size N. This doubling trick has been widely used in the stochastic second-order methods, which is nothing new and in expection.


Overall, the paper is well-written and easy to follow. However, the author didn't submit the supplementary material. Thus, I cannot check the correctness of the technical results and quantify the technical contributions.

**Time Spent Reviewing:**

2 hours

---

> ### Author Response · Authors · 2021-08-09
> **Author Response to Reviewer 6QkA**
>
> We would like to thank the reviewer for their thoughtful and constructive feedback on our work. We regret that a few important points were not sufficiently clear to the reviewer, which we will now attempt to clarify.
>
>
> •	While it is true that our work builds on the full-gradient approach of Namkoong &Duchi [21], no previous work considers an approach based on *subsampling* the training dataset in a *progressively increasing manner* to reduce the computational burden of Namkoong &Duchi [21] which uses the entire training dataset at each iteration. In particular, their full-gradient algorithm requires $O(N)$ computational effort in each iteration, where $N$ is the training dataset size. Our main focus is to greatly reduce the dependence of this computational effort to $O(\log N)$ via a carefully chosen subsampling approach.
>
> •	Moreover, our subsampling approach is fundamentally based on *sampling without replacement*, whereas the recent work of Levy et al. [15] building on Namkoong &Duchi [21] takes an approach that is very different from our progressively increased subsampling and that is based on *sampling with replacement*. We show that the latter has critically important deficiencies. More importantly, we show that our method is able to match the standard SGD convergence guarantees of $O(1/\epsilon)$ with respect to the desired accuracy $\epsilon$, which is something that the follow-on work of Levy et al. [15] cannot achieve.
>
> •	Although progressively increased subsampling has been used to a limited extent in other application domains (namely, in optimization for variance reduction of gradient estimates) and sampling without replacement is a well-known probabilistic technique, neither approach has been proposed nor even considered for bias reduction of gradient estimates, making our approach distinct from previous related work and, to the best of our knowledge, the first to propose increased subsampling for bias reduction. Even more importantly, we provide a thorough mathematical analysis of such an approach that combines progressively increased subsampling and sampling without replacement to understand bias reduction of gradient estimates and to understand the fundamental tradeoff between computational effort and stochastic error. And, as noted above, our approach provides greatly improved convergence guarantees with great reduction in the computational effort.
>
> •	We proceed to exploit the theoretical results from our new mathematical analysis of our proposed algorithm in order to set the key parameters of our algorithm with the goals of realizing the performance guarantees established in the paper and balancing the fundamental tradeoff between computational effort and stochastic error in an optimal manner. This is further supported by our empirical results which clearly demonstrate that our algorithm reaches the true optimal solutions of the DRO formulation more consistently and more quickly in comparison to other forms of distributionally robust learning. In particular, our algorithm is shown to provide better quality solutions than 10-fold cross validation with orders of magnitude reduction in computational effort; and shown to provide better or equivalent quality solutions than other DRO methods with great reduction in computational effort.
>
>
> We will be sure to explain each of these points more clearly in the paper.
>
>
> We would also like to point out that, due in large part to our new mathematical analysis, parameter tuning of our proposed algorithm is a very lightweight task whereas the other distributionally robust learning methods are far more sensitive to the setting of their core algorithmic parameters and require substantial forms of parameter tuning.
>
> We are happy that the reviewer found our paper to be well-written and easy to follow. We are not sure why the reviewer was unable to access our supplementary material, which was included as part of our paper submission. Hopefully our supplementary material is now available to the reviewer and thus these technical details with help to further clarify our important technical contributions and results.

---

> > ### Comment · Reviewer_6QkA · 2021-08-23
> > **Response**
> >
> > Thanks for your feedback! Now, the supplementary material is accessible. I'm not sure what happened. All of my technical concerns have been addressed. I'm happy to increase my score accordingly.

---

> ### Comment · Area_Chair_6mkL · 2021-08-19
> **critical review**
>
> Since your review is the most critical one I was wondering if you can take a careful look at the authors' response and see if that changes your opinion of the paper. Also it would be good to update your review to include a bit more details (no matter whether or not you decide to change your opinion).

---

### Official Review · Reviewer_pvnB · 2021-07-17

**Rating:** 6
**Confidence:** 4

**Summary:**

The paper studies a stochastic gradient descent algorithm to solve DRO problems with phi-divergence based ambiguity set, where the gradient estimate is computed from an increasing mini-batch size. The proposed algorithm is then tested numerically to demonstrate its efficiency.

**Ethics Review Area:**

["I don’t know"]

**Limitations And Societal Impact:**

By moving procedure 1 to the main text and having a more in-depth literature review, the authors could improve the readability of the paper.

**Main Review:**

The paper makes valuable contributions, and the numerical section is extensive. A few comments about the paper:

- The authors develops an efficient algorithm to solve DRO problem. However, it is unclear to me why the proposed approach leads to better generalizations. I am a bit confused because different algorithms solving the same DRO problem should not give different results or lower misclassification error.

- How does Table 1 change if the misclassification error is replaced with averaged logloss on test samples?

- What is the stopping criteria for all algorithms reported in Table 1?

**Time Spent Reviewing:**

8

---

> ### Author Response · Authors · 2021-08-09
> **Author Response to Reviewer pvnB**
>
> We appreciate and thank the reviewer for their positive comments in general and especially concerning the valuable contributions of our paper, the extensive numerical results presented in our paper, and our efficient algorithm to solve DRO problems.
>
> We would also like to thank the reviewer for their thoughtful and constructive feedback on our work. We now provide responses to each of the important questions you have raised.
>
> •	*Performance of different DRO algorithms*: The main objective of our experiments is to reach the true optimal solutions of the DRO formulation more consistently and more quickly. Each of the three DRO algorithms studied in Table 1 are guaranteed to converge to the true optimal solution of the DRO formulation, and thus you are correct in expecting that the misclassification errors at each algorithm’s termination should also be similar. These empirical results illustrate that our Progressive algorithm achieves the main objective much better than the other two competing methods for the same termination criterion. Notice in the detailed results in Tables 2&3 that the other two algorithms do indeed get the best solution for a few specific settings of their hyper-parameters, while the Progressive method is relatively less sensitive to its hyper-parameters and achieves the best performance more consistently (as well as more efficiently). The Full-gradient and Giles DRO algorithms could also achieve this level of consistency if they were allowed to run longer; however, the focus of our experiments was primarily to understand the characteristics of the Progressive method and demonstrate our main objective to reach the true optimal solutions of the DRO formulation more consistently and more quickly.
>
> •	*Using Averaged Log-Loss*:  This is an interesting question. Our experimental setup uses a support-vector machine style *non-probabilistic* binary classifier, and so we are not able to derive a Log-Loss estimate for our classification results. This is also why the previous work in this area considered misclassification error and not log-loss as the performance objective. At the same time, the log-logistic loss function is used in the experiments as $ l (\theta, \xi) $, and this is consistent with Log-Loss in that it measures the amount by which the miss-classified  data points missed the classification plane.
>
> •	*Stopping Criteria*: The description on this point provided in Lines 379-381 is incomplete and we will amend it as follows: “Each method monitors the performance of the current model in correctly classifying the set-aside test validation set *every $1/20^{th}$ of an epoch*, and stops if the average of the last 20 misclassification fraction values does not improve more than 1% when compared with the average of the previous 80 evaluations.” Hence, an algorithm-run stops if the misclassification error in the validation set over the current epoch is not improved by more than 1% over the four past epochs. We will update the discussion to explain this better.
>
> •	We thank the reviewer for their suggestion of moving Procedure 1 to the main body and for enhancing the literature review. We will act on these suggestions in the final version.
>
> We will be sure to explain each of the above points more clearly in the paper, and will update our manuscript according to your valuable suggestions.

---

### Official Review · Reviewer_Nbtt · 2021-07-20

**Rating:** 8
**Confidence:** 3

**Summary:**

The paper studies distributionally robust optimization, with a SGD method and sampling without replacement for the inner problem.

**Limitations And Societal Impact:**

The $O(1/\epsilon)$ convergence rate is asymptotic.

**Main Review:**

The paper is well-written, clearly organized and easy to follow. The results seem to be correct though I didn't check all the proof details. The convergence analysis using sampling without replacement is novel and interesting, which is also more practical and efficient. Therefore I incline to accept the paper.

1. Can the author add more explanations to what is exactly the total computational effort? And how taking a small $\nu$ decreases computational effort.
2. Since $M_{t}$ are integers, it's not clear to me what does it mean by saying $\nu$ is a constant (it might be more problematic when $\nu \to 1+$). Can the author explain more?
3. The convergence rate $O(1/\epsilon)$ seems asymptotic in terms of $\nu$. Is it possible to derive this rate for a fixed $\nu$, with possibly rebooting sampling?)

**Time Spent Reviewing:**

6

---

> ### Author Response · Authors · 2021-08-09
> **Author Response to Reviewer Nbtt**
>
> We appreciate and thank the reviewer for their positive comments in general and especially concerning the presentation and our novel interesting convergence analysis using sampling without replacement, which is concluded to be more practical and efficient.
>
> We would also like to thank the reviewer for their thoughtful and constructive feedback on our work. We now provide responses to each of the important questions you have raised.
>
> **Qn1, first part**: The total computational effort $W_{\mathcal{T}}$ is the sum of the computational effort $w_t$ of each of the iterations of our Progressive sub-sampling Algorithm 1. As provided in the proof of Proposition 2, in each iteration the inner maximization step is the majority of the work, and this involves a sorting step of complexity $O(M_t \log M_t)$  and two sequential bisection-search steps of lower complexity. This motivates the form we choose for each iteration’s computational effort: $w_t = k_e M_t \log M_t $.
>
> **Qn 2**: For the geometrically increasing case, we state that $ M_t = \nu^t $ is used. We should have stated this more accurately to be $M_t =\lfloor \nu^t \rfloor $, which is how we implemented this sub-sampling sequence in our experiments. So, $M_t$ remains low for a number of initial iterations when $\nu$ is close to one, and only eventually does it show the rapid geometric increase in terms of integer values; a specific example is described on Lines 384-385, which is hopefully now clearer with this clarification. This also motivates why we define $ {\mathcal{T}} = \min_t\{ M_t \ge N\} $ and note that this takes the value ${\mathcal{T}} = \lceil \log N / \log \nu \rceil $ in this case. We drop the integrality requirement in the analysis in Theorem 6 to solely provide a clear and insightful exposition, but this will not change the conclusions in Theorem 6 and we will modify the paper accordingly.
>
> **Qn 1, Last part**: We intended to mean that, for a small $\nu$ , the initial iterations have a small $M_t$ and thus the Progressive method enjoys many iterations where the computational effort is small. However, this also implies a larger number of total iterations ${\mathcal{T}}$ before the full dataset size $N$ is encountered, and as a consequence, Theorem 6(ii) shows that the total computational effort $W_{\mathcal{T}}(\nu)$ is *larger* for a smaller $\nu$.
>
> **Qn 3**: You have raised an excellent point. In light of your question, we note that the proof of Theorem 6 and its related Lemmas 12-16 make it possibly to directly restate Theorem 6, for a fixed $\nu$, as follows
>
> $
> W_{{\mathcal{T}}}(\nu)(\bar{E}_{{\mathcal{T}}}(\nu) - \bar{E}^{\ast})    =    \kappa_1    +    \kappa_2 (\nu)    (\nu^{1-\delta} – 1)    +     o( (\nu^{1-\delta} – 1)),
> $
>
> where $\kappa_1$ is the constant displayed in the result in Theorem 6, and $\kappa_2 (\nu) \in (\eta,\infty) $ for all $\nu \ge 1$ for a positive constant $\eta > 0$. (We use the expression $o(\cdot)$ in the sense that the term goes to zero faster than its argument, and recall the definition of $\delta$ as a small positive constant needed in the bias result in Theorem 3.) This therefore lets us state that for a fixed $\nu > 1$,  the computational effort $ W_{\mathcal{T}}(\nu)$ and the best guaranteed error
> $
> \bar{E}_{\mathcal{T}} (\nu)
> $
> are related, up to the first order, as
>
> $
> W_{\mathcal{T}}(\nu) \approx (\kappa_1 + \kappa_2(\nu) (\nu^{1-\delta}-1))/\epsilon (\nu)
> $
> where $\epsilon(\nu) := \bar{E}_{{\mathcal{T}}}(\nu)  - \bar{E}^{\ast} . $
>
> Hence, even for fixed $\nu $ larger than but close to one, the method largely preserves the SGD-like convergence rate observed in the limit $\nu\rightarrow 1^+ $.  We thank you for raising this excellent point as a question of further interest, which our results do address but we did not highlight this important point. We will restate our Theorem 6 to include this expression and provide an exact expression for $\kappa_2(\nu)$.
>
>
> We will be sure to address each of your questions and explain each of the points you raised more clearly in the paper, and will update our manuscript accordingly.

---

### Decision · Program_Chairs · 2021-09-27

**Decision:**

Accept (Poster)

**Comment:**

This paper gives a new method for solving the distributed robust learning problem. The new method uses a subsampling technique for the inner maximization problem which lead to faster convergence. The paper also demonstrates the effectiveness of the new method in practical settings. The reviewers have raised several concerns initially about novelty and theoretical guarantees, but they were addressed in the response period. Now the reviewers believe that the paper is technically sound and the empirical results are convincing.